

# Linking source with consequences of coastal storm impacts for climate change and risk reduction scenarios for Mediterranean sandy beaches

Marc Sanuy[1], Enrico Duo[2], Wiebke S. Jäger[3], Paolo Ciavola[2], José A. Jiménez[1]

[1]Laboratori d'Enginyeria Marítima (LIM), Universitat Politècnica de Catalunya – Barcelona Tech, Barcelona, Jordi Girona 1-3, 08034, Spain
[2]Dep. of Physics and Earth Science, University of Ferrara, Via Saragat 1, Ferrara, 44122, Italy
[3]Department of Hydraulic Engineering, Delft University of Technology, Stevinweg 1, Delft, 2628 CN, The Netherlands

*Correspondence to*: Marc Sanuy (marc.sanuy@upc.edu)

**Abstract.** Integrated risk assessment approaches to support coastal managers' decisions when designing plans are increasingly becoming an urgent need. To enable efficient coastal management, possible present and future scenarios must be included, disaster risk reduction (DRR) measures integrated, and multiple hazards dealt with. In this work, the Bayesian Network approach to coastal risk assessment was applied and tested at two Mediterranean sandy coasts (Tordera Delta in Spain and Lido degli Estensi-Spina in Italy). Process-oriented models are used to predict hazards at the receptor scale based on a large number of storm characteristics. Hazards are converted into impacts through vulnerability relations. A Bayesian Network integrates all results to link forcing characteristics with expected impacts through conditional probabilities. The tool has been proven successful in reproducing current coastal responses at both sites. It has also shown great utility for scenario comparisons, and is able to output significant impact change trends, despite the inherent uncertainties of the approach. This work highlights the advantages of using such a tool for present and future coastal risk assessment and planning.

**Keywords.** Disaster Risk Reduction, Source-Pathway-Receptor-Consequences, Bayesian Network, Catalunya, Emilia-Romagna, Coastal Risk Management, Erosion, Flooding.

## 1 Introduction

Increasing coastal risk due to the intensification of hazard and exposure magnitudes (IPCC, 2012; IPCC, 2013), is driving the needs of coastal managers towards more innovative approaches for coastal risk assessment and management. Highlighting these needs at the international and European levels is the impact of recent extreme events such as Hurricane Katrina in Louisiana in 2005 (Beven II et al., 2008), storm Xynthia in France in 2010 (Bertin et al., 2012; Kolen et al., 2013), Hurricane Sandy in New York in 2012 (Kunz et al., 2013; Van Verseveld et al., 2015), and the Southern North Sea storm in 2013 (Spencer et al., 2015). Similarly, in the Mediterranean, several extreme events have impacted coastal communities at the local and regional levels such as storm Klaus in 2009, as described in Bertotti et al. (2012) and cyclogenesis mechanisms



in the NW Mediterranean described in Trigo et al. (2002). In this context, the coasts of Catalunya (Spain) and Emilia-Romagna (Italy) also recently experienced coastal storm impacts that caused socio-economic losses (Jiménez et al. 2012; Perini et al., 2015; Harley et al., 2016; Trembanis et al., n.d.).

Therefore, coastal managers must properly deal with coastal risk when designing plans. This is recognised in several
initiatives such as the protocol of Integrated Coastal Zone Management (ICZM) for the Mediterranean, which includes a chapter on natural hazards and advises signed parties to implement vulnerability and risk assessments. In addition, the EU Floods directive is another example dealing specifically with floods. Therefore, the need for integrated decision support systems (DSS) based on modern models and approaches for coastal risk assessment and management is increasing. Indeed, coping with storm-induced risks in coastal areas involves testing multiple disaster risk reduction (DRR) alternatives against
multiple forcing conditions in current and future scenarios considering climate change.

The literature provides different approaches with which to implement these assessments. It is becoming increasingly important to consider multi-hazard approaches when assessing risk at all levels (i.e. from the regional to local scales). Therefore, the scientific community provides integrated and interdisciplinary approaches (e.g. Ciavola et al., 2011a; Ciavola et al., 2011b; Penning-Rowsell et al., 2014; Vojinovic et al., 2014; Oumeraci et al., 2015; Van Dongeren et al., n.d.). Up-to-
date methodologies can be used in coastal risk assessments at different scales ranging from regional approaches (up to hundreds of km) to local detailed assessments (up to 10 km). Regional methodologies aim to locate coastal sectors more prone to impacts, the so-called hotspots. Local approaches aim to achieve the highest possible level of accuracy for risk evaluation and to support decision making for previously identified hotspots. Notably, coastal risk assessments must include physical concepts to characterise physical phenomena (i.e. the source of the hazard) and socio-economic concepts to describe
the impact of the physical phenomena on human assets (i.e. the consequences). A suitable conceptual flexible framework that can capture all aspects of coastal risk assessment is the Source-Pathway-Receptor-Consequence (SPRC) model (e.g. Narayan et al. 2014, Zanuttigh et al. 2014 and Oumeraci et al., 2015).

When addressing the problem at the local scale, it is necessary to accurately predict the impact and reproduce in detail coastal hazards and responses. The analysis of physical impacts is regularly implemented in a deterministic way, with
process-based numerical models playing a central role and providing detailed information for areas prone to multiple hazards (e.g. Roelvink et al., 2009; McCall et al., 2010; Harley et al., 2011; Roelvink and Reniers, 2012). However, this must be used with multiple forcing conditions acting at the site and under different scenarios. This implies a probabilistic approach to deal with the inherent uncertainty of the problem. To obtain the best benefits from deterministic and probabilistic approaches, the integration of process-oriented tools can be combined with a probabilistic-based analysis of their results. Bayesian Networks
(BNs) have demonstrated their versatility and utility in efficiently combining multiple variables to predict system behaviour for multiple hypotheses (e.g. Plant et al. 2016). Using a BN approach, multiple multi-hazard results from process–oriented models can be integrated for joint assessment, as well as for different scenarios and alternatives (e.g. Gutierrez et al., 2011; Poelhekke et al., 2016), enabling the integration of socio-economic concepts (e.g. Van Verseveld et al., 2015).



Jäger et al. (2017) proposed the conceptual BN framework used in this work, which is based on the integration of the SPRC and was developed in the RISC-KIT EU FP7 project (Van Dongeren et al., n.d.). Plomaritis et al. (2017) applied the framework to test its potential as an early warning system (EWS) and the response of DRRs in Ria Formosa (Portugal). In this paper, the authors describe the application of the framework to select and compare strategic alternatives to reduce coastal

risk in current and projected future climate scenarios. The application in this paper was conducted at two sedimentary coasts in the Mediterranean environment, namely the Tordera Delta for the Catalan coast (Spain) and the Lido degli Estensi-Spina for the Emilia-Romagna coast (Italy).

**Figure 1: Regional and local contexts: A1) the central-northern Catalan coast; B1) Emilia-Romagna coast; A2) local hotspots of**
**Tordera Delta; B2) local hotspots of Lido degli Estensi-Spina (2b). The main locations (red dots), wave buoys (red triangles), tide gauge (red diamond), and the CSS (red squares). The domains of the large-scale and local models (dashed red lines) are highlighted for each box.**

## 2. Regional contexts and case studies

The two presented case study sites (CSS) are representative of many other coastal areas in the Mediterranean consisting of
sandy beaches where local economic activities are based on the tourist sector. These areas are characterised by urbanisation and infrastructural growth close to the shoreline (limiting natural beach accommodation processes) and economic activities directly on the beach and immediate first part of the hinterland (e.g. concessions, campsites, restaurants). The coast keeps offering its recreational function, but lacks part or all of its protective function against storms. Thus, depending on the morphological conditions of the hinterland and exposure to incoming storms, these coastal areas are prone to becoming
sectors sensitive to the impact of extreme events.

### 2.1 Tordera Delta, Catalunya (Spain)

The Catalan coast is located in the NE Spanish Mediterranean Sea (Figure 1, A1). It consists of a coastline 600 km long with about 280 km of beaches. Coastal damage has increased during the last decades along regional coasts as a result of the increasing exposure along the coastal zone and progressive narrowing of existing beaches (Jiménez et al., 2012) through
dominant erosive behaviour due to net littoral drift (Jiménez et al., 2011). Locations experiencing storm-induced problems are present along the entire coastline, and especially concentrated in areas experiencing the largest decadal-scale shoreline erosion rates. Among these areas, the Tordera delta, located about 50 km north of Barcelona, provides a good example (Figure 2).

The deltaic coast is composed of a coarse sandy coastline extending about 5 km from s'Abanell beach at the northern end
and Malgrat de Mar beach in the south (see Figure 2). This zone is highly dynamic, and is currently in retreat as a result of net longshore sediment transport directed southwest and decrease in Tordera river sediment supplies. Consequently, the beaches surrounding the river mouth, which were traditionally stable or accreting, are being significantly eroded (Jiménez et al., 2011; Sardá et al., 2013). As a result of the progressive narrowing of the beach in the area, the frequency of inundation



episodes and damage to existing infrastructure (beach promenade, campsite installations, desalination plant infrastructure, roads) has significantly increased since the beginning of the 90s (Jiménez et al., 2011; Sardá et al., 2013) (Figure 2). Subsequently, existing campsites in the most affected area have abandoned the areas closer to the shoreline, as in many cases, these areas are fully eroded or directly exposed to wave action. In other cases, owners have tried to implement local

protection measures that in many cases have enhanced existing erosion (Jiménez et al., 2017).

Coastal storms in the Catalan Sea can be defined as events during which the significant wave height (Hs) exceeds a threshold of 2 m for a minimum duration of 6 hours (Mendoza et al., 2011). Despite this, not all storms can be considered as hazardous events in terms of induced inundation and/or erosion. Mendoza et al. (2011) developed a five-category storm classification for typical conditions in the Catalan Sea based on their power content. The classification seems to well represent the

behaviour of storm events in the Mediterranean, and was successfully employed in the Northern Adriatic (Armaroli et al., 2012). Furthermore, Mendoza et al. (2011) estimated the expected order of magnitude of induced coastal hazards (erosion and inundation) for each class and beach characteristics along the Catalan coast. According to their results, storms from category III (Hs = 3.5 m, duration around 50 hours) to V (Hs = 6 m, duration longer than 100 h) are most likely to cause significant damage along the Catalan coast. One important aspect to consider is that wave-induced run-up is the largest

contribution to the total water level (TWL) at the shoreline during storm events, because the magnitude of surges along the Catalan coast is relatively low.

**Figure 2: Impacts on the Tordera Delta. Destruction of a road at Malgrat (A); overwash at campsites north of the river mouth (B); destruction of the promenade north of the river mouth (C); beach erosion, and damage to utilities and buildings at Malgrat (D and**
**E).**

## 2.2 Lido degli Estensi-Spina, Emilia-Romagna (Italy)

The Emilia-Romagna (Italy) coast is located in the northern part of the Adriatic Sea (Figure 1, B1). The coast is about 130 km long and characterized by low-lying, predominantly dissipative sandy beaches. The coastal corridor has low elevations (-2÷3m; Regione Emilia-Romagna, 2010). The area alternates between highly urbanised touristic and natural areas with dunes,

which are often threatened by flooding and erosion (Regione Emilia-Romagna, 2010). The impact of coastal erosion was emphasised by subsidence due to water and gas extraction over the last century, especially in the Ravenna area (Taramelli et al., 2015), a decrease in riverine sediment transport, because of the strong human influence on rivers and their basins (Preciso et al., 2012), and the reforestation of the Apennines (Billi and Rinaldi, 1997). Touristic activities (accommodation, restoration, sun-and-bathe) can be considered main drivers of the coastal economy. Beach concessions, which provide sun-

and-bath and restoration services, have grown exponentially in number since the second half of the last century, with negative consequences on natural areas, as in Ravenna Province (Sytnik and Stecchi, 2014). To protect the coast and its assets from the impacts of flooding and erosion, regional managers have constructed hard defences (e.g. emerged and submerged breakwaters, groins, rubble mounds; Regione Emilia-Romagna, 2010) along the entire regional coast (over 60% of the coast is protected), and regularly implement restorative nourishment plans.



During the last decades, several EU projects such as Theseus (www.theseusproject.eu) and MICORE (www.micore.eu) provided a good understanding of hydro-morphodynamics and risks to the coast. These projects and works published in the international literature such as Ciavola et al. (2007), Armaroli et al. (2009, 2012), and Perini et al. (2016) were the product of strong collaboration between scientists and regional managers (Servizio Geologico Sismico e dei Suoli, SGSS). This led to

the compilation and implementation of a storm database (Perini et al., 2011) and a regional EWS (Harley et al., 2016). The RISC-KIT project (www.risckit.eu) provided additional knowledge on this coastal area. The areas most exposed to coastal risk are well known, as can be seen in the works of Perini et al. (2016) and Armaroli and Duo (2017).

The hydrodynamics of the regional domain are well described in terms of storm waves and surges (IDROSER, 1996; Ciavola et al., 2007; Masina and Ciavola, 2011). These are as follows: the area is micro-tidal (neap tidal range: 0.3–0.4 m;

spring tidal range: 0.8–0.9 m); the surge component plays an important role (1-in-2 years storm surge: 0.61 m) and is mainly generated from the SE (Scirocco) winds (according to the orientation of the Adriatic Sea). Furthermore, the wave climate is low energy (mean Hs –0.4 m; 60% of waves are below 1 m). However, extreme events can be energetic, such as the storm of September 2004 (Hs,max=5.65m, estimated by Ciavola et al,. 2007) or the one of 5-6 February 2015 (Hs,max=4.66, measured at the Cesenatico buoy shown Figure 1, B1; Perini et al., 2015; Trembanis et al., n.d.).

The combination of high waves and storm surges, whose combined probability of occurrence in the area was assessed by Masina et al. (2015), can have strong impacts at the regional level, as demonstrated by Armaroli et al. (2009), Armaroli et al. (2012), Harley and Ciavola (2013), and Armaroli et al. (2012). Notably, based on historical data (Perini et al., 2011), Armaroli et al. (2012) provided a set of critical storm thresholds for natural and urbanised beaches to characterise potentially impacting storms. The thresholds included a combination of offshore Hs and TWL: 1) Hs ≥ 2 m and TWL (surge + tide) ≥

0.7 m for urbanised zones; 2) Hs≥3.3 and TWL (surge + tide) ≥0.8 m for natural areas with dunes.

For a more local perspective, the Lido degli Estensi-Spina coastline (Comacchio municipality, Ferrara province, Italy) area represents a highly touristic stretch of coast with concessions directly facing the sea (Figure 1, B2). The littoral drift is northward as confirmed by the width of the sandy beaches, which increases from 20 to 50 m in the southern part of Lido di Spina to 200 to 300 m in the northern part of Lido degli Estensi. Here the sediment is trapped by the groin of the mouth of a

navigation canal (Porto Canale).The beach is not protected, and regional managers implement regular nourishment in the southern part of the area (Nordstrom et al., 2015). At the back of the concessions, the villages accommodate restaurants and hotels for tourists, along with residential buildings (mainly holiday houses). South of the case study site is a natural area with dunes, which while strongly impacted by erosion, is not considered in this study. In a recent study, Bertoni et al. (2015) analysed aerial photographs of the evolution of the case study area, focusing on the stretch of coast between Porto Garibaldi

and the Reno river mouth. The area was impacted by the event in February 2015 (see Figure 3) with limited, but not negligible, consequences for several concessions (Perini et al., 2015; Trembanis et al., n.d.).



**Figure 3: Impacts of the event in February 2015 on the Lido degli Estensi-Spina case study area. Impacts of erosion and flooding on concessions at Lido di Spina south (A, B) and Lido degli Estensi (C); sandy scarp due to the erosion of the dune in the south of Lido di Spina (D); eroded Winter Dune in Porto Garibaldi (E); damages to the Porto Canale front at the Lido degli Estensi (F).**

## 3. Methodology

### 3.1 General approach: from source to consequences

The analysis framework employed in this study follows Jäger et al. (2017) and is based on the use of the SPRC model (FLOODsite, 2009; Oumeraci et al., 2015), as shown in Figure 4. This model is mostly used in coastal risk management (e.g. Narayan et al., 2014) and permits a clear representation of all risk components and their links from source to consequence. Source (S) includes the forces determining coastal response to the impact of extreme events, which are essentially a set of storms representative of the storm climates of the study sites over the entire intensity range (from moderate to extreme storms). These sources propagate to the coast and lead to different hazard pathways (P) such as erosion and inundation, the focus of the analysis. The pathways are solved through a process-oriented model to propagate storms and quantify induced processes. They are assessed for the entire coastal domain where receptors (R) are characterised according to their location on the coastal plain and typology, which define their exposure, and vulnerability to each hazard type. Finally, consequences (C) are evaluated by combining the vulnerability and exposure of each receptor with the magnitude of the hazards.

Since the main objective of the analysis is to test DRR strategies to help decision makers in future planning, the framework is applied under current conditions (hereafter current scenario, CUS) to define the baseline scenario and climate change conditions (hereafter climate change scenarios, CCS) to define a plausible future scenario. Finally, the analysis is repeated considering different DRR measures.

The general approach uses the ability of a BN to assess dependency relations between variables to reproduce the steps of the SPRC model. This conditions the application of the steps of the SPRC model, as explained in the following sections. At the same time, we use its data assimilation capabilities to integrate large amounts of data. As such, the BN can consider all dependency relations between the analysed variables, enabling the assessment of multi-hazards and the consequences on receptors for all tested incoming conditions, scenarios, and DRR alternatives in a condensed, graphic, probabilistic, and single tool.

**Figure 4: General methodology. (I) The SPRC conceptual framework is implemented through (II) a model chain, which consists of a propagation module of the source (S) and a process-oriented module for the coastal area reproducing the pathway (P). Then, (III) the consequences (C) are calculated based on the computed hazards (H) at the receptor (R) scale by using vulnerability relations (i.e. hazard-consequences functions). In the last step (IV), all variables including source boundary conditions (BC) are fitted in a BN, adding impacts after the implementation of measures (M).**

### 3.2 Source: identification and design

Since the objective of this work was to test DRR measures for risks induced by coastal storms, these are the source considered. To properly characterise storms, all relevant variables controlling the magnitude of induced hazards (erosion and





inundation) must be considered, in other words, Hs, wave period (Tp), wave direction, storm duration, and water level. In this approach, storm characteristics are defined in terms of a set of representative storms or storm scenarios that cover the typical conditions at each study site. This information is obtained from existing wave time series or bulk data of the events (recorded or modelled), usually in deep waters, propagated towards the coast to characterise storm conditions at the

nearshore of the study areas. Probable combinations that cannot be covered using existing records are represented by synthetic designed storms (e.g. Poelhekke et al., 2016; Plomaritis et al., 2017; Jäger et al., 2017). Storm scenarios are defined as a combination of the involved variables within the BN. To this end, each selected variable was categorised in discrete bins covering its probable range as a function of the climatic conditions of the study site. Then, recorded storms belonging to each bin class were selected for use in the analysis. The storm events were selected based on the information available for each

study     site     through     the     RISC-KIT     WEB-GIS     impact-oriented     database     (Ciavola     et     al.,     2017; http://risckit.cloudapp.net/risckit/#/), which provided synthetic information on the physical parameters (measured or assessed at the regional level) and socio-economic impacts of the events. In addition, time series were used to characterise all events for which this information was available.

For the Tordera Delta case, the selected variables to define storm scenarios were Hs at the peak of the storm, total storm

duration, and incoming storm direction. Tp does not significantly vary during storms in the study area (see Mendoza et al., 2011) and was not included to reduce the number of variable combinations. The coastline configuration and wave climate characteristics necessitated considering the main wave directions in terms of dominant (E) and secondary (S) directions. Finally, TWL (tide + surge) during the event was included to reproduce hypothetical future projections of MSL due to climate change. The selected discrete bins are shown in Table 1. These lead to 12 combinations (24 simulated storms)

defining the source that must be tested in the current MSL and another 12 (24 simulated storms) in the future MSL scenario. Of the 24 source storms in the current situation, 16 correspond to historic (recorded) events including the two largest, which occurred in November 2001 and December 2008. These were classified as extreme storms (category V) according to the Mendoza et al. (2011) classification. To include a full range of combinations, the remaining eight storms were completed using synthetic triangular events that correspond to combinations of Hs-duration-direction not previously recorded. Data

used to reproduce the historic events include the time series of hindcast wind fields and 2D wave spectra time series in deep waters for the NW Mediterranean (Guedes-Soares et al., 2002; Ratsimandresy et al., 2008). Wave conditions must propagate towards the coast to properly define storm events at the study site. At the Catalan coast, the storm surge contribution to the sea surface level is one magnitude lower than the wave-induced component, and the two variables are uncorrelated. All historical events with recorded associated water levels were simulated with the real storm surge, while the synthetic storms

were simulated with a storm surge of a 0.25 m constant throughout the event, as representative of the site (Mendoza and Jiménez, 2008).

**Table 1: Source characterization. Variable discretization applied at the study sites.**

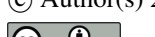



For the Lido degli Estensi-Spina case study, the source variables, identified as drivers of the impacts of flooding and erosion, were the maximum Hs and maximum TWL of the storm event. In addition, the relative sea level rise (RSLR) was considered as a Boolean variable to represent the CCS. The direction of the storms was not considered as a source characteristic variable. The ranges of the variables were classified into bins, as shown in Table 1. Seven historically based events were

selected from the RISC-KIT Database, and to cover all possible combinations, 5 additional synthetic events were considered for a total of 12 events in the CUS. Notably, for several historic events, neither reliable nor continuous time series for waves and water levels were available from local measuring stations. To ensure consistency, source events were represented based on the following methodology. Starting with the list of bulk synthetic information for each event (maximum Hs, Tp, main direction of the storm, maximum TWL or duration when available), triangular symmetric storm distributions (e.g. Carley and

Cox, 2003; Corbella and Stretch, 2012) for Hs, Tp, and surge were created for both historical and synthetic events. The peak of the waves was assumed to occur at the same time as the maximum surge (calculated as the difference between the TWL and maximum astronomical predicted tide). When bulk parameters were missing, the following 'worst case' assumptions were introduced: Tp at peak of 10 s, wave direction of 90º, and duration based on similarity with other storms.

### 3.3 Pathways: modelling multi-hazard impacts

To reproduce the pathway from source (storm) to impact (hazards), a model chain was designed and adapted for each site (Figure 4, II). The chain must be able to reproduce all hazards to be assessed (i.e. erosion and inundation). To do this, a detailed 2D process-oriented model simulating inundation and erosion in an integrated way was employed (the obtained inundation includes the morphodynamic feedback associated with coastal erosion during the storm). The XBeach model was used for this purpose in both study cases (see Roelvink et al., 2009 for model details).

The model chain for the Tordera Delta study case consists of two blocks, one 'external' and one 'internal'. The external module comprises three models (HAMSOM, HIRLAM, and WAM models) that supply the forcing conditions (time series of water levels, wind fields, and waves) and are run by Puertos del Estado (Spanish Ministry of Public Works). The output of these models is taken directly as an input for the internal module, which comprises the SWAN (Booij et al., 1996) and XBeach (Roelvink et al., 2009) models. SWAN was used to propagate wave conditions provided by the external models

(regional scale) to the offshore boundary of the XBeach model (20 m depth), while XBeach was employed to assess the extension and magnitude of inundation and erosion hazards at the study site (local scale). The model chain was validated through the St Esteve event in 2008 (Sanuy and Jiménez, n.d.).

The model chain for the Lido degli Estensi-Spina case study only included the XBeach model. This simple approach was possible based on the assumption that the information derived from the RISC-KIT Database can be considered representative

of the storm in the regional domain, as collected from different sources (e.g. offshore buoys, harbours' tide gauges, newspapers, etc.) along the Emilia-Romagna coast (Perini et al., 2011; Ciavola et al., 2017). The model was qualitatively validated with the February 2015 event (Perini et al., 2015; Trembanis et al., n.d.).





### 3.4 Receptors and consequences

The methodology applied in this work individually identified receptors located at the study sites (Figure 4, III) (Jäger et al., 2017). First, receptors with homogeneous vulnerability characteristics were defined and separately considered. Then, for each group of receptors, polygons were drawn using a GIS-based tool to account for their exact location and size. Finally,
the polygons were intersected with the cells of the 2D detailed model grid (XBeach) to assign to each receptor the nodes of the model that will affect them.

For the inundation hazard, the value of the maximum water depth inside each receptor was used as the impact variable. Then, by using flood-damage curves for the corresponding receptor typology, inundation water depth was translated to relative damage. Thus, flood-damage curves are the vulnerability relations used to quantitatively assess inundation risk. This
was then translated into four levels of impact—none, low, medium, and high—which are case and receptor dependent (see the following sections).

The magnitude of the risk associated with erosion depends on the combination of vertical erosion and distance of erosion to the receptors. This was implemented by building multiple buffers (increasing in distance) around each receptor and intersecting them with the information of maximum vertical erosion output from XBeach. The definition of risk categories
related to erosion thresholds and distances is site dependent.

### 3.4.1 Exposure and vulnerability in the Tordera Delta case study

The distribution of receptors for the Tordera Delta case study was derived from cartographic information from the Catalan Cartographic Institute (ICC) and completed manually through an orthophoto analysis. The study site was divided into eight areas, of which four are located at the south of the river mouth, corresponding to the Malgrat de Mar municipality, and the
other four to the north, corresponding to the Blanes municipality. These two sets of four areas were selected to enable an analysis of the impact at different bands regarding the limit of the public domain (which separates the public beach from the hinterland). The first band corresponds to the first 20 m of hinterland. The second band is 30 m wide and located just after the first one 20 to 50 m from the boundary of the public domain. The third covers the range from 50 to 75 m, while the fourth band covers all the hinterland omitted between the end of the third band and inland simulation domain boundary. This
enables an assessment of the distribution of the impacts of the different scenarios in terms of distance to the coastline. Thus, the effectiveness of removing receptors from each of the bands considered could be assessed, which corresponds to different setbacks as DRR measures. Three groups of receptors were considered to have homogeneous vulnerabilities, namely houses (concrete buildings), campsite elements (soft buildings and caravans), and infrastructure (promenade and road at the back of the beach). Table 2 shows the distribution of campsite elements and houses in the different areas. The infrastructural
receptors (promenade at the north and road at the south) are only located in the first 20 m band (Areas 1 and 5).

**Table 2: Distribution of receptors at the Tordera Delta study site.**





The consequences of flooding were assessed through flood damage curves used to characterise the relative damage based only on water depth. Data (see details in Table 3) was obtained from the Agència Catalana de L'Aigua (2014), which derived it from FEMA (2001).

5 The relative damage values to buildings and campsite elements were converted into the level of risk as follows: (i) No impact for 0% relative damage to buildings and campsite elements, (ii) Low impact for damages below 26% to buildings and 50% for campsite elements, (iii) Medium impact when damages to buildings range from 26 to 45% and damages to campsite elements range between 50 to 70%, and (iv) High impact for relative damages higher than those formerly exposed for both receptors.

**Table 3: Vulnerability relations for houses and campsite elements at the Tordera Delta study site with and without DRR measures (FRM).**

The buffers defined to assess the erosion hazard at the Tordera Delta are as follows: (i) The 20-m distance corresponds to the 15 average beach retreat at the site for a storm with a return period of 38 years. This was used as a threshold ranging from 'none' to 'low' risk of direct impact due to erosion. The return period is commonly used for infrastructural receptors similar to those in the Tordera Delta (low economic importance for a lifetime of about 25 years). (ii) The 12-m buffer (average retreat for the 10-year return period) was used as the threshold from low to 'medium' impact. For a medium impact, receptors are in the post-monitoring situation and begin to be exposed to the direct impact of relatively frequent storms. (iii) 20 Finally, the 3-m buffer was used as the threshold for the 'high' impact risk, meaning that the receptor is directly affected by erosion at the toe or impacted by a direct wave in the analysed scenario. A buffer was considered to have been affected when a vertical erosion threshold of 50 cm was imposed.

### 3.4.2 Exposure and vulnerability in the Lido degli Estensi-Spina case study

The analysed receptors belong to the central area of the model domain at approximately 600 m from the lateral boundaries 25 (Figure 1, B2). Two main types of receptors were selected: (i) the residential and commercial buildings mainly present in the towns of L. Estensi and L. Spina, and (ii) beach concessions on the beach directly facing the sea. In this study, only receptors belonging to the seafront of Lido degli Estensi and Lido di Spina were considered, as they are mainly impacted by sea storms. Receptors were extracted from a recent Regional Topographic Map (Carta Topografica Regionale, scala 1:25000, anno 2013), and the polygons were drawn in ArcGIS. Table 4 summarises the identified receptors. Following this, the grid 30 cells affecting each receptor were defined.

**Table 4: Distribution of receptors at Lido degli Estensi and Lido di Spina.**



The vulnerability relation for inundation hazards was defined considering a flood-damage curve from a recent study on Italian territory by Scorzini and Frank (2015). This work was based on a micro and macro-scale study of the impacts of the 2010 river flood in Veneto (Italy) on residential houses. In the current work, it was adapted and applied to the receptors of the area (see details in column A of Table 5), and relates the flood relative damage factor (FRDF; values: 0–1) to flood depth. In particular, the worst case curve was used, which represents flood-related damages to single-family detached buildings with a basement. Although this curve is for residential buildings, it is assumed the same for commercial buildings and beach concessions, as no additional and specific information was available. The curve was modified considering the DRR implementation described in Section 3.5.2. The level of flood risk was defined as follows: none, when the FRDF is null, low, when the FRDF is higher than zero but lower than 0.1, medium, for an FRDF between 0.1 and 0.2, and high, for an FRDF higher than 0.2.

**Table 5: Vulnerability relation for flooding adopted for the receptors at Lido degli Estensi-Spina without (A) and with DRR measures (B).**

The vulnerability relation for erosion was defined for concessions only. The impacts due to the erosion hazard were defined based on a two-buffer approach for each receptor. The buffers were defined as follows: (i) the first buffer was the footprint of the receptor, and (ii) the second included a corridor of 10 m around the receptor.

The erosion risk categories for each receptor were set as follows: (i) Safe: no erosion in any buffer, (ii) Potential Damage: when erosion is present (>0.05 m; negligible otherwise) in the 10-m buffer and/or is present with values less than 0.5 m in the footprint buffer, and (iii) Damage: when the erosion limit of 0.5 m is exceeded for the footprint buffer. Notably, the threshold of 0.5 m was set considering the uncertainty of the model grid topography (±0.15 m) and assuming that the foundations of the concessions are a minimum of 0.2 m thick.

## 3.5 Testing scenarios and DRR alternatives

To compute the analysis under CCS and under the implementation of DRRs, it was necessary to identify the variables and settings affected by each scenario, either a future projection or implementation of a risk reduction measure. Therefore, an appropriate approach was selected to consider these modifications in the SPRC chain.

The CCS mainly affect the hazard and therefore, are applied in the modelling chain. The DRRs can affect both hazard and vulnerability/exposure variables. In the following, the implementation of the CCS and DRRs is described for each case study, emphasising the affected variables and steps of the methodology. All assessed DRRs were considered fully implemented and completely effective (DRR uptake and effectiveness: 100%) in all cases.



### 3.5.1 Climate change scenarios in the case studies

Future projections of MSL were based on the AR5 RCP8.5 (Church et al., 2013). Other factors such as changes in storminess, winds, or waves were not expected to change significantly in the NW Mediterranean (Lionello et al., 2008; Conte and Lionello, 2013), and are characterised by high uncertainty in the Northern Adriatic (IPCC, 2013). Data to include the sea level rise (SLR) in the assessment of future scenarios was provided by the EC Joint Research Centre database (for further detail, see Vousdoukas et al., 2016). For the Tordera Delta study case, the time horizon of 2100 was chosen, while the 2050 projection was used for Lido degli Estensi-Spina, because the SLR projections in the Adriatic are more uncertain than in the NW Mediterranean. Therefore, the 2100 horizon could yield highly unreliable results.

At the Tordera Delta, the RCP8.5 estimates an increase of 0.73 m by 2100. Therefore, all 24 simulations described in Section 3.2 were repeated with the projected future MSL. Moreover the potential beach accommodation to SLR was modelled following Bosom (2014) and Jiménez et al. (2017). This was accomplished assuming an equilibrium coastal profile response following the Bruun rule (Bruun, 1962), resulting in landward and upward displacement of the beach profile. The estimated shoreline retreat due to the SLR in the area is 22 m. Thus, morphological coastal adaptation to SLR is included in the assessment. Finally, Casas-Prat and Sierra (2012) predicted a directional change in mean sea conditions from the current dominant (E) to the secondary direction (S). This effect was qualitatively explored by assessing eastern incoming storms in the CUS and imposing an equal likelihood of eastern and southern incoming storms in the CCS. Therefore, three different CCS were explored: (i) CUS + SLR with the corresponding estimated beach accommodation (CCS1), (ii) CUS + effect of direction switch in incoming storms (CCS2), and (iii) assessing the contribution of both components if occurring at the same time, i.e. SLR + switch in storm incoming direction (CCS3).

In Lido degli Estensi-Spina, the combined contribution of the predicted SLR with the subsidence component (not negligible in the area, e.g. Taramelli et al., 2015) was implemented. The resulting value of RSLR by 2050 used in the analysis is 0.30 m. The forcing events' water level time-series were modified, including the predicted RSLR by 2050 in the CCS. The morphological accommodation to the SLR was not implemented in the numerical analysis; however, its effect is discussed in Section 5.2. In total, 24 additional simulations were run for the CCS.

### 3.5.2 DRR alternatives in the case studies

Three DRR measures were tested for the Tordera Delta zone (see Figure 5): (i) Receptors Setback (RSB), (ii) Flood Resilience (set of) Measures (FRM), and (iii) Nourishment + Dune (N+D).

The RSB measure affects the exposure of the receptors. It entails removing all receptors inside a defined band measured from the public domain coastal limit (the limit between the back of the beach and hinterland). Three scenarios of the setback were simulated: 20 m, 50 m, and 75 m.

The FRM affects the vulnerability of receptors so that for a given water depth, the expected impact on campsites and houses during an inundation event decreases from the current situation when the DRR measure is implemented. It is assumed that



resilience measures such as raised electricity outlets and utilities, adapted flooring, resilient plaster, and waterproof doors and windows were installed in all houses and campsite elements.

Finally, the N+D affects the inundation/erosion hazard. It includes beach nourishment at the south of the river mouth to increase the beach width by 50 m over 1 km at the south of the river mouth, where the highest erosion occurs. In addition, the level at the top of the beach was increased on both sides of the river mouth, with non-erodible sandbags at the northern side, where the campsites are closer to the coastline, and a sandy dune at the southern side. At both sides, the final height of the protective measure was +4.8 m in terms of the MSL. Since this measure affects the pathway, 24 extra simulations, this measure was implemented in the XBeach grid, were needed, and another 24 to combine the implemented measure with the CCS.

**Figure 5: DRR measures at Tordera Delta. Coastal setbacks (20, 50, and 75 m) and Infrastructural Defence (beach nourishment at Malgrat beach + artificial dune at S'Abanell and Malgrat beaches).**

The selected DRR measures tested for the Lido degli Estensi-Spina case study were: (i) a Winter Dune (WD) system, affecting both flooding and erosion impacts, and therefore the hazards modelling process; and (ii) a set of FRM, influencing the flood vulnerability relations of receptors.

The WD (see Figure 6) is a common DRR practice along the Emilia-Romagna coast, especially in the Ravenna province (Harley and Ciavola, 2013), and regularly implemented by local concessionaires without a scientifically based design criterion. It consists of a set of embankments built on the beach in front of concessions through beach scraping or sand replenishment (less frequent option). This DRR measure was implemented in the process-oriented module (XBeach). The WD was designed as a continuous dune that protects more than one concession, introducing breaks in the continuity of the feature where natural/human obstacles or passages were located. The top of the WD was fixed at 3 m above the MSL and the width (at the top) at 10 m. The WD was integrated in the model modifying the bed levels through the Dune Maker 2.0 tool (Harley, 2014). Both the CUS and CCS were tested with the DRR WD adding 48 additional simulations.

The FRM decreases the receptor's physical vulnerability to floods. It was assumed that the effective application of these measures would decrease the damages (FRDF >0.1) for water levels lower than a certain threshold, assumed here as 0.7 m (e.g. all electrics have to be placed above the threshold). This assumption was integrated in the analysis by modifying the selected depth-damage curve, as defined in column B of Table 5, and included in the BN. Considering the adopted definition of flood risk levels (see Section 3.4.2), the FRM results in a complete obliteration of receptors for the medium flood risk, therefore increasing the receptors at the low level and not affecting receptors at high risk.

**Figure 6: Artificial winter dunes in Emilia-Romagna: A) Winter dune in Porto Garibaldi (Comacchio, Italy); B) Building of a winter dune by beach scraping at Lido di Dante (Ravenna, Italy) (Harley, 2014); C) Representative model profiles at Lido di Spina north (original: black solid line; with winter dune DRR: red dashed line).**



### 3.6 Bayesian network DSS

BNs use probability theory to describe the relationships between many variables, and can evaluate how the evidence of some variables influence other unobserved variables. For example, evidence could be a forecast of the source variables characterising an impending storm. On the other hand, local hazards and damages in the coastal area have not yet been observed, but can be predicted with the BN. The model can also be updated with artificial evidence to explore extreme event scenarios or investigate the potential of disaster risk reduction plans.

A BN is based on a graph (Figure 7). It consists of nodes connected by arcs that represent random variables and the potential influences between them. The direction of the arcs is crucial for the probabilistic reasoning algorithm of the BN, but does not necessarily indicate causality. For any two variables connected by an arc, the influencing one is called a parent, while the one influenced is referred to as the child. Thus, in Figure 7, X1, X2, and $X_3$ are the parents of $X_4$. A simple way to parameterise a BN is to discretise continuous variables after defining their data range, and to specify conditional probability tables for each node. The authors adopted this approach. The conditional probability tables indicated how much a variable could be influenced by others. Mathematically, the graph structure and conditional probability tables define the joint distribution of all variables in the network, $X_1, ..., X_n$, based on the factorisation of conditional probability distributions (Eq. 1):

$$p(X_1, ..., X_n) = \prod_{i=1}^{n} p(X_i | pa(X_i)),$$ (1)

where pa(Xi) are the parents of node Xi (Pearl, 1988; Jensen, 1996). Once the joint distribution has been defined, the effects of any evidence can be propagated with efficient algorithms throughout the network (Lauritzen and Spiegelhalter, 1988).

**Figure 7: BN graph with four nodes.**

In the RISC-KIT project, a generic structure for a BN that can support decision-making in coastal risk management was proposed. This structure is based on the SPRC and has five components: source boundary condition (BC), hazard (H), receptor (R), impact/consequence (C), and DRR measure (M). Typically, each component includes several variables. Panel (IV) in Figure 4 shows their influence on each other. In general, all boundary conditions influence all hazards, as indicated by the solid arc in Figure 4. Differently, each type of receptor (e.g. people, buildings, infrastructure, and ecosystems) has a sub-module in the BN consisting of an R node (representing the locations of receptors on the site), H nodes (representing the hazards given the locations of the receptors), and C nodes (representing the consequences given (some of) the hazards for the receptors). The dashed arcs in Figure 4 represent the fact that the sub-modules are not directly interconnected. Nevertheless, dependencies arise from the common parents, which are boundary conditions and possibly DRR measures.

Alongside the generic structure, it also provided a c++ programme that automatically creates the BN (https://github.com/openearth/coastal-dss). As input, the programme requires variable definitions and land use data,





vulnerability relationships, and a 2D gridded simulation output of numerical physical process-based models of hindcast or synthetic extreme event scenarios. Essentially, the programme extracts the values of hazard variables from the simulation output at the locations of every individual receptor so that we could obtain hazard distributions for each receptor type. Because each simulation contains the coastal response to one storm scenario under a set of DRR measures, the distributions

are conditional and can be stored directly as entries of the conditional probability tables associated with each hazard node. Being parents of the hazard nodes, boundary conditions and DRR measures define the dimensions of the conditional probability tables. By simulating those storm scenarios that correspond to all possible value combinations, the tables are completely filled. In the final step, the conditional hazard distributions were transformed to conditional impact distributions with vulnerability steps.

**3.6.1 BN implementation at the case study sites**

The schemes of the BNs implemented for the Tordera Delta and Lido degli Estensi-Spina case study sites are shown in Figure 8 and Figure 9 respectively. The nodes (circles) define the variables of the network, while arcs (arrows) show the relations between the variables. The BC is the blue nodes, and the location and distributions (R) of the receptors are the grey nodes. These nodes affect those in dark orange, which refer to the receptors' hazards (H). The hazard was then transformed

through the vulnerability relations into consequences (C), which are represented by the light orange circles. The measures' nodes (M) are indicated in green and can affect the H, C, or R nodes.

**Figure 8: Bayesian Network scheme for the Tordera Delta site.**

**Figure 9: Bayesian Network scheme for the Lido degli Estensi-Spina site.**

**4 Results**

In this section, the results of scenario testing are provided for each case study through an integrated comparison of computed risk levels in terms of percentages of receptors at each level of risk for flooding and erosion for each type of receptor and relevant location in the CUS and CCS. The DRR impacts are shown by comparing the risk levels in the CUS and CCS with

those computed in each scenario with the implemented DRR.

**4.1 Tordera Delta**

The results assessment was performed separately for both sides of the river at s'Abanell beach at the north and Malgrat beach at the south. The inundation impact assessment considered all receptors at the study site whereas the erosion analysis focussed only on the first 20-m band of hinterland because the only receptors exposed to an erosion hazard are located in that

area.





The results of the flooding impacts on campsite elements and houses (Figure 10 to Figure 13) indicate that in the CUS (E incoming storms with current MSL), campsite elements at both sides of the river mouth are expected to suffer the same magnitude of damages: 80–83% of elements will be safe, while only 2–3% of the elements are under high-impact risk (Figure 10). The situation differs slightly when assessing houses (Figure 12), since more damages are expected to occur

south of the river mouth (20% of elements are at low risk and 2% at medium risk), rather than in the northern domain (2% at low risk and 1% at medium risk).

When assessing the CCS, results demonstrate a different behaviour at each side of the river mouth. In S'Abanell, the SLR significantly increases the impacts of flooding (CCS1, Figure 10), whereas the directional shift of storm direction (equal frequency of E and S incoming storms) does not increase any of the receptors at risk (CCS2 and CCS3, Figure 11). In the

CCS1, the impact on campsite elements increases from 17% to 37% of affected receptors. Campsite elements expected to suffer high impacts increase from 2% to 14%. However, expected impacts under CCS2 are similar and even lower than those observed under the CUS, and the results obtained for CCS3 are comparable to CCS1 for the northern domain.

On the other hand, south of the river mouth, the response to CCS is equally sensitive to changes in storm incoming direction than to SLR. In fact, 50% of houses and 56% of campsite elements are affected by some level of impact under CCS1 (Figure

10 and Figure 11), while 38% of houses and 40% of campsite elements are affected under CCS2 (Figure 11 and Figure 13). Therefore, when CCS3 conditions are tested south of the river mouth, the outcome obtained from the BN shows that 63% of houses (34% at medium risk) and 69% of campsite elements (41% at high risk) are affected.

**Figure 10: Distribution of campsite elements at every level of flooding risk. Top-left: current scenario at S'Abanell; Top-right:**
**climate change scenario 1 (SLR) at S'Abanell; Bottom-left: current scenario at Malgrat; Bottom-right: climate change scenario 1 (SLR) at Malgrat. Each bar in a panel represents a DRR configuration ('None': no DRR implemented; 'N+D': Nourishment and Dune; 'FRM': Flood Resilience Measures; '20SB, 50SB, and 75SB': 20, 50, and 75 m setbacks, respectively).**

**Figure 11: Distribution of campsite elements at every level of flooding risk. Top-left: climate change scenario 2 (50-50% east-south**
**storms) at S'Abanell; Top-right: climate change scenario 3 (50-50% of east-south storms + SLR) at S'Abanell; Bottom-left: climate change scenario 2 (50-50% east-south storms) at Malgrat; Bottom-right: climate change scenario 3 (50-50% of east-south storms + SLR) at Malgrat. Each bar in a panel represents a DRR configuration ('None': no DRR implemented; 'N+D': Nourishment and Dune; 'FRM': Flood Resilience Measures; '20SB, 50SB, and 75SB': 20, 50, and 75 m setbacks, respectively).**

Comparing the effectiveness of the DRR highlights N+D as the most effective measure against flooding for the CUS and all tested CCS. As expected, the effectiveness is higher in Malgrat than in S'Abanell, as beach nourishment is located only south of the river mouth and the dune is present on both sides. It was observed that all significant impacts (medium and high) to receptors under the CUS were removed for both sides of the river. Moreover, at the Malgrat domain, the number of affected receptors was reduced by 19%–22% for the CUS, CCS1, and CCS2 scenarios, and 40–46% under CCS3.

The implementation of the FRM was effective in terms of preventing high impacts on any receptor, but did not significantly reduce the total number of receptors affected by some level of risk. The magnitude of reduction of receptors at risk was ~9%.



It should be mentioned that this is a theoretical measure, as we assumed that the FRM are properly designed and 100% effective for site conditions.

**Figure 12: Distribution of houses at every level of flooding risk. Top-left: current scenario at S'Abanell; Top-right: climate change scenario 1 (SLR) at S'Abanell; Bottom-left: current scenario at Malgrat; Bottom-right: climate change scenario 1 (SLR) at Malgrat. Each bar in a panel represents a DRR configuration ('None': no DRR implemented; 'N+D': Nourishment and Dune; 'FRM': Flood Resilience Measures; '20SB, 50SB, and 75SB': 20, 50, and 75 m setbacks, respectively).**

**Figure 13: Distribution of houses at every level of flooding risk. Top-left: climate change scenario 2 (50-50% east-south storms) at S'Abanell; Top-right: climate change scenario 3 (50-50% of east-south storms + SLR) at S'Abanell; Bottom-left: climate change scenario 2 (50-50% east-south storms) at Malgrat; Bottom-right: climate change scenario 3 (50-50% of east-south storms + SLR) at Malgrat. Each bar in a panel represents a DRR configuration ('None': no DRR implemented; 'N+D': Nourishment and Dune; 'FRM': Flood Resilience Measures; '20SB, 50SB, and 75SB': 20, 50, and 75 m setbacks, respectively).**

Finally, three RSB were tested: 20 m (20SB), 50 m (50SB), and 75 m (75SB). The results indicate that only the 75-m setback demonstrated a risk reduction magnitude comparable to infrastructural defence; however, in most cases, the efficiency of the N+D was higher than the managed retreat. Only in S'Abanell, with higher topography and where the measure consists of only dune without nourishment, a greater risk reduction was achieved through the 75SB.

Results for the erosion impact risk assessment showed similar results for the three analysed receptor categories and no significant differences between CUS-CC1 and CC2-CC3. For simplicity, results related to Campsites (Figure 14) and Infrastructure (Figure 15), for the CUS and CC1 scenarios are provided in the following.

**Figure 14: Distribution of campsite elements at every level erosion risk. Top-left: current scenario at S'Abanell; Top-right: climate change scenario 1 (SLR) at S'Abanell; Bottom-left: current scenario at Malgrat; Bottom-right: climate change scenario 1 (SLR) at Malgrat. Each bar in a panel represents a DRR configuration ('None': no DRR implemented; 'N+D': Nourishment and Dune; 'FRM': Flood Resilience Measures; '20SB, 50SB, and 75SB': 20, 50, and 75 m setbacks, respectively).**

Under the CUS, 23% of campsite receptors (Figure 14) in s'Abanell and 8% in Malgrat are at low risk, whereas only 1–2% demonstrate a medium risk in both areas. The CCS indicated that the level of erosion risk increases much more when the SLR increases than for the directional switch of incoming storms on both sides of the river mouth. In the CCS1 scenario, receptors located in s'Abanell at medium risk increase to 30%, while 5% are at high risk. In Malgrat, the same scenario results in 20% of campsite elements being at medium risk and 14% at high risk. On the other hand, the CCS2 scenario does not imply a significant difference for the CUS and similarly, the impacts in CCS3 are comparable and lower than those obtained for CCS1.

Focusing on the infrastructural receptors (Figure 15), the promenade at the north of the river mouth is currently at significant risk (70% at medium risk and 13% at high risk), whereas the road in Malgrat is potentially safe. In the CCS1 scenario, the assessment highlights that because of the increase of MSL and corresponding morphological accommodation, the percentage





of promenade under high risk and therefore direct erosion at the toe increases up to 33%, with some impact on the road in Malgrat.

**Figure 15: Distribution of Infrastructures at every level erosion risk. Top-left: current scenario at S'Abanell; Top-right: climate change scenario 1 (SLR) at S'Abanell; Bottom-left: current scenario at Malgrat; Bottom-right: climate change scenario 1 (SLR) at Malgrat. Each bar in a panel represents a DRR configuration ('"None"': no DRR implemented; '"N+D"': Nourishment and Dune; '"FRM"': Flood Resilience Measures; "20SB, 50SB, and 75SB"': 20 , 50, and 75 m setbacks, respectively).**

The assessment of the efficiency of the DRR regarding erosion indicates that the N+D does not have any significant impact on reducing risk. In fact, in some scenarios, the number of affected receptors at low risk seems to increase, because of the indirect effect of alongshore change in erosion/accretion patterns caused by the measure. In addition, the beach nourishment is regularly washed out in severe storm conditions. In the case of the road in Malgrat, the nourishment is placed in a position with higher local erosion rates; thus, the measure prevents the impact in CCS1. On the other hand, RSB is 100% effective in dealing with the impact of erosion, and the 20SB is enough to cope with risk under the present situation and for all future projected conditions.

## 4.2 Lido degli Estensi-Spina

The overall results for the risk of flooding for residential and commercial buildings located in the towns of Lido degli Estensi and Lido di Spina are provided in Figure 15. The overall results for concessions are shown in Figure 16 and Figure 17, for flooding and erosion risks respectively.

The CUS for residential and commercial buildings (Figure 15) evidenced that most receptors in Lido degli Estensi-Spina are safe from flooding impacts, with the exception of the 2% at low risk in Lido degli Estensi. In the CCS, the receptors at low risk increased in Lido degli Estensi to 10%, while 1% were at medium risk. The increase in receptors at low risk is more limited in Lido di Spina (–5%). The WD demonstrated a positive effect on receptors' level of risk. In particular, under the CCS, it decreased the receptors at low risk by almost 10% of the total. The effect of FRM is limited to the CCS at Lido degli Estensi, where the receptors at medium risk were reclassified at a low level.

**Figure 16: Distribution of residential and commercial buildings for every level of flooding risk. Top left: current scenario at Lido degli Estensi; Top right: climate change scenario at Lido degli Estensi; Bottom left: current scenario at Lido di Spina; Bottom right: climate change scenario at Lido di Spina. Each bar in a panel represents a DRR configuration ('None': no DRR implemented; 'WD': Winter Dune; 'FRM': Flood Resilience Measures).**

Focusing on the flooding risk for concessions (Figure 16), the CUS evidenced noticeable impacts. At Lido degli Estensi, almost 30% of receptors were categorised at low risk and 15% at medium risk. In comparison, at Lido di Spina, the receptors at risk increased in number and intensity. The results showed that 27% were at low risk, 25% at medium risk, and 6% at high risk of flood. The CCS exacerbated the impacts for both locations. At Lido degli Estensi, the concessions at low risk





increased to 50%, those at medium risk remained stable, but 8% of receptors were at high flooding risk. Similarly, at Lido di Spina, the percentages of concessions categorised at risk increased to 34% at low risk, 33% at medium risk, and 20% at high risk.

The WD system had a positive impact in all cases for both the CUS and CCS. At Lido degli Estensi, the concessions at risk

decreased from 44% (29% at low and 15% at medium risk) to 10% (only low risk) of the total for the CUS. The same scenario for Lido di Spina demonstrated limited impacts (13% at low and 3% at medium risk) for concessions when the WD was implemented, while the total receptors at risk without DRR was 58%. The impacts on the CCS also decreased with the WD compared to the scenario without DRR. For Lido degli Estensi, where previously more than 60% of concessions were at risk, 8% of the receptors were at low risk and 14% at medium risk. At Lido di Spina, the positive effects of the WD system

increased the percentage of safe receptors from 13% to 59%, thus decreasing the concessions at risk (26% at low, 6% at medium, and 9% at high flooding risk).

The FRM had positive effects on impacts by moving all receptors at medium risk to the low risk category. In particular, under the CUS, the concessions at low risk increased from 29% to 44% and from 27% to 52% at Lido degli Estensi-Spina respectively. For the CCS, the same results increased from 50% to 65% at Lido degli Estensi, and from 34% to 67% at Lido

di Spina.

**Figure 17: Distribution of concessions for every level of flooding risk. Top left: current scenario at Lido degli Estensi; Top right: climate change scenario at Lido degli Estensi; Bottom left: current scenario at Lido di Spina; Bottom right: climate change scenario at Lido di Spina. Each bar in a panel represents a DRR configuration ('None': no DRR implemented; 'WD': Winter**
**Dune; 'FRM': Flood Resilience Measures).**

The risk assessment results related to coastal erosion (Figure 17) showed the potential level of damage of risk under the CUS: 8% and 14% of concessions at Lido degli Estensi-Spina respectively. In the CCS, the previous results increased to 11% and 30% respectively. Notably, 1% of the concessions in Lido degli Estensi were indicated as being possibly damaged.

The WD system demonstrated positive effects on the potentially damaged concessions at Lido di Spina under the CUS by decreasing the receptors at potential risk from 14% to 5%. In contrast, at Lido degli Estensi, the potentially damaged receptors increased from 8% to 13% when implementing the WD DRR. This negative effect also occurred in the CCS. At Lido degli Estensi, the receptors at potential risk increased to 17%, while damaged receptors remained stable. At Lido di Spina, the WD had a contradictory effect. It decreased potentially damaged receptors to 14% and increased damaged

concessions to 2%.

**Figure 18: Distribution of concessions for every level of erosion risk. Top left: current scenario at Lido degli Estensi; Top right: climate change scenario at Lido degli Estensi; Bottom left: current scenario at Lido di Spina; Bottom right: climate change scenario at Lido di Spina. Each bar in a panel represents a DRR configuration ('None': no DRR implemented; 'WD': Winter**
**Dune; 'FRM': Flood Resilience Measures).**





A further step in the analysis of risk scenarios was undertaken using the BN to show the distribution of the boundary conditions that generate flood damage to concessions at Lido degli Estensi-Spina in the configuration without and with the WD DRR. The BN enables assessing the distribution of boundary conditions related to an impact scenario where all receptors suffer consequences uniformly for all risk levels (i.e. all receptors are affected by flooding at least at a low level of

risk). The results of this analysis are shown in Figure 18 for the CUS, with (Figure 18, green bars) and without the WD DRR (Figure 18, red bars). The graph on the left shows the distribution of the TWL and the Hs on the right.

Notably, under the CUS without DRR, the Hs is distributed more uniformly (values ranging from 15% to 31%) compared to the TWL, which demonstrated a strong tendency to increase (values ranging from 10% to 58%). This indicates that compared to wave conditions, the water level is the main driver for flood impacts.

The results for the WD DRR scenario showed that the most probable conditions leading to flood damages to concessions are TWL>1.45 m (93%) and Hs>4 m (4<Hs<5 m: 47%; 5<Hs<6 m: 43%). These results indicated that the WD DRR in the CUS can minimise the consequences of coastal storms with TWL<1.45 m and Hs<4 m.

**Figure 19: Distribution of boundary conditions (TWL on the left and Hs on the right) that generate flood damages in the current**
**scenario for Lido degli Estensi-Spina. The configuration without DRR (green bars) and for the implementation of the WD DRR (red bars) were compared.**

The same analysis was performed for the CCS, as shown in Figure 19. In this case, the scenario without DRR demonstrated a less dominant influence of TWL (ranging from 24% to 40%) on flood consequences to concessions, even if still stronger than the Hs, and an almost uniform distribution (all bins around 25%). As expected, the RSLR (+0.3 m; RCP8.5 by 2050)

increased the risk of lower intensity storms. Thus, in general, under the CCS, all storm combinations generated flood consequences to concessions.

The results for the WD in the CCS showed that the most probable condition leading to flood damages to concessions is when TWL>1.45 m (75%) in combination with Hs>4 m (4<Hs<5 m: 35%; 5<Hs<6 m: 33%). Thus, under the CCS, the influence of the WD system is less effective than the CUS. Indeed, lower intensity storms can still lead to flood damages to

concessions (TWL<1.45 m: 25%; Hs<4 m: 32%).

**Figure 20: Distribution of boundary conditions (TWL on the left and Hs on the right) that generate flood damages in the climate change scenario for Lido degli Estensi-Spina. The configuration without DRR (green bars) and under the implementation of the**
**WD DRR (red bars) were compared.**

## 5 Discussion

The aim of the present work is appropriate for the prevention phase of the disaster management cycle. The tool was applied as a DSS for coastal management, and therefore used for comparison purposes to support the assessment of DRR strategic



alternatives. This comparison was performed for a large set of simulations, covering many (current and future) conditions and multiple hazards.

Notably, the analysis has some inherent uncertainties associated with the implementation of the steps of the SPRC model. The authors highlight that the presented framework (Jäger et al., 2017) has demonstrated flexibility. As such, the authors are currently investigating its application to other types of risks (i.e. rock falls, landslides, etc.). In addition to the use as DSS, the tool has the potential to be applied as an EWS (Plomaritis et al., 2017) once it is properly validated. However, for that purpose, more focus on the validation of the model chain is needed.

In the following sections, uncertainties and limitations of the application of the tool at both study sites are presented and discussed alongside the obtained results.

## 5.1 Tordera Delta

The methodology starts with the Source characterisation and variable range definitions. In this work, a first test to check the method was presented, and a balance between computational expense and accuracy pursued. However, the way the bins were selected affects the accuracy of the output of the BN. Some input parameters have a wide range (–30-hour steps in duration and only 2 main wave directions) and more simulations are desirable for a better representation of the variability of the results inside each bin. Alternatively, a higher bin resolution could be tested at the expense of a significant increase in computational efforts. The tested combinations provide a representative picture of the coastal response of the site and effectively describe the input-output relations meant to be captured in the BN. Larger amounts of forcing time series would better represent the schematisation for all combinations of storms, reducing those represented through synthetic events.

Later, a model chain to obtain the hazards' Pathway was set and implemented. The validation of both models in the model chain (SWAN and XBeach) was performed using the St Esteve 2008 event (Sanuy and Jiménez, 2017). Better validations of the model chain could be achieved using more storms to cover a representative range of characteristics for the site.

Regarding receptors and consequences, the locations of receptors have little associated uncertainty. Houses and promenades were derived from accurate land use data available for the site, and the campsite elements were manually located and delimited from available GIS-based tools and raster imagery. Some uncertainty remains, associated with the natural mobility of some campsite elements between seasons. Identification of receptors was static in time and based on assuming the worst case (i.e. campsite elements present at any campsite space allocated to them). A future projection of distribution and number of receptors was not performed.

The damage curves used in the analysis for Houses and Campsite elements (Table 3) were derived from the recommendations by Agència Catalana de l'Aigua (2014) and the FEMA (2001) guidelines; therefore, no specific depth-damage curve derived and calibrated for this specific site was used, which may introduce additional uncertainty to the performed analysis on induced damages. Erosion buffers were selected according to the experience from the side, and aimed to represent the impact on the protective function of the coast. Additional assessments could include the impact on the recreational function related to the loss of beach width.



The CCS based on RCP8.5 SLR had the inherent uncertainties of said projections. The effect of directional changes of incoming storms represents a hypothetical scenario for comparison purposes. Casas-Prat and Sierra (2012) predicted directional changes related to mean sea conditions, but whether these changes will also affect storms is uncertain.

Regarding the DRR measures, it was assumed that protective strategies are completely and efficiently implemented when the storm event occurs. This means that for the FRM, it was assumed that all elements in the area (campsites and houses) implemented flood-proofing measures. However, social and economic conditions influence the percentage of campsite or house owners in the area that take flood-proofing measures, likely reducing this value to below 100%. Further research is needed at the case study site for accurate estimations.

In terms of the setback analysis, we did not consider the background erosion of the area (Jiménez et al., 2017). The measure is valid to cope with storm-induced hazards, but to be efficient in time, setback distances must be increased to include the expected magnitude of decadal shoreline retreat. This also applies to infrastructural measures, as the N+D was assumed to be in place every time a storm reaches the coast. This means that beach nourishment would have to be rebuilt each time after being eroded during a storm event to maintain the 50-m increase in beach width. Moreover, it was assumed that the position and level of the barriers are adapted to the new position of the shoreline along with the predicted SLR.

Despite the limitations, the results obtained mimic the system behaviour under present conditions. For instance, the temporary capability of beach nourishment to protect the site against erosion is well known in the Tordera Delta, where several nourishments have been implemented over the last decade at both sides of the river mouth with the same outcome. The measure was completely washed out after the first incoming storm event. However, this measure should be considered in the DRR comparison, since one of the main needs of the economic activity of the area (campsites) is having a sandy beach available in front.

North of the river mouth (S'Abanell), erosion and overwash problems are the main issue for campsites and the promenade (Jiménez et al., 2011). This is well captured by the tool, which shows a notable increase of these impacts under the SLR. This is similar to the observed increase in damages due to the decadal background erosion of the site, where campsites located in unprotected areas are forced to lose the first line of elements progressively impacted by storms. The coastal promenade has also experienced increased damages over the last decades (Jiménez et al., 2011).

Thus, the primary results summarised in the following and observed as in accordance with known reality, are useful in providing coastal managers with an integrated global picture of the impacts at the site and the best measures to counter them. The overall results indicate that both sites of the river mouth are likely to double expected flooding impacts after the SLR (CCS1). South of the river, impacts are also likely to double because of a switch in the direction of incoming storms, even without an increase in the MSL (CCS2). Therefore, at that side of the river, the combination of these two factors (CCS3) is likely to triple the expected flooding risk. This is not the case at the north of the Tordera, where the orientation of the coast means that the directional switch does not imply any significant increase on the extension and magnitude of flooding. The erosion hazard is likely to increase under CCS1, and no significant increase is expected for the other two climate change scenarios (CCS2 and CCS3). The expected increase in erosion impacts is larger than for flooding, since the beach



accommodation and future MSL mean that receptors are likely to be affected by erosion. Furthermore, the magnitude of the hazard itself is expected to worsen one magnitude in the present situation.

The most efficient DRR against flooding is beach nourishment and dunes, and the best option against erosion is managed retreat. However, each measure has drawbacks from the socio-economic standpoint, which must be assessed in a further

step, since the aim of the BN tool is to objectively assess the efficiency of reducing impacts.

## 5.2 Lido degli Estensi-Spina

Regarding storm characterisation, the events were designed using triangular design storms. Although this practice is common for numerical investigations, especially for erosion issues (e.g. Carley and Cox, 2003; Corbella and Stretch, 2012), the listed assumptions may have introduced a degree of uncertainty in the modelling. This may lead to uncertainties in the

simulation of the coastal flood extension and intensity, as well as on the erosion patterns and magnitude. Regarding overall representativeness, it must be highlighted that the forcing events (historical and synthetic) were selected to cover all possible (and realistic) combinations that can affect the area.

The XBeach model setup was affected by simplifications and assumptions as well as the uncertainty related to the input data (i.e. topo-bathymetry merged from different years). A proper calibration was not implemented, but the model was validated

against the event in February 2015, evidencing a reasonable fit with observed flood inundation. However, a slight overestimation of the inundated area for the southern part of the Lido di Spina beach was demonstrated. Proper calibration and validation of the model is needed to improve the reliability of the results.

The location of receptors was estimated by cadastral maps from 2013. As an important aspect of risk assessment is the updating of exposed elements, more recent information can improve the results. For the CCS, no increase in exposure was

considered, which may lead to underestimating the impacts. For coastal areas, an increase in exposed elements was forecast (IPCC, 2012). This aspect can exacerbate coastal risk at the regional level when compared to increases in local hazard conditions driven by climate change (Sekovski et al., 2015).

Uncertainties related to consequences are linked to the choice of vulnerability functions. The selected flood-damage curve (Scorzini and Frank, 2015) was the most recently developed for river floods in Italy. Although it was developed based on

data of damage to residential buildings, it was applied here for coastal floods impacting all types of receptors. This increases the degree of uncertainty of the results, as the expected damages to a concession are likely to be lower than for a residential building for a given water depth. The methodology applied to link erosion patterns and potential damages was set *a priori* using homogeneous thresholds of 0.5 m and a 10-m buffer.

The uncertainties of the CCS definition are related to the reliability of the SLR future projections. Indeed, the RMSE of the

predicted RSLR (including subsidence) was 28 cm in the northern Adriatic (the average RMSE for the central Mediterranean was 14 cm) (Vousdoukas et al., 2016). Thus, more detailed and consistent data may lead to more reliable impact projections. Moreover, the CCS was implemented without considering implicitly in the modelling the long-term morphological



adaptation of the beach to SLR. However, this aspect can be assessed *a posteriori* through a profile response equilibrium model (Bruun, 1962).

The WD system was implemented modifying the topography in front of beach concessions using the Dune Maker 2.0 tool (Harley, 2014). The accuracy of the representation of this feature strongly depends on the alongshore resolution of the model, as the WD develops in that direction. Moreover, only one type of WD was tested, while more configurations should be investigated in which the geometry of the system is varied. It is important to underline that the efficiency of artificial dunes strictly depends on the beach width. Therefore, as observed by Harley and Ciavola (2013), the dune height and crest width (i.e. the sand volume of the dune) should be designed differently for different coastal stretches, such as the case of Lido degli Estensi-Spina, where the beach characteristics are not uniform. The location of the sand dune with respect to the MSL is another important component that can affect the final results.

The FRM measure was assumed *a priori* as completely effective. However, as its implementation depends on physical and socio-economic factors (e.g. education, economic status, etc.), which can decrease the efficiency of the measure, the assumption leads to overestimating its effect.

Despite the highlighted limitations, the comparative analysis of scenarios is robust and valid. Impacts related to the CUS are reliable in terms of magnitude and comparable with the knowledge of the area (Perini et al., 2016; Trembanis et al., n.d.). Moreover, the dominance of water level characteristics of storms on the impacts of flooding compared to wave characteristics is highlighted in previous works (e.g. Armaroli et al., 2012).

The impacts in the CCS increased compared to the CUS for all tested cases. In addition, when beach accommodation to the SLR was assessed *a posteriori*, the obtained shoreline retreat (Bruun, 1962) is 60–100 m, leading to a significant loss of the protective function of the beach. Thus, including beach accommodation would probably lead to higher erosion and flooding impacts than those previously presented.

As expected, the WD measure is effective in decreasing the impacts of flooding, as previously demonstrated by Harley and Ciavola (2013). Their research focused on two case studies in the Ravenna province, and tested different dune geometries against the impacts of the February 2012 event. That work and others (Bruun, 1983; Wells and McNinch, 1991) highlights the need for appropriate guidelines for dune implementation to limit beach manipulation due to scraping activities. Beach manipulation can lead to undesired changes in slopes (Wells and McNinch, 1991), and consequently in morphodynamics, which can locally increase the impacts of storms. In the case of Lido degli Estensi-Spina, the increase in erosion impacts on a few receptors can be attributed to the relative position between the receptors and discontinuities of the dune, which was designed as a non-continuous morphology. The induced alongshore variability increased localised erosive impacts. This was in contrast to the general decrease of erosion consequences for the majority of receptors, as demonstrated for the WD scenario compared to the configuration without DRR. The effect of the WD on adjacent beaches was not analysed. However, the scraping depths were in the range proposed by Bruun (1983) to avoid effects on adjacent beaches. The interest in FRM was related to the opportunity to merge the measure with the WD system. Indeed, the combination of measures can effectively reduce damage caused by floods.



## 6 Conclusions

In this paper, a methodological framework for storm-induced coastal risk management purposes developed within the framework of the RISC-KIT EU project was presented and applied in two coastal study sites in the North Western Mediterranean and Northern Adriatic. The study was based on the integration of the SPRC concept in a BN. This was fed

with a large number of numerical simulations obtained through a model-chain composed of process-oriented models able to reproduce multiple storm-induced hazards at the receptor scale. The BN integrates impact results that individually account for all receptors in the hinterland. The tool can be regularly updated with additional simulations and extended with new scenarios.

The choice and discretisation of storm variables to perform the analysis covered all possible and realistic combinations at

both study sites. The entire range of characteristics of forcing events was appropriately represented.

At both study sites, the implemented model chains successfully predicted the coastal response to storm events. Target hazards were suitably captured though the process-based models, which simultaneously assessed erosion and inundation.

A BN was used to integrated results, calculated at the receptor scale, from a large number of simulations to produce a robust comparative assessment. It was successful in detecting significant changes on expected impacts. Therefore, even with the

inherent uncertainties and limitations, the BN approach allows realistic scenario testing and comparisons between DRRs.

Many types of results can be extracted from the BN tool once fed with data that are easy to interpret and quantitative (and therefore comparable). As expected, this work confirmed the potential of the BN as a probabilistic data assimilation approach.

At both study sites, the approach demonstrated impact responses in the current situation in accordance with existing

knowledge on the sites. Tordera Delta, which is characterised by quick and intense erosive responses to storms, showed greater impacts to erosion than Lido degli Estensi-Spina. Inundation and erosion impacts are likely to increase in all assessed future projections at both study sites. As expected, the flooding impact in the current situation and projected increase in future scenarios is higher for receptors located closest to the shoreline or at the most low-lying areas of the hinterland (i.e. concessions at Lido di Spina and campsites at Malgrat). Regarding the impacts of future projected erosion, the obtained

increase at the Tordera Delta was significantly higher than in the Lido degli Estensi-Spina, because of the morphological accommodation response to the projected MSL. This highlights the importance of including morphological adaptation to the SLR in impact and risk assessment studies.

The DRR assessment highlighted as effective the construction of artificial dunes as protection against inundation at both study sites, even when compared to other measures such as managed retreats or flood resilience measures applied to all

receptors. However, the dune was less effective and sometimes ineffective against erosion at both study sites. As expected, and derived from results for the Tordera Delta, dune performance against flooding improved when tested along with beach nourishment. However, beach nourishment did not improve dune performance against erosion. Managed retreat seems to be the best option to tackle the impacts of erosion.



The approach can be further improved by addressing the limitations discussed in Section 5, including data and methodological improvements that may increase computational efforts.

In conclusion, coastal management can be significantly improved by methodologies based on the integration of large amounts of data, stochastically condensed so that multiple scenarios can be easily compared. Uncertainties due to data quality, numerical approximation, simplifications, and assumptions will always be present. However, the assimilation of results and their uncertainties though a BN provides robust comparison across different conditions. Therefore, the observed variations of impacts, when significant, will help decision-makers select between strategic alternatives of DRRs.

**Acknowledgement**s. This work was conducted in the framework of the RISC-KIT (GA 603458) and PaiRisClima (CGL2014-55387-R) research projects funded by the EU FP7 and Spanish Ministry of Economy and Competitiveness and Feder respectively. MSV was supported by a PhD grant from the Spanish Ministry of Education, Culture and Sport. ED was supported by the Consorzio Futuro in Ricerca (Ferrara, Italy) through the RISC-KIT project and a co-funded PhD grant from the Ministry of Education, University and Research and the Department of Physics and Earth Science of the University of Ferrara. During the preparation of the paper, ED and PC were supported by the H2020 ANYWHERE Project (GA700099). The authors thank the Institut Cartogràfic i Geológic de Catalunya for supplying aerial photographs and Lidar data and Puertos del Estado of the Spanish Ministry of Public Works for supplying wave and water level data. For the Spanish case study, we wish to thank the Servizio Geologico Sismico e dei Suoli of the Emilia-Romagna Region, in particular Luisa Perini, for providing input data and comments on the outcomes. For the Italian case study, we are grateful to Clara Armaroli for helping in the application of the methodology, and the RISC-KIT consortium for their support during the entire project.

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





**Figure 1: Regional and local contexts: A1) the central-northern Catalan coast; B1) Emilia-Romagna coast; A2) local hotspots of Tordera Delta; B2) local hotspots of Lido degli Estensi-Spina (2b). The main locations (red dots), wave buoys (red triangles), tide gauge (red diamond), and the CSS (red squares). The domains of the large-scale and local models (dashed red lines) are highlighted for each box.**





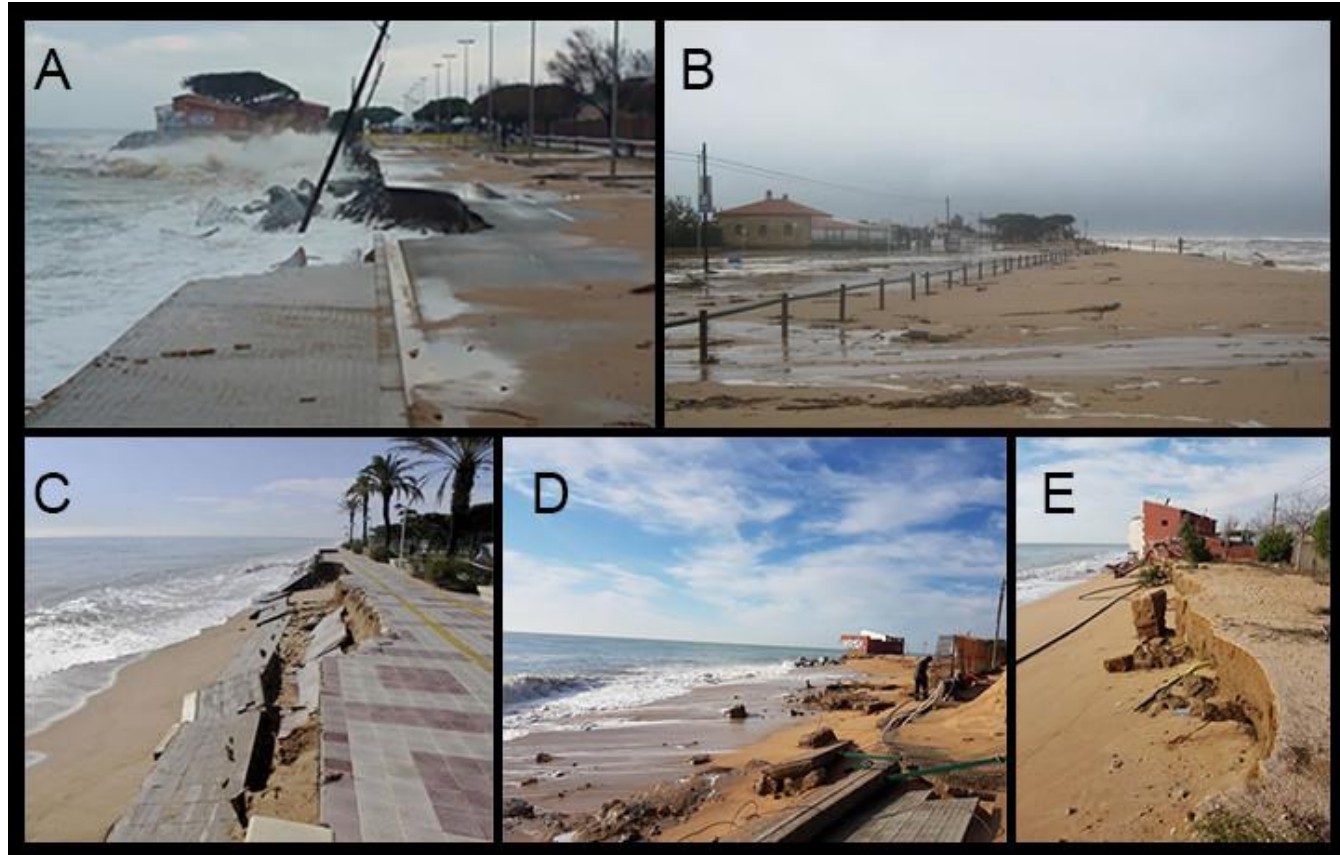

**Figure 2: Impacts on the Tordera Delta. Destruction of a road at Malgrat (A); overwash at campsites north of the river mouth (B); destruction of the promenade north of the river mouth (C); beach erosion, and damage to utilities and buildings at Malgrat (D and E).**





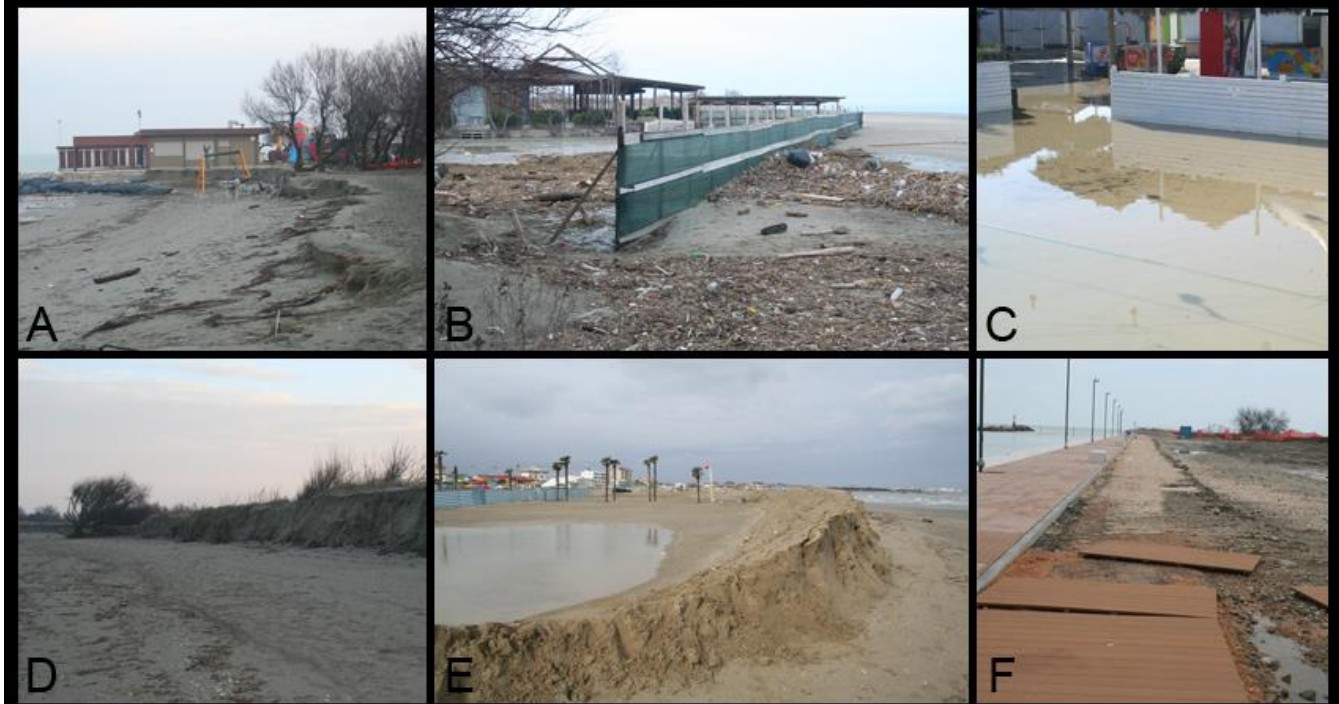

**Figure 3: Impacts of the event in February 2015 on the Lido degli Estensi-Spina case study area. Impacts of erosion and flooding on concessions at Lido di Spina south (A, B) and Lido degli Estensi (C); sandy scarp due to the erosion of the dune in the south of Lido di Spina (D); eroded Winter Dune in Porto Garibaldi (E); damages to the Porto Canale front at the Lido degli Estensi (F).**





**Figure 4: General methodology. (I) The SPRC conceptual framework is implemented through (II) a model chain, which consists of a propagation module of the source (S) and a process-oriented module for the coastal area reproducing the pathway (P). Then, (III) the consequences (C) are calculated based on the computed hazards (H) at the receptor (R) scale by using vulnerability relations (i.e. hazard-consequences functions). In the last step (IV), all variables including source boundary conditions (BC) are fitted in a BN, adding impacts after the implementation of measures (M).**





**Figure 5: DRR measures at Tordera Delta. Coastal setbacks (20, 50, and 75 m) and Infrastructural Defence (beach nourishment at Malgrat beach + artificial dune at S'Abanell and Malgrat beaches).**





**Figure 6: Artificial winter dunes in Emilia-Romagna: A) Winter dune in Porto Garibaldi (Comacchio, Italy); B) Building of a winter dune by beach scraping at Lido di Dante (Ravenna, Italy) (Harley, 2014); C) Representative model profiles at Lido di Spina north (original: black solid line; with winter dune DRR: red dashed line)..**





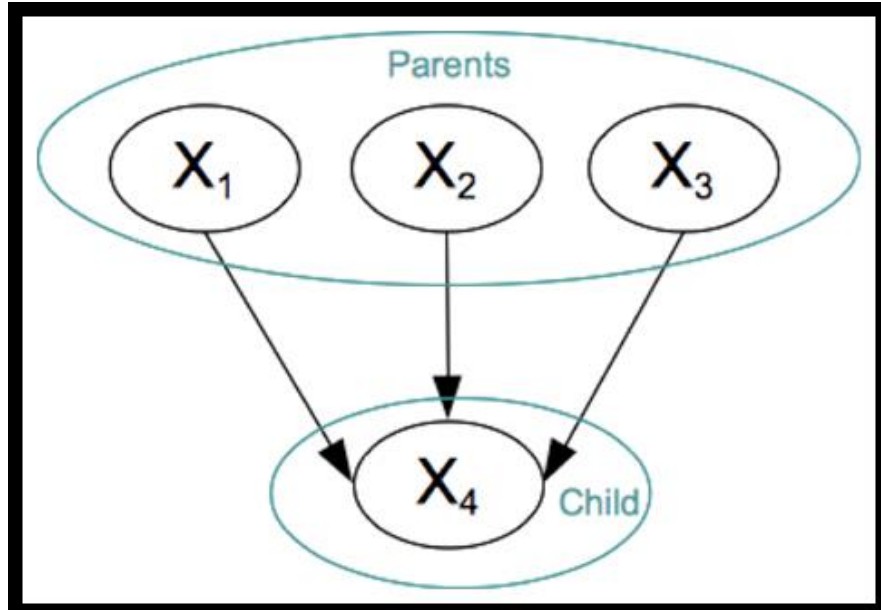

Figure 7: BN graph with four nodes.



Figure 8: Bayesian Network scheme for the Tordera Delta site.





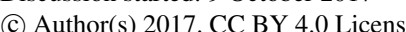

Figure 9: Bayesian Network scheme for the Lido degli Estensi-Spina site.





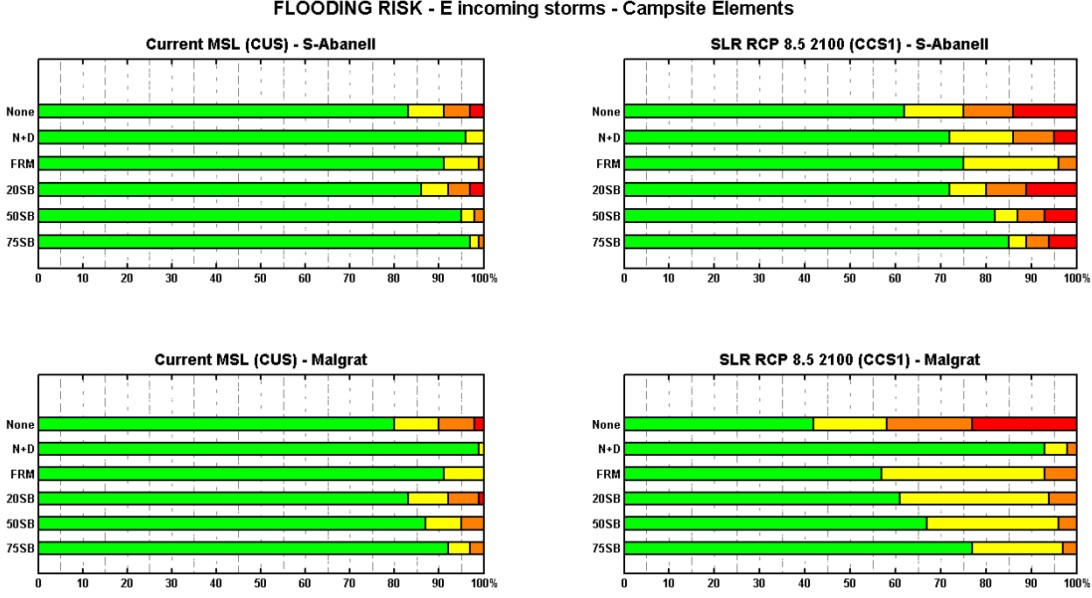

**Figure 10: Distribution of campsite elements at every level of flooding risk. Top-left: current scenario at S'Abanell; Top-right: climate change scenario 1 (SLR) at S'Abanell; Bottom-left: current scenario at Malgrat; Bottom-right: climate change scenario 1 (SLR) at Malgrat. Each bar in a panel represents a DRR configuration ('None': no DRR implemented; 'N+D': Nourishment and Dune; 'FRM': Flood Resilience Measures; '20SB, 50SB, and 75SB': 20, 50, and 75 m setbacks, respectively).**





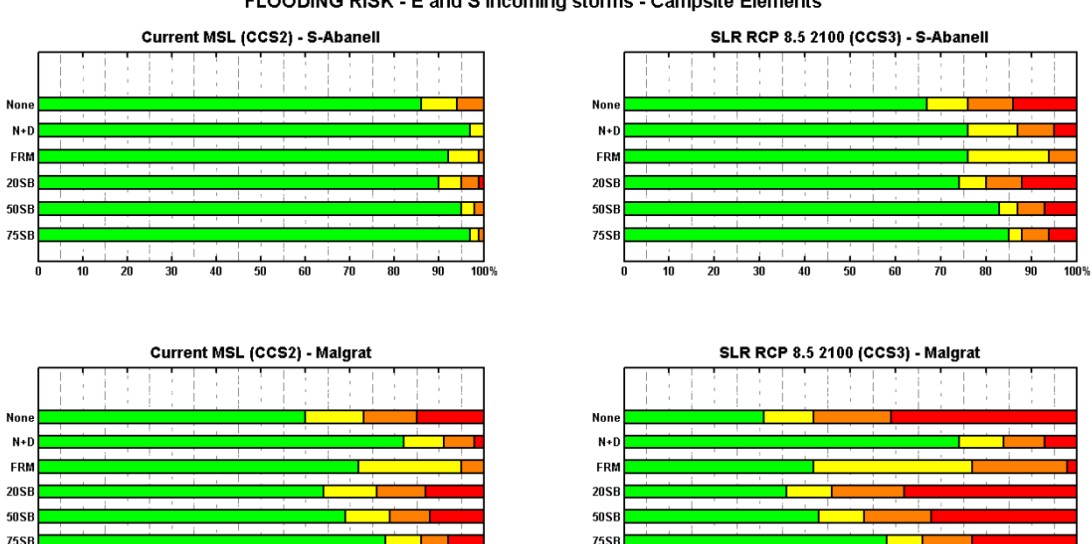

**Figure 11: Distribution of campsite elements at every level of flooding risk. Top-left: climate change scenario 2 (50-50% east-south storms) at S'Abanell; Top-right: climate change scenario 3 (50-50% of east-south storms + SLR) at S'Abanell; Bottom-left: climate change scenario 2 (50-50% east-south storms) at Malgrat; Bottom-right: climate change scenario 3 (50-50% of east-south storms + SLR) at Malgrat. Each bar in a panel represents a DRR configuration ('None': no DRR implemented; 'N+D': Nourishment and Dune; 'FRM': Flood Resilience Measures; '20SB, 50SB, and 75SB': 20, 50, and 75 m setbacks, respectively).**


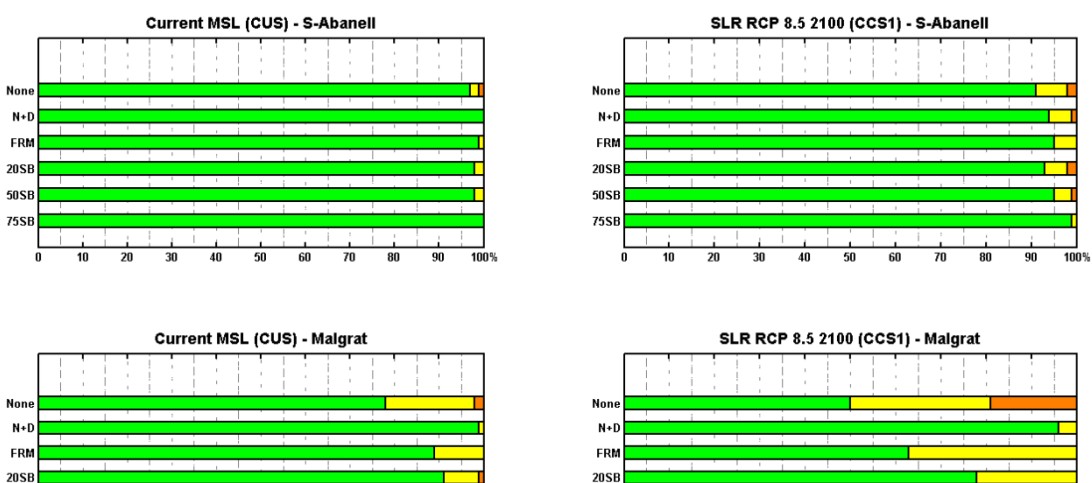

**Figure 12: Distribution of houses at every level of flooding risk. Top-left: current scenario at S'Abanell; Top-right: climate change scenario 1 (SLR) at S'Abanell; Bottom-left: current scenario at Malgrat; Bottom-right: climate change scenario 1 (SLR) at Malgrat. Each bar in a panel represents a DRR configuration ('None': no DRR implemented; 'N+D': Nourishment and Dune; 'FRM': Flood Resilience Measures; '20SB, 50SB, and 75SB': 20, 50, and 75 m setbacks, respectively).**





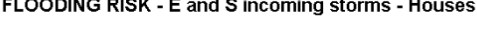

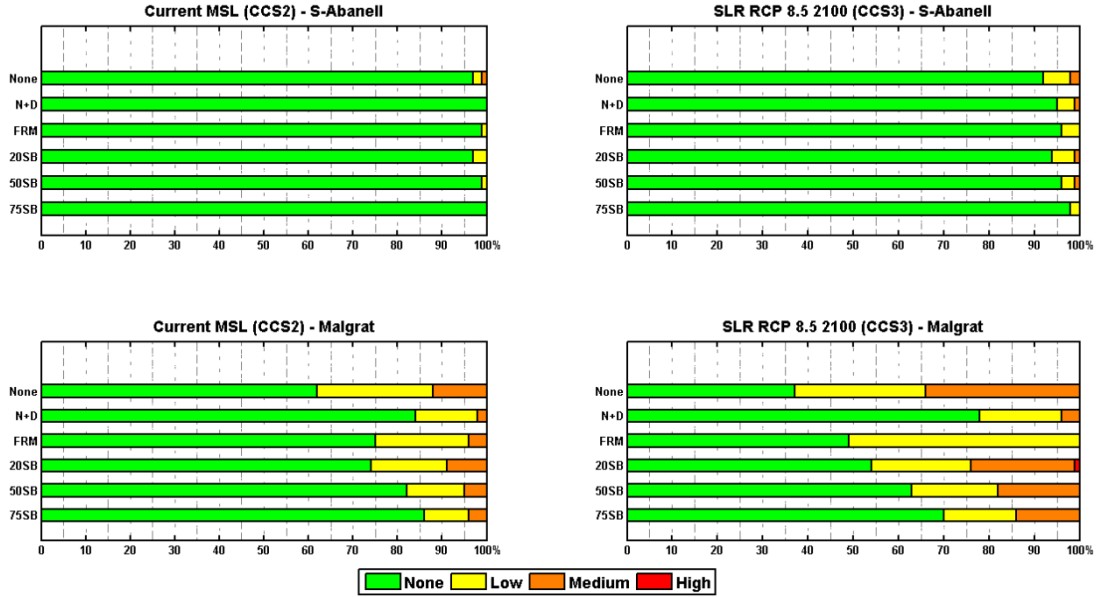

**Figure 13: Distribution of houses at every level of flooding risk. Top-left: climate change scenario 2 (50-50% east-south storms) at S'Abanell; Top-right: climate change scenario 3 (50-50% of east-south storms + SLR) at S'Abanell; Bottom-left: climate change scenario 2 (50-50% east-south storms) at Malgrat; Bottom-right: climate change scenario 3 (50-50% of east-south storms + SLR) at Malgrat. Each bar in a panel represents a DRR configuration ('None': no DRR implemented; 'N+D': Nourishment and Dune; 'FRM': Flood Resilience Measures; '20SB, 50SB, and 75SB': 20, 50, and 75 m setbacks, respectively).**




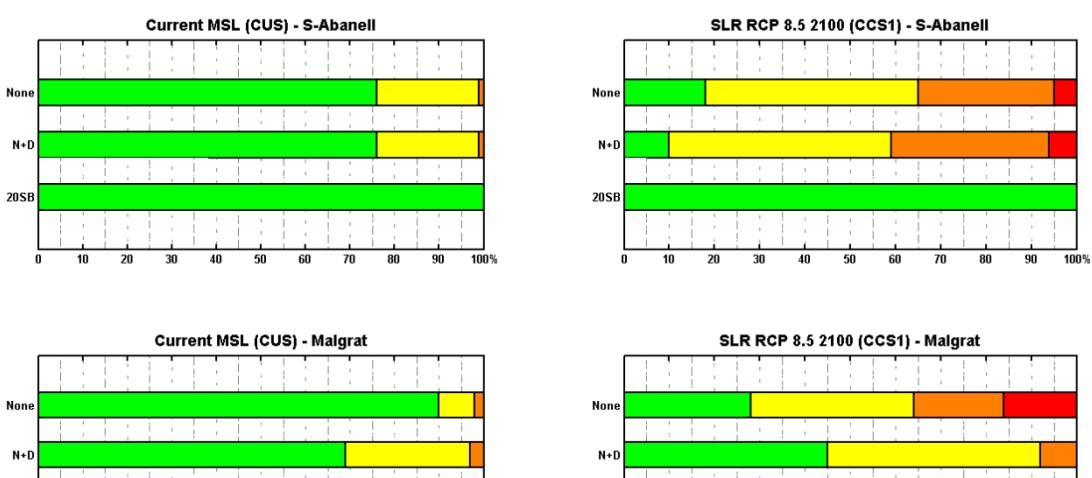

**Figure 14: Distribution of campsite elements at every level erosion risk. Top-left: current scenario at S'Abanell; Top-right: climate change scenario 1 (SLR) at S'Abanell; Bottom-left: current scenario at Malgrat; Bottom-right: climate change scenario 1 (SLR) at Malgrat. Each bar in a panel represents a DRR configuration ('None': no DRR implemented; 'N+D': Nourishment and Dune; 'FRM': Flood Resilience Measures; '20SB, 50SB, and 75SB': 20, 50, and 75 m setbacks, respectively).**

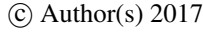


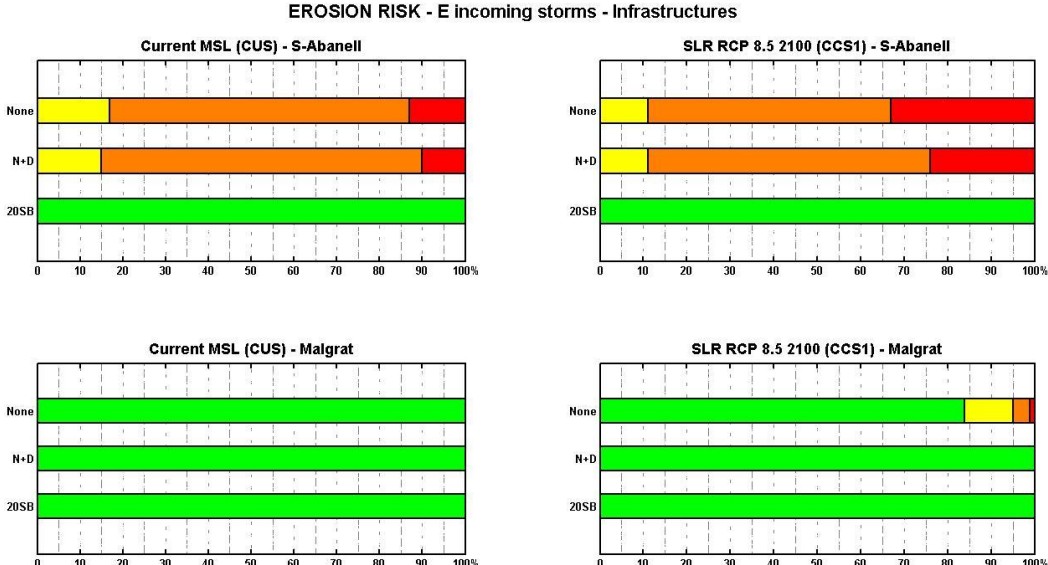

**Figure 15: Distribution of Infrastructures at every level erosion risk. Top-left: current scenario at S'Abanell; Top-right: climate change scenario 1 (SLR) at S'Abanell; Bottom-left: current scenario at Malgrat; Bottom-right: climate change scenario 1 (SLR) at Malgrat. Each bar in a panel represents a DRR configuration ('"None"': no DRR implemented; '"N+D"': Nourishment and Dune; '"FRM"': Flood Resilience Measures; '"20SB, 50SB, and 75SB"': 20 , 50, and 75  m setbacks, respectively).**





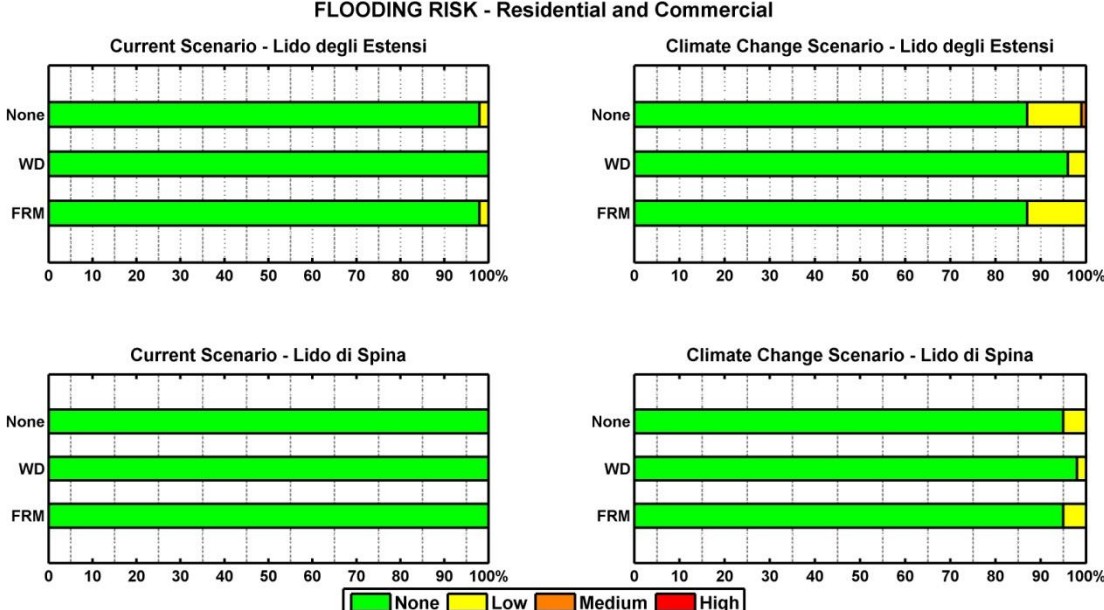

**Figure 16: Distribution of residential and commercial buildings for every level of flooding risk. Top left: current scenario at Lido degli Estensi; Top right: climate change scenario at Lido degli Estensi; Bottom left: current scenario at Lido di Spina; Bottom right: climate change scenario at Lido di Spina. Each bar in a panel represents a DRR configuration ('None': no DRR implemented; 'WD': Winter Dune; 'FRM': Flood Resilience Measures).**





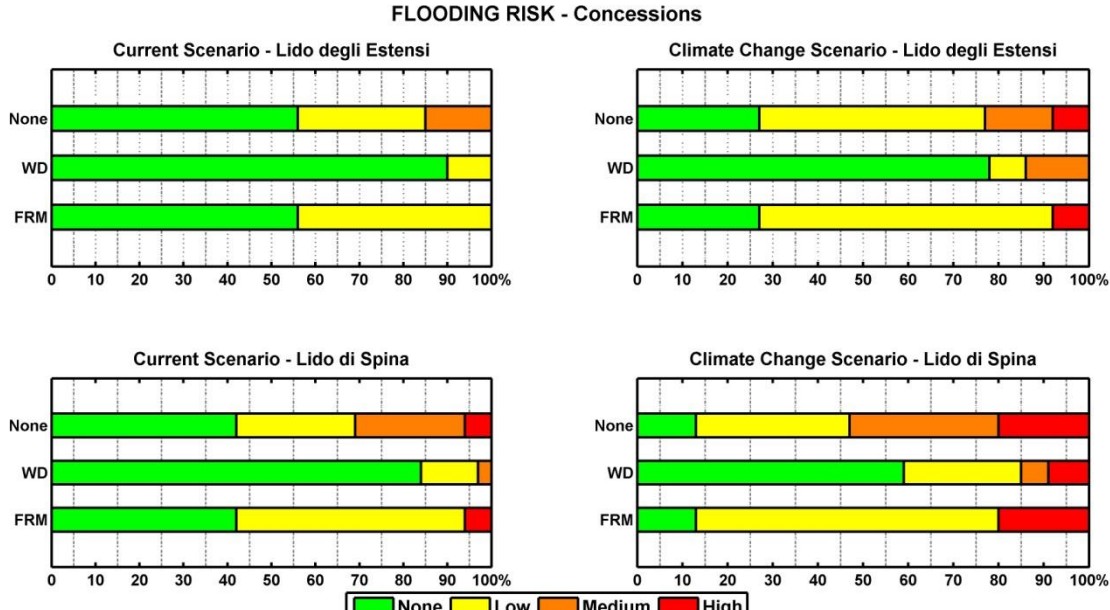

**Figure 17: Distribution of concessions for every level of flooding risk. Top left: current scenario at Lido degli Estensi; Top right: climate change scenario at Lido degli Estensi; Bottom left: current scenario at Lido di Spina; Bottom right: climate change scenario at Lido di Spina. Each bar in a panel represents a DRR configuration ('None': no DRR implemented; 'WD': Winter Dune; 'FRM': Flood Resilience Measures).**




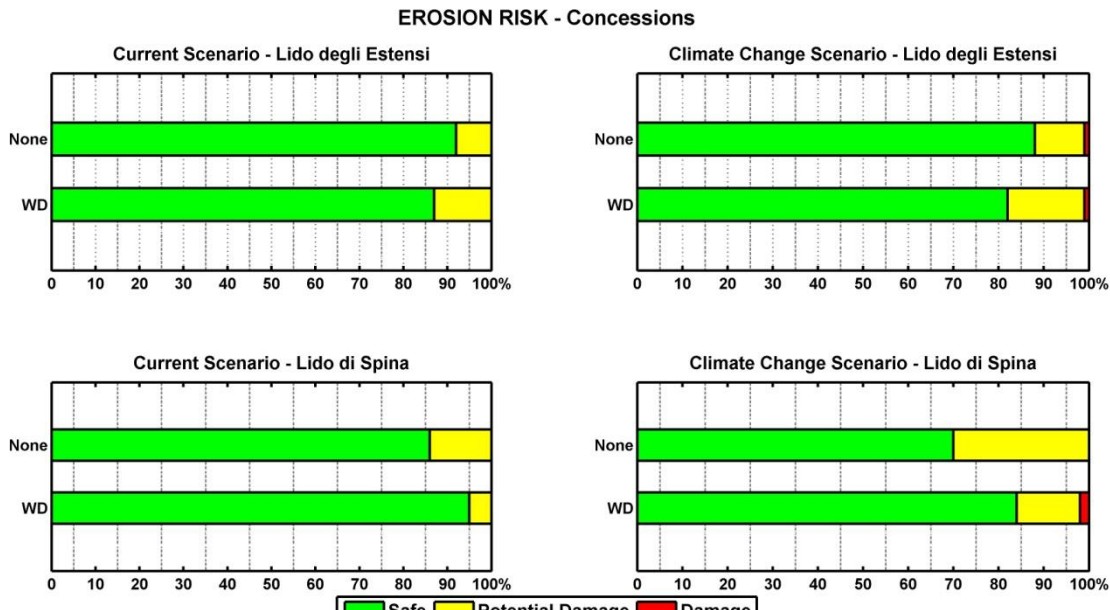

**Figure 18: Distribution of concessions for every level of erosion risk. Top left: current scenario at Lido degli Estensi; Top right: climate change scenario at Lido degli Estensi; Bottom left: current scenario at Lido di Spina; Bottom right: climate change scenario at Lido di Spina. Each bar in a panel represents a DRR configuration ('None': no DRR implemented; 'WD': Winter Dune; 'FRM': Flood Resilience Measures).**





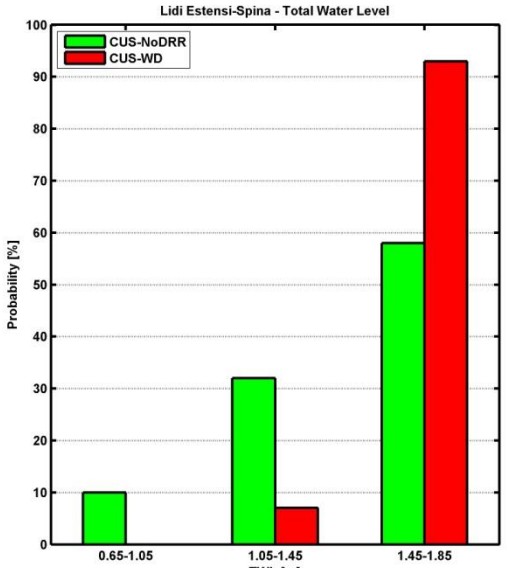 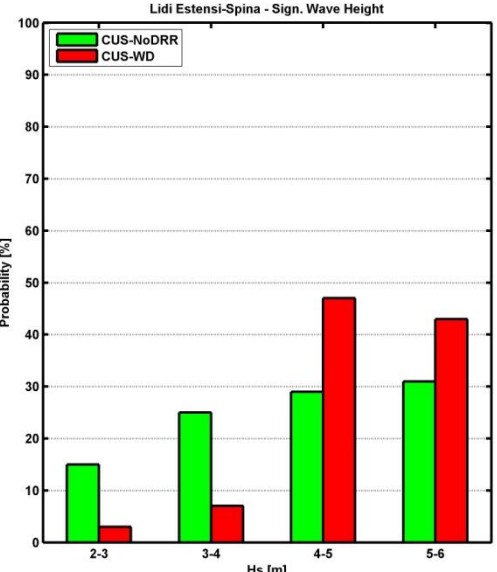

**Figure 19: Distribution of boundary conditions (TWL on the left and Hs on the right) that generate flood damages in the current scenario for Lido degli Estensi-Spina. The configuration without DRR (green bars) and for the implementation of the WD DRR (red bars) were compared.**





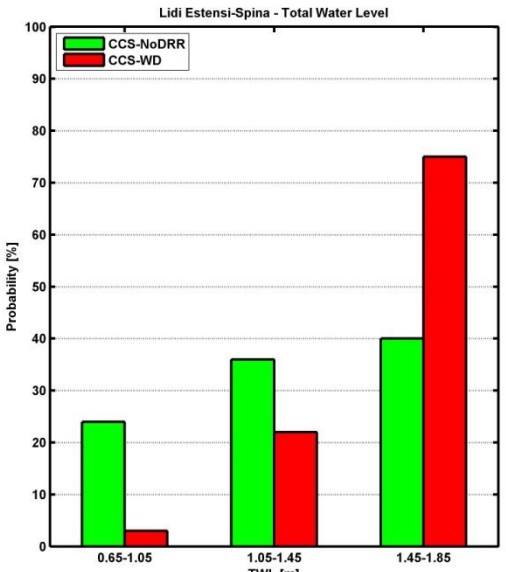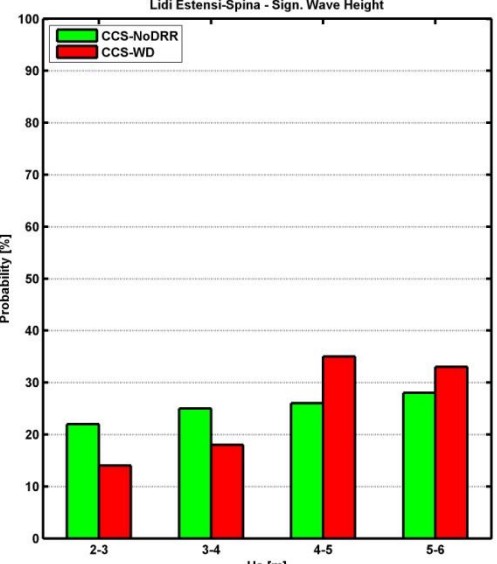

**Figure 20: Distribution of boundary conditions (TWL on the left and Hs on the right) that generate flood damages in the climate change scenario for Lido degli Estensi-Spina. The configuration without DRR (green bars) and under the implementation of the WD DRR (red bars) were compared.**





**Table 1: Source characterization. Variable discretization applied at the study sites.**

|  | Hs ranges (m) | Storm Duration (h) | Incoming direction (ºN) | Water level (m) |
|---|---|---|---|---|
| TORDERA DELTA | 2 to 3 | 6 to 30 | 30-135 (E) | 0 to 0.6 m |
|  | 3 to 4 | 30-65 | 135-220 (S) | Current + SLR |
|  | 4 to 5 |  |  |  |
| LIDO DEGLI ESTENSI-SPINA | 2 to 3 | 6 – 68 | 60 to 90 | 0.65 to 1.05 |
|  | 3 to 4 |  |  | 1.05 to 1.45 |
|  | 4 to 5 |  |  | 1.45 to 1.85 |
|  | 5 to 6 |  |  | Current + SLR |



**Table 2: Distribution of receptors at the Tordera Delta study site.**

| Area | No. of Houses | No. of Campsite Elements |
|---|---|---|
| Area 1 (0 to 20 m *Malgrat de Mar*) | 16 | 45 |
| Area 2 (20 to 50 m *Malgrat de Mar*) | 10 | 71 |
| Area 3 (50 to 75 m *Malgrat de Mar*) | 8 | 169 |
| Area 4 (> 75 m *Malgrat de Mar*) | 46 | 509 |
| Area 5 (0 to 20 m *Blanes*) | 1 | 95 |
| Area 6 (20 to 50 m *Blanes*) | 4 | 156 |
| Area 7 (50 to 75 m *Blanes*) | 7 | 72 |
| Area 8 (> 75 m *Blanes*) | 51 | 189 |
| Total | 143 | 1306 |



**Table 3: Vulnerability relations for houses and campsite elements at the Tordera Delta study site with and without DRR measures (FRM)..**

| Water depth at the receptor (m) | Relative Damage (%) | | | |
|---|---|---|---|---|
| | Houses | Campsites | Houses - FRM | Campsites - FRM |
| 0 | 0 | 0 | 0 | 0 |
| 0-0.3 | 18.3 | 50 | 0 | 0 |
| 0.3-0.6 | 26.5 | 71 | 18.3 | 50 |
| 0.6-0.9 | 33.2 | 82 | 18.3 | 50 |
| 0.9-1.5 | 44.7 | 89 | 26.5 | 71 |
| 1.5-2.1 | 54.1 | 91 | 33.2 | 82 |
| 2.1-3.0 | 64.5 | 100 | 44.7 | 89 |
| 3.0-4.0 | 71.2 | 100 | 54.1 | 91 |
| 4.0-5.0 | 75 | 100 | 64.5 | 100 |



**Table 4: Distribution of the receptors at Lido degli Estensi and Lido di Spina.**

| Area | Residential and Commercial Buildings | Concessions |
| --- | --- | --- |
| Lido degli Estensi - Seafront | 26 | 16 |
| Lido di Spina - Seafront | 47 | 28 |





**Table 5: Vulnerability relation for flooding adopted for the receptors at Lido degli Estensi-Spina without (A) and with DRR measures (B).**

| Flood Depth [m] | Flood Relative Damage Factor [-] | |
|---|---|---|
| | A - adapted from Scorzini and Frank (2015) | B - modified considering the FRM |
| 0 | 0 | 0 |
| <0.3 | <0.1 | <0.1 |
| 0.3 - 0.7 | 0.1 - 0.2 | <0.1 |
| 0.7 - 1.1 | 0.2 - 0.3 | 0.2 - 0.3 |
| >1.1 | >0.3 | >0.3 |