# Peer review of "Linking source with consequences of coastal storm impacts for climate change and risk reduction scenarios for Mediterranean sandy beaches"

_Natural Hazards and Earth System Sciences, 2017_

## Referee Comment (RC1) · Anonymous Referee #1 · 7 Nov 2017

**general comments**

The authors present a risk assessment for coastal storm impacts to support decisions on disaster risk reductions. For that purpose a Bayesian network approach is used to link process-oriented models, that predict the hazards at the receptors, with vulnerability relations to obtain the final expected impact. In a case study two Mediterranean sandy coasts are considered.

The paper is well structured and provides a well-argued motivation and problem definition. Further the study areas are described in detail and underline the relevance of the presented study. Despite a good structure in the Methodology section, some aspects of the method remain unclear to me, which might be due to the complexity of the model chain. This affects especially the Bayesian network (BNs) application. Eventhough I am familar with the constuction and application of BNs, I have problems to follow the construction (i.e. parameter setting) of the BNs and to understand the motivation for and advantages of using BNs in the presented context. The results section provides an extensive analysis of different climate change and adaptation scenarios for the considered Mediterranean coasts. Yet, I did not understand which storm intensities are considered here (this mights be due to a missing understanding of the methodology). The discussion names several aspects that pose challenges or are negleted in the presented model approach and might consequently be taggled in follow up studies. Yet, to my impression important critical points of the presented approach are missing, as will be specified in the specific comments.

**specific comments**

I found it quite difficult to keep track of all abbreviations used in the paper.

abstract:

line 15: "a large number of storm characteristics" What is a large number? To my understanding 3-4 storm characteristics were consindered.

line 17: "The tool has been proven successful in reproducing current coastal responses at both sites". I could not find any model verification in the paper. Only a reference to a paper that verifies a part of the model.

Section 3.2:

page 7, line 7: The discretization of the variables is hardly motivated. What is the motivation for choosing 2 or 3 or n intervals for a certain variable? How are the interval boundaries selected (equidistant, equifrequent, entropy-based, ...)? How is the

probable range determined (only so far observed values)? Some information about the distribution of these variables might help to motivate the discretization.

What are the effects of discretizing?

In the discussion it is mentioned that accuracy comes at computational costs, but this information is quite sparse (no information about number of intervals scales with computational costs or what are the computational costs of the current network for parameter determination and for inference).

page 7, line 12: "time series" of what?

page 7, line 19-20: the "(24 simulated storms)" confused me? Why do you consider 24 simulated storms for 12 state combinations?

page 7, line 24: What are synthetic triangular events?

page 7, line 27-28: "water level and Hs are uncorrelated" <- a reference is needed?

page 8, line 1: How are the driving variables identified? Why are the remaining variables considered to have no effects?

How is the distribution of the storm definining variables defined? To my current understanding an equal amount of storms for each state combination is considered, which infers a uniform distribution of the variables. Yet, I would expect that small Hs values or smaller durations are more likely than higher values? Is this accounted for?

Section 3.3:

Only one event (storm) is considered for each combination of states. Yet, similar events might result in different outcomes. Further, the applied model chain seems to provide deterministic results. Consequently no uncertainties are considered/captured in the model construction. Since BNs are explicitly designed to capture uncertainties, I wonder why this approach was chosen here.

[Figure]

The distribution of hazard at the receptors results from the different location of the receptors, but does not reflect the uncertainty related to the inundation or erosion at a specific object. In a strict sence, I would not judge the resulting distribution to represent probabilities.

Section 3.4:

To model the consequences flood damage curves are applied. Those are generally related with huge uncertainties (a wide range of relative damage can be observed for equal water levels), which are again neglegted and not included in the BN. On top, a damage curve that was derived for river floodind is applied. Since the process of storm surges is very different from river flooding the applicability should be discussed.

In terms of risk levels the values selected for both study sites differ significantly. E.g. medium impact building damage ranges from 0.26 to 0.45 compared to 0.1 - 0.2. Why are these intervals chosen?

Section 3.6:

To my understanding the probability tables of the BN are constructed by simulating a strom scenario for each combination of states and running the deterministic model chain to receive a predicted hazard value for each receptor in the study area. Due to the deterministiv character of the model chain, the resulting distribution for the hazard variables does not represent probabilities, but the expected fraction of receptors with the single hazard levels or impact levels respectively.

Since no uncertainties are considered, I see no need to apply BNs in this context. The same calculations can be done by applying the model chain directly. A direct application of the model chain would also avoid the discretization of the variables and consequently achieve a higher accuracy.

In my point of view the revised paper should either do without the BN approach or account for the uncertainties related to the single model components.

Section 4:

I do not understand which storm intensity is considered here? Are the presented results the joint distribution for all possible combinations of storm characteristics? If so, what is the meaning? Is this a kind of average storm? <- I don't think so.

I would rather prefer to consider specific storm scenarios in combination with their return period. E.g. what are the effects of DRR measures for a once-in-a-year or once-in-10-years event or for an extreme event.

To judge the efficiency of DRR measures, it would also be interesting to get some information about the costs of their implementation and their probability of failure.

Section 5:

page 21, line 11: "a first test to check the method was presented" <- Where?

I could not find any validation of the presented model. There is only a reference to a (not published) paper to validate the hazard component of the model chain.

page 21, line 11-15: A more detailed justification for the choosen amount of intervals and the interval boundaries, would be nice. Additionally, some information about how the computational costs scale with the number of intervals could be provided.

Several uncertainties related to the study are not disscussed (see comments about section 3).

**technical corrections**

page 4, line 24: 2-3m?

page 5, line 16-17: Armaroli et al (2012) is cited double

page 11, line 19: check >0.05m or >0.5m

page 14, line 31: "it also provided ..." What is "it"?

---

## Author Comment (AC1) · 5 Dec 2017

RC: Referee Comments AA: Authors' Answers

RC0: #### general comments The authors present a risk assessment for coastal storm impacts to support decisions on disaster risk reductions. For that purpose a Bayesian network approach is used to link process-oriented models, that predict the hazards at the receptors, with vulnerability relations to obtain the final expected impact. In a case study two Mediterranean sandy coasts are considered. The paper is well structured

and provides a well-argued motivation and problem definition. Further the study areas are described in detail and underline the relevance of the presented study. Despite a good structure in the Methodology section, some aspects of the method remain unclear to me, which might be due to the complexity of the model chain. This affects especially the Bayesian network (BNs) application. Even though I am familiar with the construction and application of BNs, I have problems to follow the construction (i.e. parameter setting) of the BNs and to understand the motivation for and advantages of using BNs in the presented context. The results section provides an extensive analysis of different climate change and adaptation scenarios for the considered Mediterranean coasts. Yet, I did not understand which storm intensities are considered here (this might be due to a missing understanding of the methodology). The discussion names several aspects that pose challenges or are neglected in the presented model approach and might consequently be taggled in follow up studies. Yet, to my impression important critical points of the presented approach are missing, as will be specified in the specific comments.

AA: Thanks for comments and suggestions on submitted manuscript. In what follows we answer to all comments/suggestions/questions raised by the reviewer. First the reviewer's comment is literally included and it is followed by the corresponding authors' answer.

RC1: #### specific comments I found it quite difficult to keep track of all abbreviations used in the paper. AA: We understand that the use of a large number of abbreviations could be cumbersome, especially for a long text as this manuscript. We have reviewed the text and reduced the number of abbreviations to a minimum.

abstract: RC2: line 15: "a large number of storm characteristics" What is a large number? To my understanding 3-4 storm characteristics were considered. AA: When we mention "storm characteristics" we refer to storms defined in terms of a combination of Hs, duration, direction and water level. In each site, we have selected 12 storm combinations for each water level (SLR) scenario, i.e 24 storm combinations per site.

This is clearly specified in the text (including table 1). Then each storm combination is represented by 2 different simulations changing slightly the storm variables inside the category ranges. We have modified this sentence in the abstract by the following "Process-oriented models are used to predict hazards at the receptor scale which are converted into impacts through vulnerability relations. In each site, a total of 24 storms have been simulated and obtained results are integrated by using a Bayesian Network to link forcing characteristics with expected impacts through conditional probabilities."

RC3: line 17: "The tool has been proven successful in reproducing current coastal responses at both sites". I could not find any model verification in the paper. Only a reference to a paper that verifies a part of the model. AA. Thanks. In order to avoid confusion with morphodynamic model validation we have changed the sentence to "Consultations with local stakeholders and experts have shown that the tool is valuable for communicating risks and the effects of risk reduction strategies. The tool can therefore be valuable support for coastal decision making."

Section 3.2: RC4: page 7, line 7: The discretization of the variables is hardly motivated. What is the motivation for choosing 2 or 3 or n intervals for a certain variable? How are the interval boundaries selected (equidistant, equifrequent, entropy-based, ...)? How is the probable range determined (only so far observed values)? Some information about the distribution of these variables might help to motivate the discretization. What are the effects of discretizing? In the discussion it is mentioned that accuracy comes at computational costs, but this information is quite sparse (no information about number of intervals scales with computational costs or what are the computational costs of the current network for parameter determination and for inference).

AA4: To avoid confusion in this section we will remove "Storm scenarios are defined [. . .] were selected for use in the analysis." (lines.6-9), since we already explain that storm scenarios are combinations of variable values covering the typical storm condition at each study site. The description of bins only makes sense in terms of the BN, which we have not explained yet in this section. Then we will motivate the discretisation

of variables in section 3.6.1. Thus, we will add in that section the following explanation: "In the boundary conditions' variables, ranges are selected equidistant covering the so far observed values at each study site (Table 1). Additional non-observed ranges are introduced only to account for SLR. The exact number of intervals is a compromise between accuracy and computational effort. Each combination in Table 1 is simulated here twice, in order to account for variability inside bins. Then, all simulations are repeated for the DRR scenario affecting the hazard. Therefore, a total number of 96 runs of the whole model train were required in the current bin set-up. More simulations would capture in better detail variability inside source combination due to bin discretization, but as a first application of the methodology the computational effort was limited. The current work took, as a reference, 2 months of model train simulations in a 48 thread cluster for the Tordera Delta case."

RC5: page 7, line 12: "time series" of what? We have modified the sentence as "In addition, time series of waves (either bulk Hs, Tp and mean direction or spectrum) and water levels during each storm event were used when this information was available.".

RC6: page 7, line 19-20: the "(24 simulated storms)" confused me? Why do you consider 24simulated storms for 12 state combinations? See answers to comments 2 and 4. In addition to avoid confusion, the sentence is rephrased to avoid the brackets. The new sentence is "The selected source combinations are shown in Table 1. These lead to 12 combinations defining the source that must be tested in the current MSL and another 12 in the future MSL scenario. Each combination of states is simulated twice by means of slightly different storms to account for variability inside the variable ranges, leading to 24 storms in the current MSL and 24 in the future projected one"

RC7: page 7, line 24: What are synthetic triangular events? We have changed the text to: "To include a full range of combinations, the remaining eight storms were completed by using combinations of Hs-duration-direction not previously recorded. These events were modelled assuming they follow a triangular-shaped development with the peak intensity at the half of their duration (see e.g. McCall et al. 2010; Poelhekke et al.,

2016)."

RC8: page 7, line 27-28: "water level and Hs are uncorrelated" <- a reference is needed? The reference is (Mendoza and Jiménez, 2008) which was located at the end of the next sentence. We have changed its position.

RC9: page 8, line 1: How are the driving variables identified? Why are the remaining variables considered to have no effects? How is the distribution of the storm defining variables defined? To my current understanding an equal amount of storms for each state combination is considered, which infers a uniform distribution of the variables. Yet, I would expect that small Hs values or smaller durations are more likely than higher values? Is this accounted for? AA9: This comment has two parts. First, the identified driving variables: In Tordera Delta it is argued that ranges of Tp are quite similar amongst storms and surge isn't a relevant variable especially when compared to waves, and both are motivated with references. For the Italian case, better motivation of the selection of drivers and justification of left-out variables is now provided. The first sentences of the Lido degle Estensi-Spina paragraph are now "For the Lido degli Estensi-Spina case study, the source variables, identified as drivers of the impacts of flooding and erosion, were the maximum Hs and maximum TWL of the storm event. The literature for the area recognizes these as main important variables with TWL having more importance (Armaroli et al 2009, 2012). In addition, the relative sea level rise (RSLR) was considered as a Boolean variable to represent the CCS. The direction of the storms was not considered as a source characteristic variable since storms are either ENE or SE, and each Hs-TWL combination is simulated twice accounting for variability inside this directional range. The source combinations were classified into the variable ranges shown in Table 1" Second, the distribution of the storm defining variables: The BN is trained with equal representation of all variable combinations, these means that once the Bayesian is trained, the "prior" probabilities of storm variables are uniform. The main reason is that the extreme value probability distributions of the source variables are not known for the two sites of interest and estimating them

was beyond the scope of this study. However, once relative frequencies of different events are available, the source nodes could be retrained and the distribution of all hazard and consequence nodes would be updated automatically. Additionally, the user could also test how different assumptions on the source variable distributions would change the hazard and impact estimates. Nonetheless, the main strength of the BN at its current stage is that it enables decision makers to explore different scenarios and helps them to design robust strategies (i.e., strategies that are successful under most scenarios.). This will be better motivated in Results section and Discussion. See answers on comments on those sections.

Section 3.3: RC10: Only one event (storm) is considered for each combination of states. Yet, similar events might result in different outcomes. Further, the applied model chain seems to provide deterministic results. Consequently, no uncertainties are considered/captured in the model construction. Since BNs are explicitly designed to capture uncertainties, I wonder why this approach was chosen here. The distribution of hazard at the receptors results from the different location of the receptors, but does not reflect the uncertainty related to the inundation or erosion at a specific object. In a strict sense, I would not judge the resulting distribution to represent probabilities. AA10: We answer the comment in splitting it in different shorter pieces: RC10.1: Only one event (storm) is considered for each combination of states. Yet, similar events might result in different outcomes. AA10.1. Two storms are simulated for each combination (we agree that this was not properly explained see previous answers). Furthermore, the user could select storms belonging to for example Hs = 3-4 meters with waves coming from the East for current MSL but leaving the duration unconstrained as an uncertain variable. In such a case the obtained output would be the integrated result from 4 simulations (2 direction categories that are represented by 2 simulations each with different values of duration). Therefore, results will account for the uncertainty on duration for a given (certain) Hs. In practice, this could be relevant because storm forecasts could contain more certainty on some variables than on others, for example as a result of ensemble forecasts. For a more detailed discussion how the BN tool can deal with ensemble forecasts also see section 5 in Jäger et al. 2017. A Bayesian network approach for coastal risk analysis and decision making. Coastal Engineering (in press). RC10.2. Further, the applied model chain seems to provide deterministic results. Consequently, no uncertainties are considered/captured in the model construction. AA10.2. Reviewer is right, we do not account for uncertainties inherent to individual models. Quantifying uncertainties of individual models is another study in itself (e.g. Wagenaar et al. 2016. Uncertainty in flood damage estimates and its potential effect on investment decisions. NHESS, 16, 1-14), and it is beyond the scope of this manuscript to do such analysis. RC10.3. Since BNs are explicitly designed to capture uncertainties, I wonder why this approach was chosen here. AA10.3. A BN can be a compact representation of a high-dimensional probability distribution. In this study, we used an existing BN approach and algorithm (Jäger et al. 2017. Bayesian network approach for coastal risk analysis and decision making. Coastal Engineering, in press) to integrate high-dimensional data from various underlying models in a compact way. As mentioned in previous answers, the main purpose is to explore scenarios (forward prediction) or to gain insight in the main drivers of hazards and impacts (backward prediction). RC10.4. The distribution of hazard at the receptors results from the different location of the receptors, but does not reflect the uncertainty related to the inundation or erosion at a specific object. In a strict sense, I would not judge the resulting distribution to represent probabilities. AA10.4. The most intuitive interpretation of the distribution of hazard and consequence at the receptors is indeed as the "expected fractions of receptors with the single hazard or impact levels". However, they could be interpreted as probability distributions for an arbitrarily selected receptor whose location is known. Nonetheless, we removed the following in the main text to not confuse the reader: Page 2, lines 27 – 29 "This implies [. . .] probabilistic-based analysis of the results." Page 26, line 6 " and their uncertainties"

Section 3.4: RC11. To model the consequences flood damage curves are applied. Those are generallyrelated with huge uncertainties (a wide range of relative damage can be observed forequal water levels), which are again neglegted and not included in

the BN. On top, adamage curve that was derived for river floodind is applied. Since the process of stormsurges is very different from river flooding the applicability should be discussed.In terms of risk levels the values selected for both study sites differ significantly. E.g.medium impact building damage ranges from 0.26 to 0.45 compared to 0.1 - 0.2. Whyare these intervals chosen? AA11. Ideally, damage curves have to be specifically built for local conditions (including associated uncertainty). However, in the study site, such information is not available and, official water management agencies recommend the use of a representative damage curve for flooding analysis. These are the selected curves used in this work (they are properly referenced). Now, we have stressed in the text the motivation and implications of the curves selection and the final risk levels. "The chosen damage curves do not include uncertainties, and are used as recommended by the administration at both study sites. This implies that damage ranges and damage-hazard relations are different at each study site and therefore the final impact levels (from none to high) are also site-specific, since they are calculated from the definition of the corresponding damage curves. This assumption aimed to better communicate results to local stakeholders."

Section 3.6: RC12. To my understanding the probability tables of the BN are constructed by simulating a storm scenario for each combination of states and running the deterministic model chain to receive a predicted hazard value for each receptor in the study area. Due to the deterministic character of the model chain, the resulting distribution for the hazard variables does not represent probabilities, but the expected fraction of receptors with the single hazard levels or impact levels respectively. See answers to comments 9 and 10.

RC13. Since no uncertainties are considered, I see no need to apply BNs in this context. The same calculations can be done by applying the model chain directly. A direct application of the model chain would also avoid the discretization of the variables and consequently achieve a higher accuracy. In my point of view the revised paper should either do without the BN approach or account for the uncertainties related to

the single model components. AA13.Uncertainties due to variability inside each bin combination are included in the assessment. Is not the aim of this application exercise to account for the uncertainties of the single model components. There is no such thing as "applying the model chain directly". Some sort of post-processing and integrating is always necessary. The use of the BN is justified by the need to integrate results from multiple simulations when assessing scenarios. If you discretized hazards and impacts according to the vulnerability curves, the only loss of "accuracy" is due to the spatial discretization. Then one could argue that it is not very useful to report the individual hazard level of every single receptor, but that an aggregation into "fractions at different hazard levels per area" is needed to convey insight to decision makers. The method can then further assess other uncertainties (related to lack of knowledge), such as knowing the distribution of Hs to be assessed but not knowing the associated durations. In this context the user can leave the duration unconstrained to integrate to results from all possible durations in the output. Additionally, we are not proposing the BN tool in this article, we are simply using it, (partly) because it is already available. That is an algorithm that automatically integrates and post-processes the model data is already available. The BN is also really useful since allows the user to gain insight in the main drivers of hazards and impacts (backward prediction). We will include on the discussion that indeed "Uncertainties related to individual process oriented models or damage curves are not included in this application. However, the methodology can easily integrate them if simulations from multiple models are used to feed the BN and uncertainties related to damage curves are known".

Section 4: I do not understand which storm intensity is considered here? Are the presented results the joint distribution for all possible combinations of storm characteristics? If so, what is the meaning? Is this a kind of average storm? <- I don't think so. I would rather prefer to consider specific storm scenarios in combination with their return period. E.g. what are the effects of DRR measures for a once-in-a-year oronce-in-10-years event or for an extreme event. To judge the efficiency of DRR measures, it would also be interesting to get some information about the costs of their implementation and

their probability of failure. M14. As explained before, uniform distribution of inputs is presented here as a simple application without prior statistical knowledge at the sites. It shows that a successful comparison of scenarios can be performed without "prior" statistical analysis. In fact, many kinds of results could be shown: real distributions of inputs, selected events, either equal to known historical ones or related to given return periods etc. However, they were not included because this requires prior analyses. For instance: What means a once-in-10-years event? An event calculated in terms of Hs and leaving the other variables (Tp, duration, direction, water level) free? An event defined in terms of the (multiple) joint probability of Hs-duration-direction and water level? (this needs further statistical analysis which is out of the scope of the present work) The aim of the presented application is to assess the efficiency of measures in terms of impact reduction, not to select the best alternative (as e.g. based on multicriteria (MCA) analysis including economics, endurance, ecological, stakeholders' perception, etc). The BN provides an output that can be combined with other information in such MCA analyses (e.g. Barquet and Cumiskey, 2017. Using participatory Multi-Criteria Assessments for assessing disaster risk reduction measures. Coastal Engineering, in press). To clarify this point we have included a paragraph illustrating these options provided a previous statistical analysis is performed, but stressing that the aim of the work is to show a selected application of the tool: "Uniform distribution of input Hs and duration is presented here, as a simple application assuming that there isn't any prior knowledge of the variable distributions. This is shown to be sufficient for the purpose of exploring scenarios". Later in the discussion section we will add "Different kinds of results could be shown with the BN: real distributions of inputs, selected events, either equal to known historical ones or related to given return periods etc. However, they were not included because this requires prior analyses. The aim of the presented application is to assess the efficiency of measures in terms of impact reduction, not to select the best alternative (as e.g. based on multicriteria (MCA) analysis including economics, endurance, ecological, stakeholders' perception, etc). The BN provides an output that can be combined with other information in such MCA analyses (e.g. Barquet and Cumiskey, 2017). Section 5: RC15. page 21, line 11: "a first test to check the method was presented" <- Where?I could not find any validation of the presented model. There is only a reference to a(not published) paper to validate the hazard component of the model chain. This is similar to comment 3 and the answer to it is the same. The sentence is not referring to models validation. We intended to state that the presented work is a simple application of the BN tool to explore its capabilities. It is a general statement that will be rephrased and moved one paragraph above, since it's common for both study sites.

RC16. page 21, line 11-15: A more detailed justification for the chosen amount of intervals and the interval boundaries, would be nice. Additionally, some information about how the computational costs scale with the number of intervals could be provided. Several uncertainties related to the study are not discussed (see comments about section 3). This is similar to comment 4 and answer is also applicable here. To clarify this point we have included the following text "In this work, a first test to check the method was presented, and a balance between computational expense and accuracy was pursued. Therefore, chosen source variables were limited to those defining most important storm features related to impacts and variable discretization was performed with equal intervals covering the whole range of so far observed values. 12 state combinations in the current situation implies 96 total simulations of the model train, which can take even months running in a 48-thread cluster." About the uncertainties, the overall additions performed in all sections, including discussion clarifies which uncertainties are not included in the assessment and how the BN approach is here used. #### technical corrections RC17. page 4, line 24: 2-3m? AA17. It has been rephrased to "The coast is about 130 km long and characterized by low-lying, predominantly dissipative sandy beaches. The coastal corridor has low elevations, mainly ranging from -2 to 3m above MSL (Regione Emilia-Romagna, 2010)" RC18. page 5, line 16-17: Armaroli et al (2012) is cited double AA18. Thanks. It has been addressed. RC19. page 11, line 19: check >0.05m or >0.5m AA19. It has been rephrased to "Erosion was considered significant (and thus, present) when

>0.05m. The erosion risk categories for each receptor were set as follows: (i) Safe: no erosion in any buffer, (ii) Potential Damage: when erosion is present in the 10-m buffer and/or is present with values less than 0.5 m in the footprint buffer, and (iii) Damage: when the erosion limit of 0.5 m is exceeded for the footprint buffer" RC20. page 14, line 31: "it also provided ..." What is "it"? AA20. It has been rephrased to "Alongside the generic structure, a c++ programme that automatically creates the BN (https://github.com/openearth/coastal-dss) is also provided"

Please also note the supplement to this comment:
https://www.nat-hazards-earth-syst-sci-discuss.net/nhess-2017-345/nhess-2017-345-AC1-supplement.pdf

---

## Referee Comment (RC2) · Anonymous Referee #2 · 14 Dec 2017

**Review of paper by Sanuy et al.: "Linking source with consequences of coastal storm impacts for climate change and risk reduction scenarios for Mediterranean sandy beaches"**

**Introduction**
This paper presents an approach to integrated risk assessment for coastal areas with regard to storm impact on beaches (*i.e.*, flooding and erosion), considering climate change. Two case studies are presented from the Mediterranean Sea, one from the northern Spanish coast and one from the Italian coast in the northwest part of the Adriatic Sea. The methodology employed involves simulation with deterministic models for a fixed number of storm scenarios, subsequently being generalized to involve a probabilistic approach using Bayesian statistics.

**Overall Assessment**
The paper presents an interesting and potentially useful methodology for estimating the risk associated with storm impact in coastal areas. It is in general clearly and well written; however, the paper is rather long and "wordy", presenting a lot of detailed information not really needed. On the other hand, certain aspects of the study should be discussed and explained more.

In summary, the following weaknesses of the paper should be addressed: (1) reduce the length of the paper by eliminating detailed results from the study sites; (2) expand the discussion on how coastal managers may use the results of the proposed risk assessment in their work; (3) motivate the selection of models in the approach; (4) discuss the importance of other factors influencing long-term coastal evolution not considered in the approach; (5) clarify the discussion of the methodology and concepts used; (6) comment upon the effects of antecedent morphology and chronology of forcing; and (7) explain the description of beach response to sea level rise.

I recommend that the paper is accepted after major revisions.

The general comments are given in more detail below followed by comments to specific points in the paper.

**General comments**
The authors are requested to address the following comments of a more general nature:

1. **Reduce the length of the paper by eliminating detailed results from the study sites.** The paper is rather long and could be shortened by cutting some of the detailed results from the two study areas. The results from these areas are interesting mainly as an illustration of what the methodology can produce; the specific values are of little interest to the readers in general. Thus, many of the figures 10-18 can be eliminated without loss of information.

2. **Expand the discussion on how coastal managers may use the results of the proposed risk assessment in their work.** The discussion section is very good and informative, indicating strength and weaknesses of the methodology. However, I would like to see the authors present more of their thoughts on how managers can use the results coming out of the proposed risk assessment and advantages compared to how things are done presently. Also, are coastal managers ready to grasp this type of information, especially when it involves probabilistic concepts? In the end risk levels are presented in a qualitative manner through different categories. Would it be possible to be more quantitative?

3. **Motivate the selection of models in the approach.** The basis of the methodology is deterministic simulations that are employed in a probabilistic approach through the Bayesian model. What was the reasoning when selecting the present deterministic models, which are rather detailed and time-consuming to run? Could simpler models have been employed for which many more simulations could have been made? How was the balance selected between the deterministic and probabilistic parts of the approach?

4. **Discuss the importance of other factors influencing long-term coastal evolution not considered in the approach.** The approach focuses on the impact of storms, specifically flooding and erosion. However, storms are only one of the many factors controlling beach evolution. On some coasts storms will be the primary drivers of beach change, but quite often other processes, such longshore transport gradients, sediment input from rivers, and subsidence, must be included to determine how the beach evolves over longer time periods. Typically there is a coupling between longshore and cross-shore processes that needs to be taken into account in estimating beach evolution. Add some discussion.

5. **Clarify the discussion of the methodology and concepts used.** The paper is rather clear on the methodology, but sometimes it is a bit difficult to follow and the sentences become long and affected by jargon. I also have a bit of a problem with how the source-pathway-receptor model is translated to the storm case. The storm is the source and erosion/flooding is the pathway; this seems a bit different (and less logical) from the experience I have in looking at pollution transport. Anyway, may be the writing about and motivation of the schematization could be made a bit clearer. Also, although abbreviations make things a bit easier, if there are too many it is difficult for the reader to remember all of them.

6. **Comment upon the effects of antecedent morphology and chronology of forcing.** Morphological response are very much a function of the antecedent conditions as well as the chronology of the forcing, especially when it comes to storms. For example, if a large storm is followed by a similar large storm the second one will cause much less erosion. Thus, looking a storm impact as individual events will cause some limitations in terms of the impact assessment. Please add some discussion on this.

7. **Explain the description of beach response to sea level rise.** The response of a profile to sea level rise requires some assumption about the evolution of different morphological features, for example the dune (e.g., will the dune grow to its pre-SLR shape?). Some additional discussion on the assumptions made in this respect would be interesting.

**Specific comments**

In the following specific comments are given to the paper (L = Line number; P = page number).

P4, L14
"wave-induced run-up" Includes wave setup? Any consideration of duration with regard to having water at a certain location?

P5, L19
"thresholds" How sensitive are the methods to the selected thresholds? Were this selection based on impact or purely on the forcing properties (offshore wave conditions)? The probability of extreme events with regard to the former and the latter are typically different.

P8, L18
"XBeach model" How good was the calibration/validation?

P9, L14
"intersecting" Meaning in this context?

P11, L16
"footprint" What is this?

P12, L14
"a directional change" But the wind did not change, right (L3)? What is causing this.

P13, L14
"winter dune" What is this?

P18, L18
From here on some of the figure numbers are wrong. Please check.

P18
Some of the DRR measures taken seem to increase the risk. What is the explanation/logic behind this? Does it mean that the characterization of impact is not proper?

---

## Author Comment (AC2) · 12 Jan 2018

**Answer to #2 referee's review of paper by Sanuy et al.: "Linking source with consequences of coastal storm impacts for climate change and risk reduction scenarios for Mediterranean sandy beaches"**

**Introduction**
This paper presents an approach to integrated risk assessment for coastal areas with regard to storm impact on beaches (*i.e.*, flooding and erosion), considering climate change. Two case studies are presented from the Mediterranean Sea, one from the northern Spanish coast and one from the Italian coast in the northwest part of the Adriatic Sea. The methodology employed involves simulation with deterministic models for a fixed number of storm scenarios, subsequently being generalized to involve a probabilistic approach using Bayesian statistics.

**Overall Assessment**
The paper presents an interesting and potentially useful methodology for estimating the risk associated with storm impact in coastal areas. It is in general clearly and well written; however, the paper is rather long and "wordy", presenting a lot of detailed information not really needed. On the other hand, certain aspects of the study should be discussed and explained more.

In summary, the following weaknesses of the paper should be addressed: (1) reduce the length of the paper by eliminating detailed results from the study sites; (2) expand the discussion on how coastal managers may use the results of the proposed risk assessment in their work; (3) motivate the selection of models in the approach; (4) discuss the importance of other factors influencing long-term coastal evolution not considered in the approach; (5) clarify the discussion of the methodology and concepts used; (6) comment upon the effects of antecedent morphology and chronology of forcing; and (7) explain the description of beach response to sea level rise.

I recommend that the paper is accepted after major revisions.

The general comments are given in more detail below followed by comments to specific points in the paper.

AA. The authors thank the referee's effort and comments. We will address all of them, giving a specific response for each of them in what follows.

**General comments**

The authors are requested to address the following comments of a more general nature:

**1. Reduce the length of the paper by eliminating detailed results from the study sites.**
The paper is rather long and could be shortened by cutting some of the detailed results from the two study areas. The results from these areas are interesting mainly as an illustration of what the methodology can produce; the specific values are of little interest to the readers in general. Thus, many of the figures 10-18 can be eliminated without loss of information.

AA1. We agree that the manuscript is "wordy" manly due to details provided about results of case studies. Following reviewer's suggestion, we shall reduce the length of the paper. Thus, we propose eliminating Figures 12, 13 and 14 from the Tordera Delta results and Figure 16 from Lido degli Estensi-Spina results. Thus, the inundation assessment will be presented for the campsite receptors and the erosion assessment will be presented for the infrastructural receptors for the Tordera Delta, whereas concessions will be the only receptor presented in the results section for the Italian case. With this we can still show the same trends between current a future situation and also show the performance of the DRR.

In addition, the text will be cut down accordingly, and also, no mention to specific values will be made, explaining only qualitatively the trends, and leaving the figures as elements where the reader can roughly check the values. With this we are taking out 4 of the 9 figures and corresponding associated text that this section has in the current version of the manuscript.

2. **Expand the discussion on how coastal managers may use the results of the proposed risk assessment in their work.** The discussion section is very good and informative, indicating strength and weaknesses of the methodology. However, I would like to see the authors present more of their thoughts on how managers can use the results coming out of the proposed risk assessment and advantages compared to how things are done presently. Also, are coastal managers ready to grasp this type of information, especially when it involves probabilistic concepts? In the end risk levels are presented in a qualitative manner through different categories. Would it be possible to be more quantitative?

AA2. We propose adding the following sentence in the discussion section: "*The aim of the presented application is to assess the efficiency of measures in terms of impact reduction, not to select the best alternative (as e.g. based on multicriteria (MCA) analysis including economics, endurance, ecological, stakeholders' perception, etc). The BN provides an output that can be combined with other information in such MCA analyses (e.g. Barquet and Cumiskey, 2017).*" The reader is referred to the Barquet and Cumiskey, 2017 paper (Barquet and Cumiskey, 2017. Using participatory Multi-Criteria Assessments for assessing disaster risk reduction measures. Coastal Engineering, in press) where it is detailed how the results of the BN are used to inform stakeholders about the scenarios and then, and MCA participative assessment takes places. The advantages and feedbacks from the MCA analysis are also presented in that paper.

About the last question "would it be possible to be more quantitative?" The answer is yes: in the methodology section, the reader can see how relative damage is the actual output from the BN for the inundation hazard. Thus, post process with quantitative results could be performed (e.g. economic impact estimation derived from relative damage), but this was out of the scope of the present work. We are presenting results as they were showed to the stakeholders, qualitatively and easy to read and assess efficiencies. This is the basis of the MCA analysis explained in Barquet and Cumiskey (2017) where the addition of other information and the participation of multiple stakeholders is key to finally obtain a DRR selection.

3. **Motivate the selection of models in the approach.** The basis of the methodology is deterministic simulations that are employed in a probabilistic approach through the Bayesian model. What was the reasoning when selecting the present deterministic models, which are rather detailed and time-consuming to run? Could simpler models have been employed for which many more simulations could have been made? How was the balance selected between the deterministic and probabilistic parts of the approach?

AA3. In principle any model can be used but results will be as good as accurate the model will be. With this in mind, the model selection is the result of the balance between accuracy and cost. Since the evaluation system is not designed to run on-line (as an Early Warning System would do) computation time is not a major issue. Due to this, we have selected a process-oriented model specifically designed to simulate coastal storm-induced processes which is able to provide integrated information on inundation and erosion, the Xbeach model. At present it is becoming the S-O-A model on coastal systems. However, the proposed framework can work with different (simpler) models provided they are able to

simulate the target processes (inundation and erosion). The motivation of model selection will be stressed using what's formerly explained, in the first paragraph of section 3.3

With independence of the model to be used, it provides the deterministic response of the system. The probabilistic character is provided by the forcing (i.e. storms). The BN works as a result integration and post-processing tool. The balance between deterministic and probabilistic will depend on the information available at the study site and the way the BN is feed. It will be stressed, as this was one of the main focus of referee#1 comments, that the present application is rather deterministic, but has the capability of dealing with a fully probabilistic feeding (see answers AA9, AA10 and AA12 to referee#1 on this topic). Therefore, more insight on this will be given in the general part of the results section (small note) and in the discussion section.

4. **Discuss the importance of other factors influencing long-term coastal evolution not considered in the approach.** The approach focuses on the impact of storms, specifically flooding and erosion. However, storms are only one of the many factors controlling beach evolution. On some coasts storms will be the primary drivers of beach change, but quite often other processes, such longshore transport gradients, sediment input from rivers, and subsidence, must be included to determine how the beach evolves over longer time periods. Typically there is a coupling between longshore and cross-shore processes that needs to be taken into account in estimating beach evolution. Add some discussion.

AA4. The reviewer is also right. However, it has to be considered that the framework here presented is designed to analyse storm-induced coastal response. In spite of that, other factors affecting coastal response can also be added to the framework as we have done with SLR. As an example, the existence of a gradient in longshore transport will induce a background erosion which will modify the morphology of the coast where the storm will impact. This will be the case with any additional processes acting on a system. It must be considered that the presented framework is not forecasting the coastal morphology at any given time (where it should be necessary to couple all processes) but it predicts the expected storm-induced changes for a given coastal configuration. In that sense, a long/medium term model could be used to forecast a future coastal morphology under a given climate scenario and then, use it as initial configuration to forecast storm-induced changes. In any case we shall include a paragraph stating that the initial configuration will be controlled by medium/long term processes and that they must be taken into account to produce initial coastal configurations when the framework is used for future risk predictions. This extra information could be included in the BN, which will automatically output the uncertainties due to future positions of the shoreline and/or the corresponding differences between scenarios.

5. **Clarify the discussion of the methodology and concepts used.** The paper is rather clear on the methodology, but sometimes it is a bit difficult to follow and the sentences become long and affected by jargon. I also have a bit of a problem with how the source-pathway-receptor model is translated to the storm case. The storm is the source and erosion/flooding is the pathway; this seems a bit different (and less logical) from the experience I have in looking at pollution transport. Anyway, may be the writing about and motivation of the schematization could be made a bit clearer. Also, although abbreviations make things a bit easier, if there are too many it is difficult for the reader to remember all of them.

AA5. If we state that erosion and flooding happen IN the pathway it will match the use of the concept more commonly experienced in other fields. We will address this in the schematization methodological section. In addition, as we also got the same comment from

referee#1, we will only leave common abbreviations such as MSL, Hs, Tp, and we will use full wording for all other concepts.

6. **Comment upon the effects of antecedent morphology and chronology of forcing.** Morphological response are very much a function of the antecedent conditions as well as the chronology of the forcing, especially when it comes to storms. For example, if a large storm is followed by a similar large storm the second one will cause much less erosion. Thus, looking a storm impact as individual events will cause some limitations in terms of the impact assessment. Please add some discussion on this.

AA6. The authors agree with the statement about the chronology of forcing and consecutive storms. This is another process controlling initial coastal configuration (morphology) where the storms will impact. See answer to question 4 (AA4).

7. **Explain the description of beach response to sea level rise.** The response of a profile to sea level rise requires some assumption about the evolution of different morphological features, for example the dune (e.g., will the dune grow to its pre-SLR shape?). Some additional discussion on the assumptions made in this respect would be interesting.

AA7. The current state of the description of the application of the Brunn rule is "*This was accomplished assuming an equilibrium coastal profile response following the Bruun rule (Bruun, 1962), resulting in landward and upward displacement of the beach profile*" (P12 L11).
We will specify here that dunes preserve the pre-SLR shape when there's enough accommodation space, and the shape was cut where there wasn't enough space. Then in the discussion we will add a note saying that this assumption has some uncertainties that are assumed to not be significantly affecting the goal of the work (first check on DRR performance on present and future conditions), but should be addressed if the goal was a detailed future scenario impact assessment. In that case, other models could be tested and integrated in the BN to estimate the role of that uncertainty on the outcomes (see e.g. Le Cozannet G, Garcin M, Yates M, Idier D, Meyssignac B (2014) Approaches to evaluate the recent impacts of sea-level rise on shoreline changes. Earth Sci Rev, 138: 47-60. doi:10.1016/j.earscirev.2014.08.005). In addition, SLR projections itself have uncertainties that may be larger than their associated morphological response. All this together could be analysed using the presented BN schematic approach but was out of the scope of the present application. This will be also stressed when we state that the BN has enough flexibility to integrate many different kinds of assessments depending on the scope.

**Specific comments**

In the following specific comments are given to the paper (L = Line number; P = page number).

P4, L14
"wave-induced run-up" Includes wave setup? Any consideration of duration with regard to having water at a certain location?

AA. It is a general statement comparing the contribution of run-up (including set-up and swash) to the total water level (astronomical tide + residual (surge) + run-up). During storms, the contribution of the first two is low (0.25 cm astronomical tide and 0.2-0.6 surge) compared to the contribution of wave-induced run-up (in the order of several meters

depending of the beach slope and period). We are not considering any time duration here, since tide and surge are never the direct cause of flooding in the NW Mediterranean, being this the wave-by-wave overtopping. Thus, the surge only plays the role of "lowering the freeboard of the beach some centimetres (tens in the worst case)", and was considered not to be significant enough compared to waves' contribution to include it as a variable in the BN (i.e. having multiple classes of MSL for the current situation).

P5, L19
"thresholds" How sensitive are the methods to the selected thresholds? Were this selection based on impact or purely on the forcing properties (offshore wave conditions)? The probability of extreme events with regard to the former and the latter are typically different.

AA. The authors are well aware of the different statistical results obtained by the event approach (selection based on storm characteristics) or the response approach (statistics based on impacts/hazards) and have studied its effect of inundation hazard statistical identification. The presented method is not sensitive to the thresholds to identify events the way it is applied. We are not assigning probabilities or return periods to a given inundation. We are integrating results from multiple scenarios by equally representing them (same storm simulations for each source characteristic state of variables). In this section we are only explaining how storm events are usually identified in the study sites, for the reader to know what a storm means in the Adriatic or in the NW Mediterranean.
In addition, obtaining storms by thresholds on impacts, would imply having available time-series data of multiple years of storms, which is not the case in the Italian study site and may not be the case in other locations.

P8, L18
"XBeach model" How good was the calibration/validation?

AA. Currently we are providing only the reference to the study at the Tordera Delta were the validation is explained in detail (currently submitted to Coastal Engineering). We will include a note at each study site paragraph describing briefly how good the validation was, in terms of BSS score or qualitative comparison respectively, so the reader can have some additional info in the present manuscript and not only the references.

P9, L14
"intersecting" Meaning in this context?

AA. Polygon intersection, between receptors 2D layout and the Xbeach grid. This way we identify which grid nodes affect each receptor. We will clarify

P11, L16
"footprint" What is this?

AA. Is the receptor polygon layout in 2D. We will use "receptor limits in the ground".

P12, L14
"a directional change" But the wind did not change, right (L3)? What is causing this.

AA. The study reporting change in wave direction (Casas-Prat and Sierra, 2012) predicts the change in direction by applying statistics to the current past 60-year evolution of wave records, and obtaining the prediction of the future wave mean climates. Therefore, there is no information/evidence in that study linking that change to any specific forcing (wind,

wave current interaction…). It is a scenario we wanted to explore as a "what if" future situation.

P13, L14
"winter dune" What is this?

AA. It's the name they give to a trapezoidal sandy dune, that they artificially build from beach scrapping every winter to protect concessions. We'll add clarification

P18, L18
From here on some of the figure numbers are wrong. Please check.

AA. This will be addressed, since some figures will be supressed and the whole text on the results section reviewed and reduced.

P18
Some of the DRR measures taken seem to increase the risk. What is the explanation/logic behind this? Does it mean that the characterization of impact is not proper?

AA. This is the case when the DRR affects the hydrodynamics at the nearshore and or swash zone, and while protecting locally some receptors, but the erosion is increased down coast and other receptors get more exposed than before. Overall it can be observed how this increase isn't significant in any case.
Nevertheless, the figure containing this effect will be supressed due to comment 1 (AA1), and thus, it will not induce confusion to the reader or require further explanation by the authors.

---

## Author Response (AR1)

**Answer to Editor's comments after first review**

**Editor Comments**

**AA: Authors' Answers**

Dear authors,

thank you very much for your response to the 2 reviews. As you have seen, both referees find your work valuable, however, both also have a number of comments and concerns. In particular, reviewer #1 questions the use of BNs in the context of your study. I feel that this is a fundamental point, as he/she argues to have experience with BNs, but still does not see its justification.

I decided on 'major revision', but my understanding is that this fundamental issue needs to be completely resolved before the work could be considered for publication. Major concerns about the appropriateness of the applied method would lead to rejection of the paper. Related to this fundamental issue is the question of the scientific contribution of the paper which was not completely clear to me.

One formal issue: Could you please clearly mark your answers to the comments in the response letter? For example, for some comments, your response follows directly to the comment without sign (AA).

Best regards, Bruno Merz

Dear Editor,

In what follow we address all questions and concerns raised by reviewers. In all the cases we include the original reviewer's comment after the sign (RC) followed by our response after the sign (AA).

The main change in the manuscript is a reduction of the Results section and the integration of the Discussion in a single section without subsection for the study sites. Nonetheless, the whole document has been thoroughly reviewed and edited to answer to the requirements and suggestions of the reviewers, enhancing a clearer interpretation of our work.

**Answer to #1 referee's review**

RC0: #### general comments

The authors present a risk assessment for coastal storm impacts to support decisions on disaster risk reductions. For that purpose, a Bayesian network approach is used to link process-oriented models, that predict the hazards at the receptors, with vulnerability relations to obtain the final expected impact. In a case study two Mediterranean sandy coasts are considered.

The paper is well structured and provides a well-argued motivation and problem definition. Further the study areas are described in detail and underline the relevance of the presented study. Despite a good structure in the Methodology section, some aspects of the method remain unclear to me, which might be due to the complexity of the model chain. This affects especially the Bayesian network (BNs) application. Even though I am familiar with the construction and application of BNs, I have problems to follow the construction (i.e. parameter setting) of the BNs and to understand the motivation for and advantages of using BNs in the presented context. The results section provides an extensive analysis of different climate change and adaptation scenarios for the considered Mediterranean coasts. Yet, I did not understand which storm intensities are considered here (this might be due to a missing understanding of the methodology).

The discussion names several aspects that pose challenges or are neglected in the presented model approach and might consequently be taggled in follow up studies. Yet, to my impression important critical points of the presented approach are missing, as will be specified in the specific comments.

AA: Thanks for comments and suggestions on submitted manuscript. In what follows we answer to all comments/suggestions/questions raised by the reviewer.

RC1: #### specific comments

I found it quite difficult to keep track of all abbreviations used in the paper.

AA: We understand that the use of a large number of abbreviations could be cumbersome, especially for a long text as this manuscript is. We have reviewed the text and reduced the number of abbreviations to a minimum.

abstract:

RC2: line 15: "a large number of storm characteristics" What is a large number? To my understanding 3-4 storm characteristics were considered.

AA: When we mention "storm characteristics" we refer to storms defined in terms of a combination of Hs, duration, direction and water level. In each site, we have selected 12 storms for each sea level rise scenario, i.e. 24 storm combinations per site. This is later specified in the text (including table 1). Additionally, each storm combination is represented by 2 different simulations where storm variables are slightly changed within the range corresponding to its category. We have modified this sentence in the abstract by the following "*Process-oriented models are used to predict hazards at the receptor scale which are converted into impacts through vulnerability relations. In each site, a total of 48 storms have been simulated under different scenarios and obtained results are integrated by using a Bayesian Network to link forcing characteristics with expected impacts through conditional probabilities.*"

RC3: line 17: "The tool has been proven successful in reproducing current coastal responses at both sites". I could not find any model verification in the paper. Only a reference to a paper that verifies a part of the model.

AA. In order to avoid confusion with morphodynamic model validation we have changed the sentence to "*Consultations with local stakeholders and experts have shown that the tool is valuable for communicating risks and the effects of risk reduction strategies. The tool can therefore be valuable support for coastal decision making.*"

Section 3.2:

RC4: page 7, line 7: The discretization of the variables is hardly motivated. What is the motivation for choosing 2 or 3 or n intervals for a certain variable? How are the interval boundaries selected (equidistant, equifrequent, entropy-based, ...)? How is the probable range determined (only so far observed values)? Some information about the distribution of these variables might help to motivate the discretization. What are the effects of discretizing? In the discussion it is mentioned that accuracy comes at computational costs, but this information is quite sparse (no information about number of intervals scales with computational costs or what

are the computational costs of the current network for parameter determination and for inference).

AA: To avoid confusion in this section we remove *"Storm scenarios are defined […] were selected for use in the analysis."* (lines.6-9), since we already explain that storm scenarios are combinations of variable values covering the typical storm condition at each study site. The description of bins only makes sense in terms of the BN, which we have not explained yet in this section. Then we will motivate the discretisation of variables in section 3.6.1.

Thus, we have added in section 3.6.1 the following text: *"The bin ranges for variables characterising boundary conditions is selected to be equidistant covering the observed values at each study site (Table 1). Additional non-observed ranges are introduced to account for SLR. The used number of intervals is a compromise between accuracy and computational effort. Each combination showed in Table 1 has been simulated twice to account for potential variability inside bins. Then, all simulations are repeated for DRR scenarios affecting hazards (i.e. Winter Dune and Nourishment + Dune). Therefore, a total number of 96 model runs were required for the applied bin set-up. As a reference, using parallel simulations with 48 threads, the ratio computation time over real storm time was ~0.2, meaning that a 40 hr storm takes ~8 hours of simulation time."*

RC5: page 7, line 12: "time series" of what?

AA. We have modified the sentence as *"In addition, time series of waves (either bulk Hs, Tp and mean direction or spectrum) and water levels during each storm event were used when this information was available."*.

RC6: page 7, line 19-20: the "(24 simulated storms)" confused me? Why do you consider 24simulated storms for 12 state combinations?

AA. See answers to comments 2 and 4. In addition to avoid confusion, the sentence is rephrased to avoid the brackets.

The new sentence is *"The selected bins for each variable can be seen in Table 1. These lead to 12 combinations defining the source under current MSL and 12 under future MSL (given by a*

*SLR scenario). Each combination of states is simulated twice by means of slightly different storms to account for potential variability within variable ranges, leading to a total of 24 storms under the current MSL and 24 under SLR*"

We have also changed Table 1 for better interpretation of variable discretization.

RC7: page 7, line 24: What are synthetic triangular events?

AA. It is the way to reproduce storm events not previously recorded to be used in the numerical model. To clarify this, we have included this text:

"*To include the full range of cases, the remaining eight storms were completed by using combinations of Hs-duration-direction not previously recorded. These events were modelled assuming they follow a triangular-shaped evolution with the peak intensity at the half of their duration (see e.g. McCall et al. 2010; Poelhekke et al., 2016).*"

RC8: page 7, line 27-28: "water level and Hs are uncorrelated" <- a reference is needed?

AA. The reference is (Mendoza and Jiménez, 2008) which was located at the end of the next sentence. We have rephrased to avoid confusion.

RC9: page 8, line 1: How are the driving variables identified? Why are the remaining variables considered to have no effects? How is the distribution of the storm defining variables defined? To my current understanding an equal amount of storms for each state combination is considered, which infers a uniform distribution of the variables. Yet, I would expect that small Hs values or smaller durations are more likely than higher values? Is this accounted for?

AA: Considered driving variables in the analysis have been selected taking into account storm characteristics at each site. Thus, in the Tordera Delta case, Tp does not significantly vary during storms so, we don't consider it a variable to be discretized. Moreover, storm surges play a secondary role in comparison with wave contribution to total water level. This now justified in the text with specific references describing storm conditions in the area: *"For the Tordera Delta case, the selected variables to define storm scenarios were Hs at the peak of the storm, total storm duration, and incoming storm direction. Tp does not significantly vary during storms in*

*the study area (see Mendoza et al., 2011) and was not included as a characteristic variable. Due to the coastline configuration and morphology, the area is sensitive to storm incoming direction (Sanuy and Jiménez, n.d.). Thus, the main directions in terms of dominant (E) and secondary (S) storms needed to be considered separately. Finally, the position of the mean sea level (MSL) during the event was included to reproduce hypothetical future projections of sea level rise (SLR) due to climate change".*

This is also completed for the Italian case as:

*"Previous works in the area of the Lido degli Estensi-Spina case study have identified the dominant role of wave height and total water level in controlling the magnitude of storm-induced erosion and inundation (Armaroli et al 2009, 2012). Due to this, variables used to characterize the source were the maximum Hs and maximum TWL (surge+tide) during each storm event. Thus, wave period and the direction of the storms was not considered as a source characteristic variable to be discretized. Each storm was simulated for current and climate change (SLR) scenarios. Finally, and similarly to the Tordera case study, each Hs-TWL combination was simulated twice to account for potential variability"*

With respect to the distribution of the storm defining variables, it is true that the BN is trained with equal representation of all variable combinations. This implies that once the Bayesian is trained, the "prior" probabilities of storm variables are uniform. We have followed this approach because extreme value probability distributions of source variables are not known for the two sites and estimating them was beyond the scope of this study. In spite of this, once relative frequencies of different events are available, source nodes could be re-trained which will result in an automatic update of the distribution of all hazard and consequence nodes. Moreover, the user could also test how different assumptions on the source variable distributions would change the hazard and impact estimates. Nonetheless, the main strength of the BN at its current stage is that it enables decision makers to explore different scenarios and helps them to design robust strategies (i.e., strategies that are successful under most scenarios.). We have covered this point in Results and Discussion sections. See answers on comments on those sections.

Section 3.3:

RC10: Only one event (storm) is considered for each combination of states. Yet, similar events might result in different outcomes. Further, the applied model chain seems to provide

deterministic results. Consequently, no uncertainties are considered/captured in the model construction. Since BNs are explicitly designed to capture uncertainties, I wonder why this approach was chosen here.

The distribution of hazard at the receptors results from the different location of the receptors, but does not reflect the uncertainty related to the inundation or erosion at a specific object. In a strict sense, I would not judge the resulting distribution to represent probabilities.

AA: We answer the comment in splitting it in different shorter pieces:

RC10.1: Only one event (storm) is considered for each combination of states. Yet, similar events might result in different outcomes.

AA. Two storms are simulated for each combination (we agree that this was not properly explained, see previous answers). Furthermore, the user could select storms belonging to for example Hs = 3-4 meters with waves coming from the East for current MSL but leaving the duration unconstrained as an uncertain variable. In such a case the obtained output would be the integrated result from 4 simulations (2 direction categories that are represented by 2 simulations each with different values of duration). Therefore, results will account for the uncertainty on duration for a given (certain) Hs. In practice, this could be relevant because storm forecasts could contain more certainty on some variables than on others, for example as a result of ensemble forecasts. For a more detailed discussion how the BN tool can deal with ensemble forecasts also see section 5 in Jäger et al. 2017. A Bayesian network approach for coastal risk analysis and decision making. Coastal Engineering (in press, 10.1016/j.coastaleng.2017.05.004).

RC10.2. Further, the applied model chain seems to provide deterministic results. Consequently, no uncertainties are considered/captured in the model construction.

AA. Reviewer is right, we do not account for uncertainties inherent to individual models. Quantifying uncertainties of individual models is another study in itself (e.g. Wagenaar et al. 2016. Uncertainty in flood damage estimates and its potential effect on investment decisions. NHESS, 16, 1-14), and it is beyond the scope of this manuscript to do such analysis.

RC10.3. Since BNs are explicitely designed to capture uncertainties, I wonder why this approach was chosen here.

AA. A BN can be a compact representation of a high-dimensional probability distribution. In this study, we used an existing BN approach and algorithm (Jäger et al. 2017.. Coastal Engineering, 10.1016/j.coastaleng.2017.05.004) to integrate high-dimensional data from various underlying

models in a compact way. As mentioned in previous answers, the main purpose is to explore scenarios (forward prediction) or to gain insight in the main drivers of hazards and impacts (backward prediction).

RC10.4. The distribution of hazard at the receptors results from the different location of the receptors, but does not reflect the uncertainty related to the inundation or erosion at a specific object. In a strict sense, I would not judge the resulting distribution to represent probabilities.

AA. The most intuitive interpretation of the distribution of hazards and consequences at the receptors is indeed as the "expected fractions of receptors with the single hazard or impact levels". However, they could be interpreted as probability distributions for an arbitrarily selected receptor whose location is known. Nonetheless, we removed the following in the main text to not confuse the reader:

Page 2, lines 27 – 29 "This implies […] probabilistic-based analysis of the results."

Page 6, line 33: "probabilistic".

Conclusions, last sentence " and their uncertainties"

Section 3.4:

RC11. To model the consequences flood damage curves are applied. Those are generally related with huge uncertainties (a wide range of relative damage can be observed for equal water levels), which are again neglected and not included in the BN. On top, a damage curve that was derived for river flooding is applied. Since the process of storm surges is very different from river flooding the applicability should be discussed. In terms of risk levels, the values selected for both study sites differ significantly. E.g. medium impact building damage ranges from 0.26 to 0.45 compared to 0.1 - 0.2. Why are these intervals chosen?

AA. Ideally, damage curves have to be specifically built for local conditions (including associated uncertainty). However, in the study site, such information is not available and, official water management agencies recommend the use of a representative damage curve for flooding analysis. These are the selected curves used in this work (they are properly referenced). Now, we have stressed in the text the motivation and implications of the curves selection and the final risk levels. The following text has been included.

*"The chosen damage curves do not include uncertainties, and they are used as recommended by the Administration at each study site. This implies that damage ranges and damage-hazard relations are different and therefore, final impact levels (from none to high) are site-specific. This assumption aimed to better communicate results to local stakeholders."*

Section 3.6:

RC12. To my understanding the probability tables of the BN are constructed by simulating a storm scenario for each combination of states and running the deterministic model chain to receive a predicted hazard value for each receptor in the study area. Due to the deterministic character of the model chain, the resulting distribution for the hazard variables does not represent probabilities, but the expected fraction of receptors with the single hazard levels or impact levels respectively.

AA. See answers to comments 9 and 10.

RC13. Since no uncertainties are considered, I see no need to apply BNs in this context.

The same calculations can be done by applying the model chain directly. A direct application of the model chain would also avoid the discretization of the variables and consequently achieve a higher accuracy.

In my point of view, the revised paper should either do without the BN approach or account for the uncertainties related to the single model components.

AA. It is true that not all potential uncertainties are considered, but we disagree with the fact that uncertainties are not considered. Thus, uncertainties due to variability inside each bin combination are included in the assessment. With respect to model-related uncertainty, it was not the aim of this application to account for the uncertainties of single model components. There is no such thing as "applying the model chain directly" since we need to integrate all results for all possible combinations in a usable way, i.e. in a format that is also suitable for coastal managers. In this sense, the use of the BN facilitate the integration of obtained results from multiple simulations when assessing scenarios. If hazards and impacts are discretized according to the vulnerability curves, the main loss of "accuracy" is due to the spatial discretization of the hazard and/or receptors exact location and size/shape. Then it could be argued that it is not very useful

to report the individual hazard level of every single receptor, but that an aggregation into "fractions at different hazard levels per area" is needed to convey insight to decision makers.

The developed BNs can also be used to assess other uncertainties related to lack of knowledge. This would be the case in which the distribution of Hs is known but we have not information about associated durations. In this context, the user can leave the duration unconstrained to integrate the results from all possible durations in the output.

In addition to this, the use of the BN also allows the user to gain insight in the main drivers of hazards and impacts (BN in reverse mode or backward prediction, see results about figures 15 and 16).

With respect to this we have included the following text in the Discussion section

"*Uncertainties associated with the pathway are related to the selection of the process-oriented models used to simulate induced hazards. In the current analysis, we have not considered this source of uncertainty since the framework is applied by using previously selected models and recommended damage curves. As it was mentioned in the method section, the selected model to simulate storm-induced hazards is XBeach (Roevilnk et al. 2009), which is currently one of the most applied at the international level. Applied model setting has been selected for each case study based on local calibrations and validations for selected storm impacts. This step has to be done prior to BN development since it will control the accuracy of estimated hazards intensity and it is also a source of uncertainty. In any case, the methodology can easily deal with this source of uncertainty if simulations from multiple models or model settings are used to feed the BN.*".

Section 4:

RC14. I do not understand which storm intensity is considered here? Are the presented results the joint distribution for all possible combinations of storm characteristics? If so, what is the meaning? Is this a kind of average storm? <- I don't think so. I would rather prefer to consider specific storm scenarios in combination with their return period. E.g. what are the effects of DRR measures for a once-in-a-year oronce-in-10-years event or for an extreme event. To judge the efficiency of DRR measures, it would also be interesting to get some information about the costs of their implementation and their probability of failure.

A.A. The following paragraphs have been included to clarify these points:

At the beginning of the Results section:

"*In this section, the results of scenario testing are provided for each case study through an integrated comparison of percentages of receptors at each level of flooding and erosion risks. This is done by comparing the risk levels under current and climate change scenarios with and without measures. In any case, it has to be taken into account that this assessment does not include the statistical distribution of storm variables. We assume that there is no prior knowledge on their distributions and, as consequence, we simply describe them with a uniform distribution. This approach is adequate to explore scenarios and to assess the efficiency of protection measures in terms of impact reduction.*".

And in the Discussion section:

"*No prior knowledge of storm characteristic variables was assumed, representing them with uniform distributions. This was enough to communicate scenarios and measure efficiencies to stakeholders by integrating the BN in a multicriteria analysis such as in Barquet and Cumiskey (2017). In such multicriteria assessments, BN output is combined with information on additional elements required for decision making such as economics, endurance, ecological, stakeholders' perception, allowing for the final evaluation of alternatives. As it has been mentioned before, the next step should be to reproduce the local maritime climate to analyse this performance taking into account the relative frequency of each condition.*".

Section 5:

RC15. page 21, line 11: "a first test to check the method was presented" <- Where?I could not find any validation of the presented model. There is only a reference to a (not published) paper to validate the hazard component of the model chain.

AA. This is similar to comment RC3 and the answer to it is the same. The sentence was not referring to models validation.

RC16. page 21, line 11-15: A more detailed justification for the chosen amount of intervals and the interval boundaries, would be nice. Additionally, some information about how the computational costs scale with the number of intervals could be provided. Several uncertainties related to the study are not discussed (see comments about section 3).

AA. This is similar to comment RC4 and answer is also applicable here. To clarify this point, we have included the following paragraph in Section 5

"*With respect to the definition of sources, the BN has been built by chosen storm variables limited to those previously identified as the most important to control the magnitude of storm-induced hazards at each site. Once identified, they were discretized in equal intervals covering the whole range of so far observed values. We have used a limited number of combinations to cover the most important storm classes in terms of induced hazards and damages (Armaroli et al., 2009, 2012; Mendoza et al., 2011). Increasing the number of storms will allow to better reproduce the inherent climate variability and to characterize better this source of uncertainty in the assessment. In spite of this, used values can be considered as representative for forcing conditions in both areas and, in this sense, they will allow to use the framework to assess the efficiency of tested measures to reduce inundation and erosion risks for each given conditions*".

We have also included a note on computational effort in section 3.6.1:

"*The used number of intervals is a compromise between accuracy and computational effort. Each combination showed in Table 1 has been simulated twice to account for potential variability inside bins. Then, all simulations are repeated for DRR scenarios affecting hazards (i.e. Winter Dune and Nourishment + Dune). Therefore, a total number of 96 model runs were required for the applied bin set-up. As a reference, using parallel simulations with 48 threads, the ratio computation time over real storm time was ~0.2, meaning that a 40 hr storm takes ~8 hours of simulation time*.".

About the uncertainties, the overall additions performed in all sections, including discussion, clarifies which uncertainties are not included in the assessment and how the BN approach is here used. Additionally, see also answers to referee #2 on comments 4,6 and 7, where we have included more discussion on uncertainty sources.

**technical corrections**

RC17. page 4, line 24: 2-3m?

AA17. It has been rephrased to "*The coast is about 130 km long and characterized by low-lying, predominantly dissipative sandy beaches. The coastal corridor has low elevations, mainly ranging from -2 to 3m above MSL (Regione Emilia-Romagna, 2010)*"

RC18. page 5, line 16-17: Armaroli et al (2012) is cited double

AA18. Thanks. It has been addressed.

RC19. page 11, line 19: check >0.05m or >0.5m

AA19. It has been rephrased to "*Erosion was considered present if >0.05m (vertical) and significant when >0.5m. The erosion risk categories for each receptor were set as follows: (i) Safe: no erosion in any buffer, (ii) Potential Damage: when erosion is present in the 10-m buffer and/or present but not significant in the receptor itself, and (iii) Damage: when the erosion limit of 0.5 m is exceeded within the receptor limits*"

RC20. page 14, line 31: "it also provided ..." What is "it"?

AA20. It has been rephrased to "*Alongside the generic structure, a c++ programme that automatically creates the BN (https://github.com/openearth/coastal-dss) is also provided*"

**Answer to #2 referee's review**

RC. Introduction

This paper presents an approach to integrated risk assessment for coastal areas with regard to storm impact on beaches (*i.e*., flooding and erosion), considering climate change. Two case studies are presented from the Mediterranean Sea, one from the northern Spanish coast and one from the Italian coast in the northwest part of the Adriatic Sea. The methodology employed involves simulation with deterministic models for a fixed number of storm scenarios, subsequently being generalized to involve a probabilistic approach using Bayesian statistics.

RC. Overall Assessment

The paper presents an interesting and potentially useful methodology for estimating the risk associated with storm impact in coastal areas. It is in general clearly and well written; however, the paper is rather long and "wordy", presenting a lot of detailed information not really needed. On the other hand, certain aspects of the study should be discussed and explained more.

In summary, the following weaknesses of the paper should be addressed: (1) reduce the length of the paper by eliminating detailed results from the study sites; (2) expand the discussion on how coastal managers may use the results of the proposed risk assessment in their work; (3) motivate the selection of models in the approach; (4) discuss the importance of other factors influencing long-term coastal evolution not considered in the approach; (5) clarify the discussion of the methodology and concepts used; (6) comment upon the effects of antecedent morphology and chronology of forcing; and (7) explain the description of beach response to sea level rise.

I recommend that the paper is accepted after major revisions.

The general comments are given in more detail below followed by comments to specific points in the paper.

AA. Thanks for comments and suggestions on submitted manuscript. In what follows we answer to all comments/suggestions/questions raised by the reviewer.

RC. General comments

The authors are requested to address the following comments of a more general nature:

RC1. Reduce the length of the paper by eliminating detailed results from the study sites. The paper is rather long and could be shortened by cutting some of the detailed results from the two

study areas. The results from these areas are interesting mainly as an illustration of what the methodology can produce; the specific values are of little interest to the readers in general. Thus, many of the figures 10-18 can be eliminated without loss of information.

AA. We agree that the manuscript is "wordy" manly due to details provided about results of case studies. Following reviewer's suggestion, we have reduced the length of the paper. Thus, we have eliminated Figures 12, 13 and 14 from the Tordera Delta results and Figure 16 from Lido degli Estensi-Spina results (and corresponding pieces of text). With this, we focus on the most relevant receptors for each case and hazard. Thus, in the Tordera case study, inundation and erosion assessments are analysed for campsites and infrastructures respectively. On the other hand, "beach concessions" is the only receptor considered for the Italian case. This does not affect to already observed future trends neither the estimated performance of DRR measures. In addition to the mentioned text cut, we have avoided to provide too much specific values and we have concentrated in characterising the general trends, and leaving the figures as elements where the reader can check obtained values. With this we are taking out 4 of the 9 figures and corresponding text from the original version of the manuscript.

In addition, the whole document was reviewed to avoid repetitions, long sentences and number of acronyms.

RC2. Expand the discussion on how coastal managers may use the results of the proposed risk assessment in their work. The discussion section is very good and informative, indicating strength and weaknesses of the methodology. However, I would like to see the authors present more of their thoughts on how managers can use the results coming out of the proposed risk assessment and advantages compared to how things are done presently. Also, are coastal managers ready to grasp this type of information, especially when it involves probabilistic concepts? In the end risk levels are presented in a qualitative manner through different categories. Would it be possible to be more quantitative?

AA. We have added the following short note in the Introduction:
*"At both study sites, the tested measures were pre-selected taking into account the outcome of interviews to stakeholders (see Martinez et al., 2017) and the obtained results were used in a participatory process to select acceptable measures on the basis of a multicriteria analysis (see Barquet and Cumiskey, 2017)".*

Additionally, we have added the following paragraph in Discussion:

*"The presented work is part of a larger investigatory process (see Martinez et al., 2017) where stakeholders and end-users were interviewed to select possible measures for critical coastal areas (i.e. local scale). The objective of the present work was to provide rather simple information on the efficiency of measures to be used in a participatory process (see Barquet and Cumiskey, 2017) aiming at selecting acceptable measures to be applied as part of an integrated local strategy for risk reduction."*

*[…]*

*"We have used a limited number of combinations to cover the most important storm classes in terms of induced hazards and damages (Armaroli et al., 2009, 2012; Mendoza et al., 2011). Increasing the number of storms will allow to better reproduce the inherent climate variability and to characterize better this source of uncertainty in the assessment. In spite of this, used values can be considered as representative for forcing conditions in both areas and, in this sense, they will allow to use the framework to assess the efficiency of tested measures to reduce inundation and erosion risks for each given conditions. No prior knowledge of storm characteristic variables was assumed, representing them with uniform distributions. This was enough to communicate scenarios and measure efficiencies to stakeholders by integrating the BN in a multicriteria analysis such as in Barquet and Cumiskey (2017). In such multicriteria assessments, BN output is combined with information on additional elements required for decision making such as economics, endurance, ecological, stakeholders' perception, allowing for the final evaluation of alternatives. As it has been mentioned before, the next step should be to reproduce the local maritime climate to analyse this performance taking into account the relative frequency of each condition."*.

About the last question "would it be possible to be more quantitative?" The answer is yes: in the methodology section, the reader can see how relative damage is the actual output from the BN for the inundation hazard. Thus, further analysis with quantitative results could be performed (e.g. economic impact estimation derived from relative damage). However, we are presenting results as they were showed to the stakeholders, in order to easily interpret efficiencies. This is the basis of the MCA analysis explained in Barquet and Cumiskey (2017) where the addition of other information and the participation of many (multiple) stakeholders is key to finally obtain a DRR selection.

RC3. Motivate the selection of models in the approach. The basis of the methodology is deterministic simulations that are employed in a probabilistic approach through the Bayesian model. What was the reasoning when selecting the present deterministic models, which are rather detailed and time-consuming to run? Could simpler models have been employed for which many more simulations could have been made? How was the balance selected between the deterministic and probabilistic parts of the approach?

AA. In principle any model can be used but results will be as good as accurate the model will be. With this in mind, the model selection is the result of the balance between accuracy and cost. Since the model chain is not designed to provided daily forecasting (as an Early Warning System would do) computation time is not a major issue. Due to this, we have selected a process-oriented model specifically designed to simulate coastal storm-induced processes which is able to provide an integrated assessment of inundation and erosion hazards, the Xbeach model, which is one of the best available models to simulate storm-induced morphodynamics. However, the proposed framework can work with different (simpler) models provided they are able to simulate the target processes (inundation and erosion). The motivation of model selection will be stressed using what's formerly explained, in the first paragraph of section 3.3. Thus, the new first paragraph now states: *"To simulate the pathway and obtain hazards of interest, a model chain was designed and adapted for each site (Figure 4, II). Any model can be used within the model chain, and results will be as good as accurate the model. The chain must be able to reproduce all hazards to be assessed (i.e. erosion and inundation). To do this, a detailed 2D process-oriented model designed to simulate coastal storm-induced processes is used, which is able to provide integrated information on inundation and erosion, the Xbeach model (see Roelvink et al., 2009 for model details). At present it is becoming the S-O-A model on coastal systems. However, the proposed framework can work with different (simpler) models provided they are able to simulate the target processes (inundation and erosion). The Xbeach model was used in both study cases."*

And also in the Discussion section:

*"Uncertainties associated with the pathway are related to the selection of the process-oriented models used to simulate induced hazards. In the current analysis, we have not considered this source of uncertainty since the framework is applied by using previously selected models and recommended damage curves. As it was mentioned in the method section, the selected model to simulate storm-induced hazards is Xbeach (Roevilnk et al. 2009), which is currently one of the*

*most applied at the international level. Applied model setting has been selected for each case study based on local calibrations and validations for selected storm impacts. This step has to be done prior to BN development since it will control the accuracy of estimated hazards intensity and it is also a source of uncertainty. In any case, the methodology can easily deal with this source of uncertainty if simulations from multiple models or model settings are used to feed the BN.*"

With independence of the model to be used, it provides the deterministic response of the system. The probabilistic character is provided by the forcing (i.e. storms). The BN works as a result integration and post-processing tool. The balance between deterministic and probabilistic will depend on the information available at the study site and the way the BN is feed. We have addressed this point in answers to comments [9, 10 and 12] of referee#1. More insight on this is given in the general part of the results section (small note) and in the discussion section.

RC4. Discuss the importance of other factors influencing long-term coastal evolution not considered in the approach. The approach focuses on the impact of storms, specifically flooding and erosion. However, storms are only one of the many factors controlling beach evolution. On some coasts storms will be the primary drivers of beach change, but quite often other processes, such longshore transport gradients, sediment input from rivers, and subsidence, must be included to determine how the beach evolves over longer time periods. Typically, there is a coupling between longshore and cross-shore processes that needs to be taken into account in estimating beach evolution. Add some discussion.

AA. The reviewer is also right. However, it has to be considered that the presented framework is designed to analyse storm-induced coastal response. Thus, the presented framework is not forecasting the coastal morphology at any given time (where it should be necessary to couple all processes) but it predicts the expected storm-induced changes for a given coastal configuration. In that sense, a long/medium term model could be used to forecast a future coastal morphology under a given climate scenario and then, use it as initial configuration to forecast storm-induced changes. This was done here with long term coastal response to SLR, where coastal morphology was modified to simulate its effect in Tordera Delta. This could be done externally with any additional processes acting on a system such as the existence of a gradient in the longshore transport which will induce a background erosion. We have included the following paragraph (in Discussion, before 5.1):

*"Another point to be considered is that this assessment framework has just been designed to analyse the storm-induced coastal response. This implies that used models does not forecast the coastal morphology at a given time (where it should be necessary to couple all governing processes) but predict the expected storm-induced changes for a given coastal configuration. As storm-induced hazards depend on existing morphology at the time of the impact (e.g. Cohn and Ruggiero, 2016), the initial morphology used in the model is also a source of uncertainty. To overcome this, a long/medium term morphological model (Hanson et al. 2003;Lesser et al. 2004) could be used to forecast the future coastal morphology under a given climate scenario at a given time and then, to use it as the initial configuration to assess storm-induced changes. This has been illustrated here by considering the change in estimated risks due to sea level rise in Tordera Delta. This approach can also be applied to assess the effects of consecutive storm impacts (Coco et al. 2014) by using estimated post-storm bed levels as pre-storm morphology for given storm combinations. Once this extra information is included in the BN, the uncertainty associated to future shoreline configurations on assessed risks can be analysed."*

RC5. Clarify the discussion of the methodology and concepts used. The paper is rather clear on the methodology, but sometimes it is a bit difficult to follow and the sentences become long and affected by jargon. I also have a bit of a problem with how the source-pathway-receptor model is translated to the storm case. The storm is the source and erosion/flooding is the pathway; this seems a bit different (and less logical) from the experience I have in looking at pollution transport. Anyway, may be the writing about and motivation of the schematization could be made a bit clearer. Also, although abbreviations make things a bit easier, if there are too many it is difficult for the reader to remember all of them.

AA. See PATHWAY concept rephrased to: *"These storms propagate through the pathway, causing erosion at the coast and inundation on the hinterland. Both hazards are the main focus of the analysis"* in section 3.1.
In addition, as we also got the same comment from referee#1 regarding abbreviations. We have reduced abbreviations to a minimum, and we use full wording for most concepts.

RC6. Comment upon the effects of antecedent morphology and chronology of forcing. Morphological response is very much a function of the antecedent conditions as well as the chronology of the forcing, especially when it comes to storms. For example, if a large storm is followed by a similar large storm the second one will cause much less erosion. Thus, looking a

storm impact as individual events will cause some limitations in terms of the impact assessment. Please add some discussion on this.

AA. The authors agree with the statement about the chronology of forcing and consecutive storms. This is another process controlling initial coastal configuration (morphology) where the storms will impact. See answer to comment 4.

RC7. Explain the description of beach response to sea level rise. The response of a profile to sea level rise requires some assumption about the evolution of different morphological features, for example the dune (e.g., will the dune grow to its pre-SLR shape?). Some additional discussion on the assumptions made in this respect would be interesting.

AA. The current state of the description of the application of the Brunn rule is "*This was accomplished assuming an equilibrium coastal profile response following the Bruun rule (Bruun, 1962), resulting in landward and upward displacement of the beach profile*" (P12 L11). We specify here that dunes preserve the pre-SLR shape when there's enough accommodation space, and the shape was cut where there wasn't enough space (right after the sentence). Then in the discussion we have added a paragraph about uncertainty associated to this choice. The included text is: "*When considering SLR-induced effects on time evolution of storm-induced risks, we have to take also into account existing uncertainties. Thus, the first uncertainty is related to the magnitude of the change itself. Here we have used the RCP8.5 SLR scenario but other scenarios could be possible (Church et al. 2013). The other source of uncertainty is controlled by the way in which this forcing is translated to the system. In this work we have assumed the Bruun rule to be valid and it was used to generate a morphological accommodation of the Tordera Delta site to SLR. Since there is no consensus on the best model to simulate this effect, other existing models and approaches (see e.g. Le Cozannet et al. 2014) could be tested and integrated in the BN to include this uncertainty. In any case, the effect of the uncertainty on the SLR projections may be larger than their associated morphological response.*".

RC. Specific comments

In the following specific comments are given to the paper (L = Line number; P = page number).

P4, L14

"wave-induced run-up" Includes wave setup? Any consideration of duration with regard to having water at a certain location?

AA. It is a general statement comparing the contribution of run-up (including set-up and swash, we will add this in brackets) to the total water level (astronomical tide + residual (surge) + run-up). We are not considering any time duration here, since tide and surge are never the direct cause of flooding in the NW Mediterranean, being this the wave-by-wave overtopping. Thus, the surge only plays the role of "lowering the freeboard of the beach some centimetres (tens in the worst case)", and was considered not to be significant enough compared to waves' contribution to include it as a variable in the BN (i.e. having multiple classes of sea level for the current situation).

P5, L19

"thresholds" How sensitive are the methods to the selected thresholds? Was this selection based on impact or purely on the forcing properties (offshore wave conditions)? The probability of extreme events with regard to the former and the latter are typically different.

AA. The authors are well aware of the different statistical results obtained by the event approach (selection based on storm characteristics) or the response approach (statistics based on impacts/hazards) and have studied its effect of inundation hazard statistical identification. The presented method is not sensitive to the thresholds to identify events the way it is applied in the sense that we are not assigning probabilities or return periods to a given inundation. We are integrating results from multiple scenarios by equally representing them (same storm simulations for each source characteristic state of variables). In this section we are only explaining how storm events are usually identified in the study sites, for the reader to know what a storm means in the Adriatic or in the NW Mediterranean.

In the text the reader is currently pointed to the corresponding references where these thresholds were derived, which are based on impact.

P8, L18

"XBeach model" How good was the calibration/validation?

AA. Currently we are providing only the reference to the study at the Tordera Delta were the validation is explained in detail. We have included a note at each study site paragraph describing briefly how good the validation was, in terms of Brier Skill Score, so the reader can have some additional info in the present manuscript and not only the references. See in the text: *"The model chain was validated through the St Esteve event in 2008, obtaining a Brier Skill Score of 0,682*

*for the morphological response of the emerged part of the beach (Sanuy and Jiménez, n.d.). Simulation results can be considered excellent for scores over 0.6 (Sutherland et al., 2004)"*

P9, L14

"intersecting" Meaning in this context?

AA. Polygon intersection, between receptors 2D layout and the Xbeach grid. This way we identify which grid nodes affect each receptor. We have clarified

P11, L16

"footprint" What is this?

AA. Is the receptor polygon layout in 2D. We now use "receptor limits in the ground".

P12, L14

"a directional change" But the wind did not change, right (L3)? What is causing this.

AA. The study reporting change in wave direction (Casas-Prat and Sierra, 2012) predicts the change in direction by applying statistics to the current past 60-year evolution of wave records, and obtaining the prediction of the future wave mean climates. Therefore, there is no information/evidence in that study linking that change to any specific forcing (wind, wave current interaction…). It is a scenario we wanted to explore as a "what if" future situation. In addition, we have changed the text from *"Other factors such as changes in storminess, wind, or waves were not expected to change significantly in the NW Mediterranean"* to *"Other factors such as changes in storminess, wind speeds, or wave high were not expected to change significantly in the NW Mediterranean"*

P13, L14

"winter dune" What is this?

AA. It's the name given to an artificial dune which is built every winter to protect beach concessions in Emiglia-Romagna (Italy). It is explained in the consecutive paragraph and we will cross-reference it

P18, L18

From here on some of the figure numbers are wrong. Please check.

AA. This will be addressed, since some figures have been supressed and the whole text on the results section reviewed and reduced.

P18

Some of the DRR measures taken seem to increase the risk. What is the explanation/logic behind this? Does it mean that the characterization of impact is not proper?

AA. In the Tordera Delta, this is the case when the DRR affects the hydrodynamics at the nearshore and or swash zone, and while protecting locally some receptors, but the erosion is increased down coast and other receptors get more exposed than before. Overall it can be observed how this increase isn't significant in any case. Nevertheless, the figure containing this effect will be supressed due to comment 1 and thus, it will not induce confusion to the reader or require further explanation by the authors.

In Lido degli Estensi and Spina, this is the case when the Winter Dune is close to the receptors it must protect, and it fails to prevent overwash. The measure increases water speed and can enhance scouring in such specific cases. We have added a small note about this phenomenon in the Results section of the Italian case: "*
[revised manuscript text omitted]

---

## Referee Report (RR1)

**Review on „Linking source with consequences of coastal storm impacts for climate change and risk reduction scenarios for Mediterranean sandy beaches "**

**by Marc Sanuy, Enrico Duo, Wiebke S. Jäger, Paolo Ciavola and José A. Jiménez**

Recommendation: Major Revisions

The authors present a decision support framework to assess the effect of risk reduction measures on impacts of coastal storms under current and future conditions. A Source-Pathway-Receptor-Consequences (SPRC) model is implemented in the form of a Bayesian Network and applied to two Mediterranean sandy coast case study areas.

The study is well written and structured, provides interesting insights and presents an interesting approach for decision support for coastal risk managers. In a previous review round, the reviewers identified 2 main aspects which require major revisions before the manuscript can be considered for publication:

1) The use of Bayesian Networks as implementation of a SPRC model in this study

2) The storm intensity used for the scenario testing

I acknowledge that the authors have addressed all the mentioned issues. However, some important information are still missing or should be rephrased/restructured to make all aspects of the study clear and comprehensible to the reader. This especially includes the use of the BN approach, which in the previous version of the manuscript has been a major source of confusion, which is still not entirely resolved in the current version. Therefore, I would recommend to accept this manuscript for publications only after the two main issues have been resolved.

**Specific Comments**

*1. Bayesian Network*

The authors use BNs as graphic implementation of a SPRC model, to use it for an intuitive communication of different risk reduction scenarios under different storm scenarios for decision makers. To my understanding the authors combine in their BN the spatial distribution of receptors (e.g. buildings) with spatial distribution of the boundary conditions (storm properties) to gain a deterministic joint distribution, where each bin represents the hazard at the location of the receptor and eventually of the consequences. Although the revised version of the manuscript does not provide detailed information about the advantages of BNs compared to established raster based GIS analysis, the general approach seems valid to me. One reason the authors gave in their response but did not mention it in the revised manuscript is that using the BN framework "facilitates the integration of multiple simulations when assessing scenarios". In case the authors are convinced that their approach has considerable advantages over established approaches, I recommend to clearly give reasons in the manuscript why this is the case (apart from the pragmatic reason that the framework was already there).

In general, I think the term "Bayesian Network" in this context is at least confusing and should be avoided. Bayesian Networks are generally described as representation of the probabilistic dependencies between a given set of random variables as a directed acyclic graph (DAG). Since the network in this study rather represents spatial than probabilistic dependencies, I would recommend renaming it to "Decision Network" or "Conditional (In)dependence Network" to avoid confusion. One of the main reasons for the application

of BNs in the domain of natural hazards is the representation of uncertainties through probabilistic inputs and outputs. As mentioned in the previous reviews, this study is not using BNs in a real probabilistic sense. That's why it should be made clear that uncertainty is only considered by using two different variations of each storm scenario and not by the model itself. Therefore, the manuscript should be carefully revised to avoid claiming that the network model incorporates uncertainty (i.e. last sentence of the conclusion). In addition, in Section 3.6 and 3.6.1 it should be made clear that the structure as well as the parameters of the network model were pre-defined by the user and not learned by an algorithm (at least that's how I interpret the model description).

2) Storms used for scenario testing

This issue was previously raised by Reviewer 1 and revised in the manuscript, but I found it still difficult to follow the point of the authors. According to the revised text in Section 4 and 5, all storm variables are described with a uniform distribution. However, it is not clear what a uniform distribution means in a deterministic setup. In order to get one result per scenario like shown in Figure 10 to 14 one would either have to set the boundary conditions (wave height, storm duration) constant or calculate each scenario for each wave height – storm duration combination and calculate the average. In order to make clear how the different storm simulations are used in the BN to generate the plots in Figure 10 to 14, I would recommend to include an example in the text for at least one of the plots. It should also made clear why these in total 48 storm scenarios described in Section 3.1 were calculated in the first place, as they don't seem to be considered in the generation of the results.

**Other comments**

*Figure 4:* I recommend changing the illustration in Figure 4 Panel (IV) since the dashed arcs between the nodes do not represent actual direct connections between the nodes and can potentially be confusing to the reader.

P.7 L. 14-22: What is the reason for simulating the 12 storm combinations exactly twice? I would assume that one would need more than 2 simulation runs to get an acceptable range in variability of storms. I also don't understand how this corresponds to the 16 recorded events. Why is it necessary to slightly change the parameters of the storm for the 12 combinations if you already have 16 measurement points with observed representations of storms?

P.9 L.4: Please provide information how the model was qualitatively validated.

P.23 L20-23: I think this conclusion cannot be made here, since the study compares the projected MSL for the year 2100 for the Spanish site and 2050 for the Italian site.

**Technical Corrections**

P.4 L.29: food service instead of restoration?

P.5 L.7: EWS = Early warning system?

P.7: Wrong table number

P. 8 L.19: "will be as good as the model is accurate"

P.8 L. 22: S-O-A: jargon, please revise

P.8 L23: …provided if they…

P. 12 L1: with stakeholders

P.12 L15: there is

P.12 L15: where it was not

P. 12 L21: switch of incoming storms

---

## Author Response (AR2)

Dear authors,

your revised manuscript has been carefully reviewed by a referee, considering in

particular the main critical comments of the first round of reviews. Although he/she

acknowledges clear improvements, there is still some confusion and lack of clear

presentation. I would like to ask you to address his/her comments in another revision.

What puzzles me in particular is that this reviewer, and reviewer 1 of the first round, are

both working with Bayesian Networks, and both could not understand the claimed

benefit of using Bayesian Networks in this context. Now this referee suggests that your

application may not be a Bayesian Network - in any way, this confusion needs to be

clarified.

Best regards, Bruno Merz

AA1. In what follows we address all comments and suggestions raised by the reviewer.

Review on "Linking source with consequences of coastal storm impacts for climate

change and risk reduction scenarios for Mediterranean sandy beaches"

by Marc Sanuy, Enrico Duo, Wiebke S. Jäger, Paolo Ciavola and José A. Jiménez

Recommendation: Major Revisions

The authors present a decision support framework to assess the effect of risk reduction

measures on impacts of coastal storms under current and future conditions. A Source-

Pathway-Receptor-Consequences (SPRC) model is implemented in the form of a

Bayesian Network and applied to two Mediterranean sandy coast case study areas.

The study is well written and structured, provides interesting insights and presents an

interesting approach for decision support for coastal risk managers. In a previous review

round, the reviewers identified 2 main aspects which require major revisions before the

manuscript can be considered for publication:

- 1) The use of Bayesian Networks as implementation of a SPRC model in this study
- 2) The storm intensity used for the scenario testing

I acknowledge that the authors have addressed all the mentioned issues. However, some important information is still missing or should be rephrased/restructured to make all aspects of the study clear and comprehensible to the reader. This especially includes the use of the BN approach, which in the previous version of the manuscript has been a major source of confusion, which is still not entirely resolved in the current version. Therefore, I would recommend to accept this manuscript for publications only after the two main issues have been resolved.

AA2. The authors thank the reviewer for his/her comments and suggestions. In what follows we answer all raised issues and specify how we have addressed them in the manuscript.

**Specific Comments**

**1. Bayesian Network**

The authors use BNs as graphic implementation of a SPRC model, to use it for an intuitive communication of different risk reduction scenarios under different storm scenarios for decision makers. To my understanding the authors combine in their BN the spatial distribution of receptors (e.g. buildings) with spatial distribution of the boundary conditions (storm properties) to gain a deterministic joint distribution, where each bin represents the hazard at the location of the receptor and eventually of the consequences.

AA3. The BN combines the spatial distribution of receptors (e.g. buildings) with the spatial distribution of the **hazards** (e.g. inundation water depth and erosion depth) to obtain a deterministic joint distribution. This distribution is linked through the BNs conditional probability tables with the storm properties (i.e., boundary conditions, which consist on multiple variables, such as wave height, duration, direction and water

level). Thus, this tool can be used to obtain hazard (and eventually consequence) joint distributions per receptor type and location conditioned to any possible combination of boundary conditions (which are discretized in bins). In the present application, these joint distributions are deterministic because the BN has been fed with a subset of events in which every combination of boundary condition's bins is equally represented (by two simulations). Thus, boundary conditions' unconstrained distributions in the BN are uniform. All this will be better motivated in the manuscript (see following answers) following the guidelines of the reviewer.

Although the revised version of the manuscript does not provide detailed information about the advantages of BNs compared to established raster based GIS analysis, the general approach seems valid to me. One reason the authors gave in their response but did not mention it in the revised manuscript is that using the BN framework "facilitates the integration of multiple simulations when assessing scenarios". In case the authors are convinced that their approach has considerable advantages over established approaches, I recommend to clearly give reasons in the manuscript why this is the case (apart from the pragmatic reason that the framework was already there).

AA4. The manuscript states in the introduction (p2. L28): *Using a BN approach, many multi-hazard results from process—oriented models can be integrated for joint assessment, combining different scenarios and alternatives (e.g. Gutierrez et al., 2011; Poelhekke et al., 2016), enabling the integration of socio-economic concepts (e.g. Van Verseveld et al., 2015).*

We have rephrased the sentence as follows: "The data assimilation capacity of BN approaches allows integrating many multi-hazard simulations from process—oriented models for joint assessment of different scenarios and alternatives (e.g. Gutierrez et al., 2011; Poelhekke et al., 2016), including also socio-economic concepts (e.g. Van Verseveld et al., 2015). This is an advantage compared to classical GIS-based approaches, which are more limited when combining large number of simulations in multiple subsets of scenarios."

And added this paragraph to section 3.6.:

In the present application, the BN-based approach is applied assuming no prior knowledge on the statistics of the source. Thus, all source variable combinations are equally fed into the BN resulting as uniform distributions of either Hs, duration, WL or direction. Each combination is represented by two simulations of slightly different storms to include some uncertainty due to intra-bin variability. No other uncertainty is included. Therefore, the present application is deterministic, a Bayesian-based Decision Network (BDN) which mainly uses the data assimilation capacity of the BN as principle advantage with respect to other methodologies (e.g. GIS-based assessments). Additionally, the BDN allows also reverse assessments, where output variables (i.e. consequences) can be constrained to get conditioned results on the source variables. In the Discussion section further guidance into a fully-probabilistic BN approach integrating multiple sources of uncertainty is presented."

In general, I think the term "Bayesian Network" in this context is at least confusing and should be avoided. Bayesian Networks are generally described as representation of the probabilistic dependencies between a given set of random variables as a directed acyclic graph (DAG). Since the network in this study rather represents spatial than probabilistic dependencies, I would recommend renaming it to "Decision Network" or "Conditional (In)dependence Network" to avoid confusion. One of the main reasons for the application of BNs in the domain of natural hazards is the representation of uncertainties through probabilistic inputs and outputs. As mentioned in the previous reviews, this study is not using BNs in a real probabilistic sense. That's why it should be made clear that uncertainty is only considered by using two different variations of each storm scenario and not by the model itself. Therefore, the manuscript should be carefully revised to avoid claiming that the network model incorporates uncertainty (i.e. last sentence of the conclusion).

AA5. We will stress out that the only considered uncertainty is the intra-bin variability. We will also stress that as a first illustrative application we have assessed this variability using 2 simulations per bin combination (see AA4 and second answer in "Other Comments").

About the use of the term "Bayesian Network": We are assessing spatial probabilities linked to (conditioned to) boundary conditions. In this work we have fed the BN with equal representation of all boundary conditions. But if the same set-up was fed with a dataset of recorded or hindcasted storms with their real frequencies in terms of boundary conditions, the tool would be fully-probabilistic. Thus, the presented structure has the potential to be a real probabilistic BN, but these will depend on the data used to feed it. In this sense, the reviewer is right, although the presented framework has the capability to work in probabilistic terms, the work showed in this study does not apply it. Therefore, we change the term, and we propose to use "Bayesian-based Decision Network (BDN)", in order to make the reader aware that our application is not fullyprobabilistic and it does not include other uncertainties than intra-bin variability. But we propose to keep using the concept "Bayesian Network-based (BN) approaches" in the introduction and general description of the methodology since we are actually using the data assimilation potential of the BN in order to build our decision support tool. This concept will guide better the reader through the Discussion, where we explain that with proper feeding the tool develops into a fully-probabilistic BN that can include multiple uncertainties.

We understand that last sentence in the conclusions is misleading (it doesn't point out to our application but rather to the overall potential of BN-based applications for coastal risk assessments). After the Discussion the reader knows how different sources of uncertainty can be integrated in the tool, although not included in the presented application. The sentence has been rephrased to "Finally, the developed framework has proven to be efficient to analyse storm-induced risks and strategies to cope with them. Moreover, a series of elements to be addressed to further improve it and to extend its applicability have been identified and discussed. In this sense, the BN approach is a versatile tool to make robust comparisons across different conditions".

In addition, in Section 3.6 and 3.6.1 it should be made clear that the structure as well as the parameters of the network model were pre-defined by the user and not learned by an algorithm (at least that's how I interpret the model description).

AA6. We include in section 3.6. the following sentence: "The variables and bin ranges characterising boundary conditions are pre-selected by the user".

**2) Storms used for scenario testing**

This issue was previously raised by Reviewer 1 and revised in the manuscript, but I found it still difficult to follow the point of the authors. According to the revised text in Section 4 and 5, all storm variables are described with a uniform distribution. However, it is not clear what a uniform distribution means in a deterministic setup. In order to get one result per scenario like shown in Figure 10 to 14 one would either have to set the boundary conditions (wave height, storm duration) constant or calculate each scenario for each wave height – storm duration combination and calculate the average. In order to make clear how the different storm simulations are used in the BN to generate the plots in Figure 10 to 14, I would recommend to include an example in the text for at least one of the plots. It should also be made clear why these in total 48 storm scenarios described in Section 3.1 were calculated in the first place, as they don't seem to be considered in the generation of the results.

AA7. What the reviewer describes is exactly what the BN does: the BN has been fed with equal representation of all boundary conditions (2 simulations per Boundary Condition combination). This means that when the BN variables are unconstrained, the BC are uniform and the output variables (i.e. consequences) are the average of all 96 simulations. Then in order to produce Figures 10 to 14 (now Figures 11 to 15) we constrain certain variables: the TWL is constrained to current TWL, incoming direction constrained to E and the risk reduction measures are constricted to None to produce the first bar of the CUS scenarios in the figures. This means the bar is the integration (average in this BN) of the results of 12 simulations. Another 12 (different) simulations are integrated in CC1 and 24 simulations (since we leave unconstrained the direction, and thus 50% E 50% S) are integrated to each CCS2 and CCS3 scenarios.

We are including additional text and a figure in section 4 (Results) to improve the description on how the BN integrates our results. See the whole general introduction of

Results Section before 4.1: "The results of scenario testing are provided for each case study through an integrated comparison of percentages of receptors at each level of flooding and erosion risks. This is done by comparing the risk levels under current and climate change scenarios, with and without measures. The results of the scenarios that will be presented in the following sections are produced by integrating in subsets all 96 simulations at each study site.

Figure 10 shows an example of the integration of simulations at the Tordera Delta considering the CUS without measures. The figure includes 3 boxes with different level of (un)constrained boundary conditions and corresponding results in terms of erosion risk to infrastructures. In box A, both Hs and storm duration are constrained to a specific bin (in this case given by the highest values) and thus, results of two different simulations are integrated to obtain the final output. In box B, Hs is unconstrained while duration is constrained to the highest bin. In this case, the final result is produced by integrating six simulations (two per each Hs bin). Finally, in box C, both Hs and Duration are unconstrained and the output is given by integrating 12 simulations (2 per each Hs and duration bin combination) which represent the overall dataset for CUS without measures for Tordera Delta.

The current BDNs have been fed assuming no prior knowledge on the boundary conditions' distributions (i.e. any boundary condition is uniform when unconstrained). This approach is adequate to explore scenarios and to assess the efficiency of protection measures in terms of impact reduction".

We are also adding additional info on the BN application in sections 3.6 and 3.6.1, clarifying the issues 1 and 2. See AA 4, 5 and 6.

**Other comments**

Figure 4: I recommend changing the illustration in Figure 4 Panel (IV) since the dashed arcs between the nodes do not represent actual direct connections between the nodes and can potentially be confusing to the reader.

AA8. As it can be seen in figures 8 and 9, the dependencies between variables meet those in Panel IV of figure 4. These connections exist, and any update on the knowledge of one of the variables affect the conditional probabilities with all the other variables connected to that. However, this comment has made us aware that the paragraph in section 3.6 citing the figure was misleading, leading to confusion.

In this sense, we clarified this point by editing the figure with all connections as continuous lines and we have rephrased reference to Figure 4 in section 3.6 as: "Hazard intensity is conditioned by the location of the receptors and the presence of measures. Consequences are conditioned by hazard intensity, receptor type and presence of measures".

P.7 L. 14-22: What is the reason for simulating the 12 storm combinations exactly twice? I would assume that one would need more than 2 simulation runs to get an acceptable range in variability of storms. I also don't understand how this corresponds to the 16 recorded events. Why is it necessary to slightly change the parameters of the storm for the 12 combinations if you already have 16 measurement points with observed representations of storms?

AA9. The reason is that in this work we have illustrated the process by using a limited number of simulations to reduce the required computational time. We have selected 12 storm combinations (within a given range (bin) of values) and each combination was represented by 2 different data within the corresponding range to include some intrabin variability. Since the 16 recorded events do not cover the total possible 24 storms needed to cover possible conditions for a sea level scenario, they were complemented

with 8 synthetic (not recorded but possible) events. Therefore, bin combinations can be composed of 2 historical (recorded) events, or 2 synthetic (not recorded but possible) events or 1 historical and 1 synthetic. The objective was to equally represent all boundary condition combinations to illustrate how the BN approach works on coastal risk assessments without prior knowledge on the (multi-variable) event frequencies (See answers AA4 to AA7).

This is clarified in section 3.2. where we introduce the following sentence: "In order to be used in a BN approach, storm characteristic variables must be discretized in ranges which define the resolution of the source description. In this application, used simulations cover uniformly all variable combinations, assuming no prior knowledge of their statistics".

At the same section we have rephrased as follows P7-L9: "Each combination of states is represented by two simulations of slightly different storms to account for potential variability within variable ranges, leading to a total of 24 simulations under the current MSL and 24 under SLR. Of the 24 simulations under current MSL, 16 correspond to historic (recorded) events including the two largest, which occurred in November 2001 and December 2008. These were classified as extreme storms (category V) according to the Mendoza et al. (2011). To include the full range of cases, the remaining 8 storms were completed by using combinations of Hs-duration-direction not previously recorded.".

Further information regarding and completing this issue is presented in answers AA4 to AA7.

**P.9 L.4: Please provide information how the model was qualitatively validated.**

AA10. We include additional info: "The model was qualitatively validated using observed inundation extension and profile beach response of the February 2015 event (Perini et al., 2015; Trembanis et al., n.d.).".

P.23 L20-23: I think this conclusion cannot be made here, since the study compares the

projected MSL for the year 2100 for the Spanish site and 2050 for the Italian site.

AA11. Reviewer is right. Using different temporal horizons in the assessment do not

permit to directly compare them to obtain conclusions in relative terms. Our point was

to illustrate the importance of the long term morphodynamic evolution due to changes

in MSL. To avoid this, this conclusion has been rephrased as follows: "The estimated risk

significantly increases for the climate change scenario. The morphological

accommodation response to the projected MSL, which was only included at the Tordera

Delta, was identified as a major process to be considered in the impact assessment to

properly account for modifications in erosion and inundation hazards".

**Technical Corrections**

P.4 L.29: food service instead of restoration?

AA. Changed

P.5 L.7: EWS = Early warning system?

AA. Changed

P.7: Wrong table number

AA. Changed

P. 8 L.19: "will be as good as the model is accurate"

AA. Rephrased

P.8 L. 22: S-O-A: jargon, please revise

P.8 L23: ...provided if they...

AA. Changed to "state-of-art".

AA. Rephrased

P. 12 L1: with stakeholders

AA. Corrected

P.12 L15: there is

AA. Corrected

P.12 L15: where it was not

AA. Rephrased

P. 12 L21: switch of incoming storms

AA. Corrected

[revised manuscript text omitted]
     | Storm Duration | Incoming direction | TWL(tide+surge) | Mean Sea Level           |
|----------|--------|----------------|--------------------|-----------------|--------------------------|
|          | (m)    | (h)            | (°N)               | (m)             | (MSL)                    |
| TORDER   | 2 to 3 | 6 to 30        | 30-135 (E)         | 0 to 0.6 m      | Current                  |
| A DELTA  | 3 to 4 | 30-65          | 135-220 (S)        | NC              | Current +0.73 m          |
|          | 4 to 5 |                |                    |                 | Morph. response included |
| LIDO     | 2 to 3 | 12 - 68        | 60 to 135          | 0.65 to 1.05    | Current                  |
| DEGLI    | 3 to 4 | NC             | NC                 | 1.05 to 1.45    | Current+0.30 m           |
| ESTENSI- | 4 to 5 |                |                    | 1.45 to 1.85    | No morph. response       |
| SPINA    | 5 to 6 |                |                    |                 |                          |

Table\_23: Distribution of receptors at the Tordera Delta study site.

| Area                               | No. of Houses | No. of Campsite Elements |
|------------------------------------|---------------|--------------------------|
| Area 1 (0 to 20 m Malgrat de Mar)  | 16            | 45                       |
| Area 2 (20 to 50 m Malgrat de Mar) | 10            | 71                       |
| Area 3 (50 to 75 m Malgrat de Mar) | 8             | 169                      |
| Area 4 (> 75 m Malgrat de Mar)     | 46            | 509                      |
| Area 5 (0 to 20 m Blanes)          | 1             | 95                       |
| Area 6 (20 to 50 m Blanes ) | 4             | 156                      |
| Area 7 (50 to 75 m Blanes ) | 7             | 72                       |
| Area 8 (> 75 m Blanes )     | 51            | 189                      |
| Total                              | 143           | 1306                     |

Table  $\underline{33}$ : Vulnerability relations for houses and campsite elements at the Tordera Delta study site with and without Flood Resilience Measures (FRM).

| Water depth at the receptor | Relative Damage (%) |           |              |                 |
|-----------------------------|---------------------|-----------|--------------|-----------------|
| (m)                         | Houses              | Campsites | Houses - FRM | Campsites - FRM |
| 0                           | 0                   | 0         | 0            | 0               |
| 0-0.3                       | 18.3                | 50        | 0            | 0               |
| 0.3-0.6                     | 26.5                | 71        | 18.3         | 50              |
| 0.6-0.9                     | 33.2                | 82        | 18.3         | 50              |
| 0.9-1.5                     | 44.7                | 89        | 26.5         | 71              |
| 1.5-2.1                     | 54.1                | 91        | 33.2         | 82              |
| 2.1-3.0                     | 64.5                | 100       | 44.7         | 89              |
| 3.0-4.0                     | 71.2                | 100       | 54.1         | 91              |
| 4.0-5.0                     | 75                  | 100       | 64.5         | 100             |

Table\_44: Distribution of the receptors at Lido degli Estensi and Lido di Spina.

| Area                          | Residential and Commercial Buildings | Concessions |
|-------------------------------|--------------------------------------|-------------|
| Lido degli Estensi - Seafront | 26                                   | 16          |
| Lido di Spina - Seafront      | 47                                   | 28          |

Table 5: Vulnerability relation for flooding adopted for the receptors at Lido degli Estensi-Spina without (A) and with Flood Resilience Measures (B).

| Flood Depth [m] | Flood Relative Damage Factor [-]           |                                  |  |
|-----------------|--------------------------------------------|----------------------------------|--|
|                 | A - adapted from Scorzini and Frank (2015) | B - modified considering the FRM |  |
| 0               | 0                                          | 0                                |  |
| <0.3            | <0.1                                       | <0.1                             |  |
| 0.3 - 0.7       | 0.1 - 0.2                                  | <0.1                             |  |
| 0.7 - 1.1       | 0.2 - 0.3                                  | 0.2 - 0.3                        |  |
| >1.1            | >0.3                                       | >0.3                             |  |